# Time-dependence of Heterogeneous Ice Nucleation by Ambient Aerosols:   Laboratory Observations and a Formulation for Models

Jonas Jakobsson[1] (JJ), Deepak Waman[2] (DW), Vaughan T. J. Phillips[2]* (VTJP), Thomas Bjerring Kristensen[1] (TBK)

[1] Division of Nuclear Physics, Department of Physics, Lund University, Lund, Sweden
[2] Department of Physical Geography and Ecosystem Science (INES), Lund University, Lund, Sweden

*Correspondence to*: Vaughan Phillips (Vaughan.phillips@nateko.lu.se)

**Abstract.** The time dependence of ice-nucleating particle (INP) activity is known to exist, yet for simplicity it is often omitted in atmospheric models as an approximation.  Hitherto only limited experimental work has been done to quantify this time dependency, for which published data are especially scarce regarding ambient aerosol samples and longer time scales.

In this study, the time dependence of INP activity is quantified experimentally for six ambient environmental samples.  The experimental approach includes a series of hybrid experiments with alternating constant cooling and isothermal experiments using a recently developed cold-stage setup called the Lund University Cold-Stage (LUCS).  This approach of observing ambient aerosol samples provides the optimum realism for representing their time dependence in any model. Six ambient aerosol samples were collected at a station in rural Sweden representing aerosol conditions likely influenced by various types of INPs: marine, mineral dust, continental pristine, continental polluted, combustion-related and rural continental aerosol.

Active INP concentrations were seen to be augmented by about 40% to 100% (or 70% to 200%), depending on the sample, over 2 (or 10) hours.  Mineral dust and rural continental samples displayed the most time-dependence. This degree of time dependence observed was comparable to, but weaker than, that seen in previous published works.  A general tendency was observed for the natural timescale of the freezing to dilate increasingly with time. The fractional freezing rate was observed to decline steadily with the time since the start of isothermal conditions following a power-law.  A representation of time dependence for incorporation into schemes of heterogeneous ice nucleation that currently omit it is proposed.

Our measurements are inconsistent with the simplest purely stochastic model of INP activity, which assumes that the fractional freezing rate of all unfrozen drops is somehow constant and would eventually over-predict active INPs. In reality, the variability of efficiencies among INPs must be treated with any stochastic theory.

# 1 Introduction

The presence of ice-nucleating particles (INPs) of aerosol has been shown to influence cloud formation and cloud properties (Phillips *et al*. 2003; Cantrell and Heymsfield 2005; Boucher *et al.* 2013; Kudzotsa 2014; Kudzotsa *et al.* 2016; Phillips and Patade 2022), precipitation (Lau and Wu 2003; Lohmann and Feichter 2005) and thereby both local and global weather systems and climate (Gettelmann *et al.* 2012; Murray *et al.* 2012; Kudzotsa 2014; Storelvmo 2017; Schill *et al.* 2020ab; Sanchez-Marroquin *et al.* 2020). Even though INPs have been studied for many decades, some aspects of their influence are still not fully understood (DeMott *et al.* 2011). One aspect where much uncertainty remains is the relevance of time for their ice nucleation and related atmospheric ice processes in clouds.

According to the Intergovernmental Panel of Climate Change (Stocker *et al.* 2013) much of the uncertainty in projections of climate change by current global models is associated with the effects from atmospheric particles involving aerosol-cloud-radiation interactions. Recent reviews indicate that a large degree of uncertainty still prevails (Bellouin *et al*. 2020; Sherwood *et al*. 2020). The mechanisms for aerosol interactions with cold clouds have been explored with cloud models and are complex (Kudzotsa *et al.* 2016), as is also true globally (Storelvmo 2017). Ice in clouds is potentially influential for the climate because most of the volume of the atmosphere is at subzero temperatures. At any given moment, only a small fraction of all condensed water in the atmosphere resides in the form of ice crystals. Yet this small fraction has a disproportionately large impact on global precipitation, which is mostly associated with the ice phase (Field and Heymsfield 2015; Mülmenstädt *et al.* 2015), affecting also global radiative fluxes governing climate (DeMott *et al.* 2010).

An emerging area of interest is ice initiation, for which there are many possible pathways in real clouds (Cantrell and Heymsfield 2005; Phillips *et al.* 2007, 2020). Some mechanisms for ice formation involve fragmentation of pre-existing particles, such as by shattering of freezing raindrops or breakup in ice-ice collisions (Field *et al.* 2017). This is termed 'secondary ice production' (SIP). Also, homogeneous freezing of cloud droplets in the atmosphere normally occurs at about -37 °C (Heymsfield *et al*. 2005). Heterogeneous freezing can occur at much warmer temperatures by the action of INPs, which are rare, usually solid, aerosol particles catalysing ice formation. INPs can be influential since they initiate crystals that can grow to become snow and graupel, which may melt, forming the '*ice crystal process*' of precipitation production (Rogers and Yau 1989).

The first ice in any mixed phase cloud is from activation of INPs, if its top is below the level of homogeneous-freezing (about -36 °C depending on drop size; Pruppacher and Klett, 1997 ['PK97']). These have variable chemical composition, concentrations and activities in nature (Knopf *et al.* 2021). Mineral dust particles (e.g. from deserts) and soil dust particles may efficiently act as INPs of relevance to mixed-phase clouds (e.g. Kanji *et al*., 2017; DeMott *et al.*, 2018). A range of primary biological aerosol particles (PBAPs), such as bacteria, viruses, marine exudates, phytoplankton, fungal spores, pollen, lichen

and plant fragments, may initiate immersion freezing, potentially at relatively high temperatures, even above -10 °C (e.g.
Szyrmer and Zawadski, 2007; Morris et al. 2014; Kanji *et al.*, 2017). It is less clear to what extent combustion emissions may
play a role as INPs in mixed-phase clouds. Fresh and photochemically aged 'modern' car engine emissions do not appear to
contain significant concentrations of immersion freezing INPs (Chou *et al.*, 2013; Schill *et al.*, 2016; Korhonen *et al.*, 2022).
On the other hand, ships may emit INPs (Thomson *et al.*, 2018). Soot particles comprise a fraction of the immersion freezing
INPs in biomass combustion aerosol (Levin *et al.,* 2016) and the soot particle properties of relevance are likely to depend on
the combustion conditions (Korhonen *et al.*, 2020). A fraction of the organic compounds emitted from biomass combustion
may also initiate immersion freezing (Chen *et al.*, 2021). In addition, fly ash (e.g., from coal fired power plants) may act as
immersion freezing INPs (e.g. Umo *et al.,* 2019).

For very thin wave clouds that are 'mixed-phase' (ice and liquid) without precipitation, number concentrations of ice crystals
in-cloud are observed to be similar to those of active INPs in their adjacent environmental inflow (Eidhammer *et al.* 2010). In
such cold clouds, the chemistry and loading of INPs in the environment must affect the cloud-microphysical properties since
INPs then influence the ice concentrations observed, which in turn must influence the humidity (Korolev 2007) and extent of
any supercooled cloud-liquid. More generally for deeper clouds, some of the other mechanisms for ice initiation noted above
are potentially more prolific (Cantrell and Heymsfield 2005; Field *et al.* 2017). Anyway, it is beneficial to simulate the first
ice in mixed phase clouds accurately if detailed models are to represent the subsequent ice-microphysical processes adequately.

There exists a paradox in observations of mixed-phase stratiform glaciated clouds. Westbrook and Illingworth (2013)
observed that the light ice precipitation from such a thin layer-cloud (mixed-phase from about -11 to -13 °C) over UK persisted
for about a day and was produced by the ice crystal process. They argued qualitatively that this longevity could not be explained
by mixing of environmental INPs into the layer cloud as the vertical motions were very weak and the cloud top level was
constant, although they did not quantify the turbulent fluxes of INP entrained from the environment. They hypothesised that
time dependence of the activity of INPs was the cause. Despite in-cloud temperatures being practically isothermal, the
stochastic nature of INP activity was supposed to create a weak yet persistent long-term source of primary crystals from this
time-dependence over times of many hours. However, their interpretation of the observations with this hypothesis has been
contested by Ervens and Feingold (2013), who instead suggested an alternative explanation of in-cloud vertical motions and
weak turbulence continually activating INPs. The issue has not been resolved conclusively.

The first lab studies of time dependence of freezing began in the 1950s (reviewed by PK97). Two categories of models were
proposed to explain the lab data, one with ('*the stochastic hypothesis'* that eventually became classical nucleation theory),
(Bigg 1953ab) and one without time dependence (*'the singular hypothesis',* sometimes referred to as '*the deterministic
model'*), (Langham and Mason 1958). The singular hypothesis is an approximation and treats ice nucleation as a process
occurring on active sites that become active instantaneously at distinct conditions that vary statistically among the INPs. An

ice crystal is supposed to be initiated immediately when an INP's characteristic conditions of freezing temperature and humidity are reached, as if it were a digital switch. This neglect of time dependence yields a simple dependence of primary ice initiation on thermodynamic vertical structure of the environment.

Classical or stochastic theory assumes that embryonic ice clusters are continuously forming and disappearing (reviewed by PK97) at the interface of immersed aerosol particles. This is assumed to occur with a frequency that depends on the temperature. If an ice embryo reaches a critical size of stability, determined by the features of the surface, then ice is nucleated. Hypothetically during isothermal experiments, a population of INPs with identical composition and size, and with only one INP per drop, would produce an exponential reduction of the number of unfrozen drops in a sample, with the fractional rate of change of the number of unfrozen drops being constant with time according to stochastic theory. This constancy is seldom observed for immersion freezing (PK97). In reality there is a probability distribution of efficiencies of active sites among INPs of any given aerosol species, even for a population of identically sized particles (Vali 1994, 2008, 2014; Marcolli *et al.* 2007).

Although modern lab observations have confirmed the existence of time dependence of INP activity, nevertheless the singular hypothesis is still used in most cloud models owing to its validity as an approximation to the leading order behaviour of crystal initiation. Temperature is observed to produce a far stronger dependency than time for ice nucleation. Moreover, classical stochastic theory ('classical nucleation theory') can be difficult to represent because it is complex requiring many parameters for various components of INPs. Classical nucleation theory can easily over-predict active INPs by orders of magnitude at long times if such statistics are not properly considered (e.g., Vali 1994, page 1854 therein; Vali and Snider 2015). There is controversy about whether stochastic/classical schemes or singular approaches are more realistic (Fan *et al.* 2019).

In recent decades, laboratory experimental work to investigate time dependency of heterogeneous ice nucleation has been scarce. This is especially true for ambient environmental aerosols. Vali (1994) reported his earlier results for a small number of isothermal experiments (4 simple isothermal experiments, and 4 with a brief warming of the sample by either 0.5 K or 1.3 K just before the isothermal period) with an isothermal period of 10-15 min on samples described as "*distilled water containing freezing nuclei of unknown composition*", and isothermal temperatures between -16 and -21 °C. They concluded that the rate of freezing was dependent both on temperature and on time (see also Vali and Stansbury 1966). Welti *et al.* (2012), investigated aerosolized kaolinite particles in the immersion mode with the Zürich Ice Nucleation Chamber (ZINC) and observed time dependence by altering the flowrate through the instrument. They concluded that immersion freezing is at least partly a stochastic phenomenon, and recommended that time dependence should be included in numerical calculations of the evolution of mixed-phase clouds. Herbert *et al.* (2014) observed that more efficient INPs have a steeper gradient of nucleation rate coefficient (per unit surface area per second) with respect to temperature and hence a weaker effect from time-dependence. Herbert *et al.* found that the shift in temperature for a given fraction frozen of an INP population when the cooling rate is

changed is independent of temperature and governed by the temperature gradient of the nucleation-rate coefficient of the INP material.

Wright and Petters (2013) did experiments with Arizona Test Dust (ATD) with a droplet freezing array for various cooling rates between 0.1 and 5 K min⁻¹. They also included data for a total of two isothermal experiments (13.3 and 15.9 hours respectively). They concluded that their results implied a limited effect from time dependence equivalent to a few degrees of error in the freezing temperature of aqueous droplets. Their observations of time-dependence agreed with a modified stochastic model (expressed in terms of an areal nucleation rate) with only one component and variable INP surface area per drop by Alpert and Knopf (2016). Equally, Budke and Koop (2015) performed experiments with the commercially available snow inducer, Snowmax®, which was derived from *Pseudomonas syringae*, in the Bielefeld Ice Nucleation ARraY (BINARY). The technology applied in the present study is similar to that of BINARY. Budke and Koop measured time dependency with experiments for cooling rates ranging between 0.1 and 10 K min⁻¹ and were able to show a weak dependency on time for Snowmax®. Knopf *et al*. (2020) investigated time-dependent freezing of illite for up to 2 hours and confirmed that classical nucleation theory applies to represent its stochastic freezing behaviour, provided the variability of immersed INP surface area in each drop is accounted for. Similarly, Peckhaus *et al*. (2016, their Figure 7a) observed feldspar in drops in isothermal experiments for up to an hour and observed an approximate doubling of frozen fractions. The observed time-dependence was predicted by a multi-component CNT model (Niedermeyer *et al.* 2014).

None of these aforementioned studies investigated the time-dependence of real environmental aerosol. Equally, no experiments studied time periods longer than a few minutes, except for Vali (1994), Wright and Petters (2013), Peckhaus *et al.* (2016) and Knopf *et al.* (2020). This lack of lab observations of freezing at long times (> 10 mins) has allowed a profusion of implementations of classical nucleation theory without much data to constrain them, especially for real tropospheric aerosols. Only a limited degree of time dependency has been observed in the few salient lab studies hitherto. For example, an enhancement by about 50% in numbers of active INPs during the initial 20 s for a frozen fraction of about half was measured by Welti *et al*. (2012). Yet Vali (1994) observed this 50% change after 5-10 min.

Some recent studies confirm the importance of studying environmental aerosol samples, when applicability of results to real clouds is sought. Beydoun *et al*. (2016) showed that active site density (number of active sites per unit of surface area) schemes derived from lab observations over-predict the activity of small INPs of sizes typically found in cloud-droplets in real clouds. Equally, gradual 'drift' and sudden jumps in freezing temperature in repeated cycles of freezing and cooling for a given INP were sometimes observed (Vali 2008; Wright and Petters 2013; Kaufmann *et al.* 2017). This implies there may be non-stochastic features of time-dependence in the context of cycles of re-freezing. Such cycles might be expected to happen in nature during long-range transport of INPs in the real atmosphere as aerosols enter and leave clouds.

Such controversy about the extent of time dependence in ambient INPs and the reasons underlying it motivates the present study. We aim to use an experimental approach to quantify time dependency for ambient environmental samples and to suggest a way to represent it in atmospheric models. Our rationale here is that the optimum approach is to study ambient aerosol sampled directly from the environment, if the time dependence of INP activity in atmospheric clouds is to be understood.

The empirical parametrization of heterogeneous ice nucleation by Phillips *et al.* (2008, 2013) followed a similar approach by treating the dependency of active INPs on chemistry, size and loading of aerosol species in terms of field observations of the background troposphere. Studying ambient aerosol samples provides the optimum representativeness of the aerosols observed, conferring realism on the cloud models that use the inferred schemes. On the other hand, there is an inevitable cost from lack of certain identification of the precise chemical species initiating the ice in observed samples.

## 2 Method

### 2.1 Overview

There were three major stages to the experimental work performed in this study. Firstly, aerosol samples were collected for a period of about a year (from 28 February 2020) at the Hyltemossa research station, which is located in southern Sweden. Aerosol data from various instruments were also collected at the research station, and nearby, during this period.

Secondly, the collected data were analysed to identify candidate samples that could be dominated by, or at least possibly influenced by, aerosol particles representing six broadly defined aerosol classes:

- Marine dominated aerosol
- Mineral dust influenced aerosol
- Continental pristine aerosol
- Continental polluted aerosol
- Combustion dominated aerosol
- Rural continental aerosol

Thirdly these candidate samples were analysed with respect to ice nucleation activity with a combination of experiments for both continuous cooling rates and isothermal experiments for more than 10 hours.

The experimental setup enables automated control of the evolution over time of temperature of the freezing array for many hours (> 10) with minimal risk of contamination. As noted below, any sample may be exposed to repeated freezing experiments with high precision (Sec. 2.3). This enables fresh questions to be addressed about time dependence of ice nucleation in natural clouds.

## 2.2 Selection and characterization of samples

### 2.2.1 Sample collection

Ambient air samples were collected at the Hyltemossa research station in southern Sweden (56°06'00"N 13°25'00"E). Hyltemossa is in a forested area in the ACTRIS network. Daily air samples were collected with a continuous sequential filter sampler (model SEQ47/50-RV, Sven Leckel Ingenieurbüro GmbH, Berlin, Germany) with a $PM_{10}$ inlet and flowrate of 1 $m^3$/h.

The samples were collected on 47 mm polycarbonate track-etched membrane filters with 0.4 μm pore size of high collection efficiency (Zíková *et al.* 2015). There was 24-hour sampling for each filter and filter changes were at midnight. Because of the high pressure drop over the membrane filters, not all filters were able to achieve a full 24-hour sampling. This issue limited the selection of available samples, but the sampling coverage was deemed sufficient for this study. Filters were retrieved from the field station every 1-2 weeks, placed in sterile petri-slide filter cassettes and stored at a temperature of about -20°C until

analysis. Field blank samples were collected and handled in a manner identical to that for the sampled filters.

### 2.2.2 Sample classification according to likely dominant composition of INPs

2.2.2.1 Observational methodology

Many aerosol samples were collected as noted above (Sec. 2.2.1). Some of these were classified into six basic aerosol types when possible, as follows (section 2.1). Many samples could not be classified, however. In this study, we aimed at selecting samples likely to be dominated by different INP types at least of relevance to Northern Europe, and most likely of wider spatial-temporal relevance. In Table 1, we present supportive aerosol data related to the six selected samples.

Black carbon (BC) concentrations were measured with an Aethalometer (Model AE33, Magee scientific, Ljubljana, Slovenia) at Hyltemossa. The absorption at 880 nm was used to estimate the BC concentration. The chemical composition of non-refractory particulate matter with particle diameter below 1 μm ($PM_1$) was measured online with an aerosol chemical speciation monitor (ACSM) (Aerodyne) (Ng *et al.*, 2011). The default ACSM components are reported in this study. Fluorescent and total particle concentrations in the diameter range from 1 to 10 μm and 0.5 to 10 μm, respectively, were measured with a Bio

Trak (TSI) instrument during the last part of the sampling period. Particle induced X-ray emission (PIXE) was applied on some

of the filter samples to obtain the $PM_{10}$ concentrations of elements such as e.g. Cl, Ca and Si (Swietlicki *et al.*, 1996). The PIXE measurements were corrected for a blank filter background. Concentrations of particulate matter with diameters below 1 ($PM_1$) and 10 μm ($PM_{10}$) respectively were measured with an optical particle counter (OPC) (model Fidas 200, Palas GmbH, Karlsruhe, Germany) at the nearby Hallahus site (56°04'25" N, 13°14'88" E) at Svalöv. According to the manufacturer, the

instrument covers the size range for particles larger than 0.18 μm in diameter, and it is equipped with a heated inlet (intelligent aerosol drying system) to ensure reasonably low levels of relative humidity. The instrument is approved for ambient air quality measurements, and the measurements at Hallahus are part of the European Monitoring and Evaluation Programme (EMEP). In addition, air mass back trajectories were inferred with the online HYSPLIT model (Stein *et al.*, 2015; Rolph *et al.*, 2017) and they are presented in Fig. 1. The simulations were based on the 1-degree GDAS meteorological data and vertical motion

was modeled. Trajectories arriving at Hyltemossa 50, 500 and 2000 m above ground level (AGL) were obtained for every 6 hours. The trajectories presented in Fig. 1 arrived at 500 m AGL, and similar horizontal wind directions were observed at lower and higher altitudes.

The obtained physico-chemical aerosol properties for the studied samples are presented in Table 1. Before describing the

characteristics of the samples it is worthwhile to point out some general aspects of and limitations to the reported properties. The filter samples from 26 March 2020 and 23 February 2021 had been used up in prior analysis, so it was not possible to perform PIXE on those samples. Instead, a filter sample from 25 February 2021 was analysed with PIXE, since most other meteorological and aerosol properties were very similar to what was observed for the 23 February 2021 filter sampling period (see Table 1). It was not possible to select another filter sample fully resembling the meteorological, aerosol and INP properties

as observed for the 26 March 2020 sample.

The Palas OPC was not functional during most of February 2021. Hence, we do not have direct measurements of $PM_{10}$ and $PM_1$ of relevance for two of the filter samples included in Table 1. However, we have included estimates of the $PM_{10}$-$PM_1$ fraction based on the Biotrak OPC measurements. Spherical particles and a particle density of 2.0 g/cm$^3$ were assumed in the

calculations since that gave rise to a good agreement between the inferred $PM_{10}$-$PM_1$ fractions when data from both OPCs were available. The $PM_{10}$-$PM_1$ fraction in southern Sweden has previously been reported to be dominated by sea salt and/or dust particles (Kristensson, 2005) but we would suggest that fly ash potentially also could contribute to the $PM_{10}$-$PM_1$ fraction in that region.

In Table 1, we report BC concentrations inferred from the optical aethalometer measurements. However, it is well known that other particle components (e.g., Saharan dust) may bias the inferred 'BC' levels high (Fialho *et al.*, 2005), and we did not have a reliable algorithm to isolate the potential dust absorption in the current study. The ACSM cannot reliably detect potassium concentrations and non-refractory components (e.g. sea salt and mineral dust). Hence, the Cl concentration measured with the ACSM is typically not associated with sea salt.

Before interpreting the supportive aerosol chemical properties, we will summarise the chemical characteristics of (i) marine aerosol, (ii) Saharan dust particles and (iii) biomass combustion emissions. Marine aerosol is characterized by varying concentrations of sea salt, non-sea salt (NSS) sulfate and organic compounds. The supermicron size range is typically highly dominated by sea salt, while the submicron size range may be dominated by NSS sulfate and organics depending on the season and biological activity (O'Dowd et al., 2004). Saharan dust is composed of different minerals involving the elements, Si, Ca and Al (Murray et al. 2012; Kaufmann et al. 2016; Boose et al. 2016). Linke et al. (2006) reported varying compositions between four different Saharan dust samples. On an elemental mass-basis, roughly 50% of the mineral dust was comprised of O, while about 25-37%, 1-17% and 4-6% were comprised of Si, Ca and Al, respectively (Linke et al., 2006). Biomass combustion emissions depend highly on the composition of the fuel and the combustion conditions. The emitted aerosol typically include BC, organic and inorganic compounds. Species such as $KCl$, $K_2CO_3$, $KNO_3$ and/or $K_2SO_4$ may be present in the submicron size range while fly ash particles comprised of Al-, Ca- and/or Si-oxides may be present in the supermicron size range (e.g., Obernberger et al., 2006; Obaidullah et al., 2012)

In summary, comparison of Fig. 1 and Table 1 reveals how regional to long-range atmospheric transport radically controls aerosol properties at the Hyltemossa site. Changing weather patterns cause wide variability in the physico-chemical aerosol composition.

2.2.2.2 Observed aerosol composition of samples

In the following, we will describe the characteristics of the various aerosol samples based on results presented in Table 1 and Fig. 1. We will start out by describing the samples that were easier to classify. These descriptions will be followed by an overall discussion of the more likely INP types to be present in the different samples.

The marine sample is characterized by a relatively low BC concentration, modest concentrations of organic matter and sulfate dominate the measured $PM_1$ components. A significant concentration of Cl was detected with PIXE, while non-refractory Cl in the $PM_1$ fraction did not appear elevated. If the composition of standard sea salt was assumed (Millero et al., 2008) then the majority of the detected Ca and K were present in the sea salt. Hence, this sample was very characteristic of the marine aerosol during periods with some biological activity and sea salt was highly likely the dominant component by mass. A relatively low to moderate concentration of Si was inferred. However, pronounced levels of Si was detected in the blank filter which resulted in pronounced random errors associated with relatively low sample levels of Si. The reported Si concentration of 0.20 μg/m³ was associated with a random error of 22% corresponding to one standard deviation. Hence, we cannot exclude a minor dust component to be present, but then likely a dust component with low levels of Ca and K relative to Si.

The combustion sample is characterized by elevated levels of $PM_1$, BC, organic matter, non-refractory chloride, nitrate and sulfate, which all potentially could be associated with (residential) biomass combustion. 7 December was during the residential heating season and the air masses had previously passed over Eastern Europe (Fig. 1) which was the likely main source region. Significant levels of the elements Ca, Si and K were observed and it was not possible to determine to which extent those were indicative of dust or fly ash particles since those three elements may be present in both. However, the $PM_{10}$-$PM_1$ level, which was likely to be dominated by dust or fly ash was rather low (3.0 µg/m$^3$) relative to the $PM_1$ (18 µg/m$^3$). Hence, this sample was by mass highly dominated by components which all potentially could be associated with combustion – and a potential dust component would only comprise a low mass fraction. It is also worth pointing out that the PBAP concentration of 3.5 L$^{-1}$ may not be negligible in an INP context.

The dust-dominated sample was chosen with the expectance of pronounced concentrations of Saharan dust. According to the online dust forecast from the University of Athens ('Skiron'), Saharan dust is expected in the boundary over or in the vicinity of Southern Sweden from 20 to 26 February. The Bio Trak OPC data show that the number concentration of supermicron particles started to become elevated on 18 February with a more stable high plateau being reached on 20 February. The maximum concentrations of supermicron particles appeared from 23 February until midnight between 26 and 27 February. The sample from 23 February was selected for time-dependent INP measurements, and the entire sample was used up in the analysis, which did not allow for PIXE analysis. As can be observed from Table 1, many of the aerosol properties were comparable between the samples from 23 and 25 February, and the latter sample was selected for PIXE analysis. Both samples were collected over 5-6 hours and are characterized by high levels of estimated $PM_{10}$-$PM_1$, BC and organic matter. The sample from 25 February appeared to contain approximately twice as much $NH_4$ and $NO_3$ relative to that from 23 February, but all other measured aerosol components appeared comparable in magnitude. Relatively high concentrations of Si, Ca and K were indeed present in the sample from 25 February along with a very low concentration of Cl. These observations suggest that mineral dust levels were elevated. The total mass of the detected dust would likely be a factor of 3-4 times higher than the Si mass if a composition similar to the ones reported by Linke *et al*. (2006) was assumed. The estimated dust concentration of 4-6 µg m$^{-3}$ does not appear to explain all the $PM_{10}$-$PM_1$ estimated to be 14 µg m$^{-3}$. However, since the Cl concentration is very low, it seems highly likely that the elevated levels of $PM_{10}$-$PM_1$ in both samples from 23 and 25 February are directly associated with Saharan dust, while a contribution from European dust or fly ash cannot be excluded based on these measurements.

The 'continental polluted' sample appears similar to the 'combustion dominated' sample in terms of many properties including the BC, Organics, $PM_1$ and the back trajectories passing over Eastern Europe during the residential heating season in March. Much of the $PM_1$ can be associated with combustion emissions. Yet the $PM_{10}$-$PM_1$ level is significantly higher for the 'continental polluted' sample (7.8 versus 3.0 µg m$^{-3}$). The $PM_{10}$-$PM_1$ time series (not shown) peaks in the local afternoon, and we speculate that it in part is associated with local soil dust. Hence, it is not entirely clear whether (soil) dust and/or potentially combustion emissions may dominate the INP population in that sample.

The 'rural continental' sample (Sec. 2.1) is characterised by very low to low levels of most measured components such as $PM_1$, BC, $NO_3$, $SO_4$, Si, Ca, K and an intermediate level of $PM_{10}$-$PM_1$ (4.7 µg m$^{-3}$) and Cl. Thus, sea salt likely contributed to the $PM_{10}$-$PM_1$ while the dust concentration must have been negligible to very small. In light of the air mass back trajectories, it is not possible to tell to which extent the sources of organic matter and $SO_4$ could be land-based in Denmark/Norway or marine. The 'continental pristine' sample (Sec. 2.1) is characterised by very low to low levels of all the measured aerosol components including $PM_1$, $PM_{10}$, BC, $NO_3$, $SO_4$, Cl, Si, Ca, K, Cl and PBAPs. Hence, there are no clear indications of pronounced anthropogenic, biogenic or marine emissions in the sample. Hence, the classification as continental pristine appeared suitable.

In general, there was a clear tendency of the optically measured $PM_1$ to be lower than what was obtained from the summation of individual chemical components detected in the $PM_1$ fraction. We mainly ascribe the offset between the different measurement approaches to (i) the size range up to 0.18 µm not being detected optically, and (ii) the heated inlet before the optical measurements, which may lead to evaporation of (semi-) volatile aerosol particle components. The latter effect is likely to be more pronounced when nitrate species and (semi-) volatile organic species contribute significantly to the $PM_1$ (e.g. Huffman *et al.*, 2009).

A discussion on the potentially dominant INP types in the respective samples will be presented in context of the ambient INP concentrations below (Sec. 3.1.1).

### 2.2.3 Sample preparation

The sampled filters were cut and a half filter was placed in sterile cryogenic vials while the other half of the filter was refrozen and stored. Two mL of ultra-pure water (18.2 MΩ, <3 ppb TOC) was added to the vials and the sample was shaken at highest effect for 3 minutes on a laboratory vibrating vortex shaker. All sample preparation and handling of sampled filters were done in an ultra-clean environment in a laminar airflow cabinet, and all pipetting of sample and water was done with sterile pipette tips, discarded after single use. Field blank samples were treated identically to the collected filter samples.

### 2.3 Experimental apparatus for measuring the ice activity of the samples

The freezing apparatus used to perform the experimental work in this study was designed using elements inspired from several previously described similar cold-stage setups (Wright and Petters 2013; Budke and Koop 2015) and was named the Lund University Cold-Stage (LUCS). A schematic overview of the freezing array and the LUCS system is shown in Figure 2.

In LUCS one hundred 1 µL sample drops are dispersed on siliconized hydrophobic glass slides, mounted in a freezing assembly

on a 40 x 40 mm temperature-controlled stage, (here forth termed the cold-stage) and control system (model LTS120, Linkham Scientific, Tadworth, United Kingdom). The cold-stage works by means of the Peltier effect, is fitted with internal temperature sensor and control system is capable to provide cooling down to -40°C (±0.1K). The device can be programmed to apply cooling or heating rates from $0.1 – 10$ K min$^{-1}$, including isothermal temperature holds for extended periods of time.

The freezing array used to hold the sample is a layered construction (Fig. 2, upper left panel I). It consists of 1: siliconized hydrophobic glass slides (HR3-217, Hampton research, Aliso Viejo, US) on which the sample drops are dispersed. As the slides are hydrophobic, the sample drops do not float out on the surface, but maintains a roughly spherical form. The slides were flushed with ultra-pure water before use, and discarded after each drop population. A silicon grid (2) was used to keep the drops separated on the glass slides, and sealed each sample drop in an individual cell between the slides and a polycarbonate

lid (3), minimizing interaction between the drops (i.e. by Wegener-Bergeron-Findeisen type transfer of vapour or seeding of neighbouring drops by ice-splintering, surface growth or frost halos). The drops were spread out in an approximate circle on the cold-stage to avoid the corners, where the temperature may be less precise during temperature ramps. The grid was laser cut from medical grade silicone. The assembly was centred and held on the stage by a polycarbonate holder/guide. The assembly and the stage were encased in a small environmental chamber (part of the LTS120 Linkham system) and the sample

was observed through a quartz glass window (Figure 2, panel I, 4). Figure 2 (panel II) also shows the sample mounted in the assembly on the cold-stage.

Figure 2 (panel III) shows a schematic overview of the full LUCS setup. The cold-stage with the mounted sample (A) is placed in a laminar airflow cabinet (B) to avoid any airborne contamination of the sample during preparation and handling. A camera

system (C) consisting of a digital SLR (Canon Eos 6D mk II, Canon, JP), fitted with a 150 mm macro lens (IRIX 150mm f:2.8 Macro 1:1, Irix, ROK) and a continuous circular light source (R300, FandV, Helmond, NL) is fixed over the viewing window and controlled by a computer (D). The camera captures images of the sample during the experiments. A 15 L cryogenic water circulator (E) (model DC-3015, Drawell International Technologies Limited, Shanghai, China) is connected to the temperature-controlled stage and provides a steady flow of 8.5°C cooling water for the cold-stage, acting as a heat sink. The

relatively large water circulator is an important element in the setup for maintaining the thermal performance and stability of the cold-stage over extended periods of time, such as during long isothermal experiments. The environmental chamber was continuously purged with a low flow of dry, clean nitrogen gas (F), and a steady flow of dry filtered air (G) was directed over the viewing window to avoid any problems with fogging that may obscure the imaging.

The camera system captured images of the cold-stage in intervals as different cooling programs were applied to the sample (as detailed below in sections 2.4.1 and 2.4.2). The ice spectra for the samples were inferred from the generated images by semi-supervised image analysis. An example of an image generated by LUCS, which was used as the input for automated analysis

of imagery, is shown in Figure 3. This displays unfrozen and frozen sample drops. As seen in Figure 3, the reflection of the circular light source is clearly visible in the liquid phase droplets, but rapidly disappears at the onset of freezing. This transformation was used to determine the freezing temperature and time for all sample drops from the recorded images.

## 2.4 Experimental design

The ice nucleation activity of the samples was measured on five drop populations from each sample consisting of 100 drops, each with a volume of 1 μL. There were six samples of ambient environmental aerosol material in total (Sec. 2.1), collected and classified as noted above (Sec. 2.2). For each of these drop populations at least three 2-hour isothermal experiments and one longer (11-16 hours) isothermal experiment were performed. The number of 2-hour isothermal experiments on each drop population was dictated by practical reasons, and the longer isothermal experiments were performed overnight. Measurements with a constant cooling rate of 2K min$^{-1}$ were performed before and after the isothermal experiments, and also between some of the isothermal experiments to assure that the freezing spectra remained unchanged during the experimental time for each drop population. The cooling programs used are defined as follows.

### 2.4.1 Experiments with constant cooling rate

The cooling program used for the experiments at constant rate of cooling is illustrated in Figure 4 (left panel), (the 'constant cooling-only experiment'). The sample was dispersed on the cold-stage and initially cooled with a fast cooling rate (8K min$^{-1}$) from room temperature to -5°C. The sample was then held at -5°C for one minute to assure thermal stability before a constant cooling rate of 2K min$^{-1}$ was applied to the sample until it was fully frozen.

### 2.4.2 Isothermal experiments

The cooling program used for the isothermal experiments is illustrated in Figure 4 (right panel). The sample was initially cooled with a fast cooling rate (8 K min$^{-1}$) from room temperature to -5 °C. The sample was then held at -5°C for one minute to assure thermal stability before a constant cooling rate of 2 K min$^{-1}$ was applied to the sample until 1K warmer than the target isothermal temperature, where the cooling rate was decreased to 1K min$^{-1}$ to avoid "under-shooting" the target temperature. When the target temperature was reached, the sample was held at this temperature for a determined period of time, ranging from 2 hours to over 10 hours. When the isothermal phase had elapsed, a constant cooling rate of 2K min$^{-1}$ was then applied to the sample until it was fully frozen.

The target temperature was chosen based on the initial constant cooling rate experiments for each sample to correspond to a temperature where about 20-30% of the sample was frozen at the onset of the isothermal phase. This resulted in an isothermal temperature of -16°C for the 'continental polluted' sample and -14°C for all other samples in this study.


### 2.4.3 Quality control

The cold-stage temperature was measured during different operation regimes with external thermocouples on different occasions, and a very good agreement with the cold-stage temperature sensor was observed within the errors. The experiments with constant cooling rates performed before, between and after the isothermal experiments (Sec. 2.4.2) were primarily

included to ensure that the freezing spectra of the samples remained consistent during the experimental time.  This was effectively a check on both the consistency of performance of the LUCS apparatus and the stability of samples with respect to repeated measurements. There was a total of about 24 hours of exposure to repeated cycles of heating and cooling for each drop population.

There were multiple drop populations for each sample and in total 3000 different droplets have been studied in detail, which allowed for a statistical analysis of freezing temperatures. We found that the 50% of studied droplets located farthest away from the cold-stage centre on average tended to freeze at nominal temperatures 0.20 K lower than the central 50% of droplets for the 2.0 K min$^{-1}$ constant cooling ramps and for a temperature range around -16°C. Hence, the temperature was likely biased high by about 0.20 K during those cooling ramps for the droplets located closer to the cold-stage edge. The ambient

concentrations of INPs are presented further below, and the temperature bias has been corrected for in those results. It was not possible to estimate a potential similar temperature bias for the isothermal experiments, but the bias was likely smaller for these more constant cold-stage operation conditions. Hence, we cannot rule out that droplets closer to the cold-stage edges were exposed to temperatures of 0.1-0.2 K higher than the reported temperatures during isothermal experiments, and we note that such an offset is comparable to the instrumental accuracy.


The freezing spectra for ultra-pure water and a constant cooling rate of 2.0 K min$^{-1}$ are shown in Figure 5. In general, the first droplet would tend to freeze for a temperature below -20°C while 50% of the droplets would freeze for temperatures below -30°C. However, on rare occasions, some freezing incidents of ultra-pure water droplets for temperatures above -20°C have been observed, which was the case for one of the samples included in Fig. 5. We associate that with poor quality of single

hydrophobic glass slides as closer visual inspection often would indicate. Hence, a very thorough cleaning and visual inspection of the hydrophobic glass slides were carried out for experiments with the ambient samples in this study.

It was seen that the cooling-only experiments remained consistent for all samples during the experimental time, although statistical variations were observed between individual drop populations. In addition, measurements were also carried out with

field blank samples to assure that no significant freezing could be observed to arise from either the measurement apparatus, the polycarbonate filters or the handling procedures at temperatures overlapping with the samples (Fig. 5). Specifically, isothermal experiments were also done with both ultra-clean water and field blank samples, and no freezing events were observed during 2-hour isothermal experiments at the target temperatures used in this study (-14 and -16 °C).


## 3 Results

### 3.1 Validation of repeatability and sample stability

#### 3.1.1 Freezing spectra

Figure 6 shows the average INP concentrations for the different samples inferred from five different droplet populations per sample and the first cooling ramp per droplet population. The INP concentrations were inferred as described by Vali (1971). The random error ranges corresponding to ±one standard deviation are also depicted. The presented INP spectra have not been corrected for the background, as the background was negligible relative to INP concentrations in the ambient samples. The Fletcher parameterization (Fletcher, 1962) is included for comparison. The relative differences between these samples depend on the temperature range of relevance. For the temperature range from -18 to -14 °C, the lowest INP concentrations were found in the rural continental sample (0.02 L$^{-1}$ at -15 °C), with higher concentrations by 30% to 100% in the continental polluted sample (0.03 L$^{-1}$ at -15 °C). The INP concentrations for the other samples were significantly higher than the rural continental sample, for example with intermediate levels in the marine sample (0.1 L$^{-1}$ at -15 °C). The INP concentrations in the continental pristine, combustion dominated and mineral dust influenced samples appeared similar within the random errors within this temperature range (about 0.1 to 0.2 L$^{-1}$ at -15 °C).


For temperatures above -12°C, the random errors are rather pronounced due to the low numbers of droplets frozen. The INP concentrations for the mineral dust influenced sample show pronounced variability between the different droplet populations in the high temperature range, which results in the random error being the same order of magnitude as the inferred concentrations. Hence, the inferred INP concentrations cannot be considered statistically different from what is inferred for any of the other samples in the high temperature range. However, it appeared likely that the highest concentrations were found in the marine sample, while the continental pristine sample contained the lowest INP concentrations in the high temperature range. A relatively low concentration of 0.2 PBAPs per liter of air was observed for the continental pristine sample (Table 1), which is a likely explanation for the relatively low INP concentrations in the high temperature range.

Minima in biological particle concentrations and INP concentrations at a temperature of -16°C have been observed in Finland during the winter season (Schneider *et al.*, 2021). Mason *et al.* (2015) reported a strong correlation between PBAP and INP concentrations for a temperature of -15°C at a coastal site, with the PBAP concentrations being about 2 to 3 orders of magnitude higher than the INP concentrations. In the current study, we find the fluorescent particle concentration (9.2 $L^{-1}$; e.g., from PBAPs) to be almost 2 orders of magnitude higher than the INP concentration at -15°C (0.14 $L^{-1}$) for a Saharan mineral dust

influenced sample—where contributions from other INP types are likely (e.g. mineral dust).   So, we would only expect a minor fraction of the PBAPs to be ice-active at temperatures around -15°C. For the pristine continental sample, we observed a fluorescent concentration of 0.2 $L^{-1}$, while the INP concentration for a temperature of -15°C was about 0.02 $L^{-1}$. We consider it unlikely that as many as 10% of the PBAPs were ice active at that temperature—so it is likely that other INP types contributed significantly to the INP concentration in that sample, particularly when it comes to the   INPs active below -14°C. Thus we

would expect a large fraction of the time-dependent freezing events to be facilitated by other INP types than PBAPs in that sample, but it is not quite clear to which extent it may be linked to the low concentrations of combustion or mineral-dust associated aerosol components detected in that sample (Table 1). It is not possible to rule out significant fractions of ice-active PBAPs at temperatures below -14°C in the other samples either due to higher PBAP concentrations or lack of PBAP data, especially with the rural continental sample that occurred in a warmer season.  Low levels of combustion and dust particles in

the rural continental sample suggest that PBAPs most likely made up a relatively larger fraction of INPs in that sample.  Indeed, near the detection threshold of -8 °C, the rural continental sample shows the weakest temperature gradient of INP concentration among all samples (Fig. 6), suggesting undetected enhanced activity at even warmer temperatures; PBAP-IN are unique among INP types generally in their ice-activity above -10 °C (Phillips *et al.* 2013; Morris *et al.* 2014), (Sec. 1).

It is noteworthy that the marine aerosol sampled (0.1 $L^{-1}$ at -15 °C), and the combustion-dominated and mineral dust influenced samples (~0.15 $L^{-1}$ at -15 °C), all had comparable concentrations of INPs within a wide temperature range ($10^{-2}$ to $10^{0}$ $L^{-1}$ from -20 to -10°C). Those samples were very different in terms of aerosol properties (Table 1) and relative levels of main INP candidates which roughly can be divided into being of biogenic origin or associated with either mineral dust or combustion. The mineral dust influenced sample most likely contained the highest mass concentration of mineral dust due to the elevated

$PM_{10}$-$PM_1$ level and the dust related components detected in a similar sample. However, there are also indications of significant combustion emissions in the mineral dust influenced sample. It is not clear to which extent the combustion dominated sample had moderate levels of mineral dust and/or fly ash components present in the $PM_{10}$-$PM_1$ fraction, but the level was significantly lower than observed for the mineral dust influenced sample. Hence, the relative INP importance of the dust component was likely higher in the mineral dust influenced sample. The components indicative of combustion emissions and dust were

approximately an order of magnitude lower in the marine sample relative to the combustion-dominated and mineral dust influenced samples (see Table 1). Hence, it appeared likely that other INP candidates played a significant role for the marine sample in order for the INP concentrations to be of comparable magnitude. Marine biogenic components with small particle

sizes (diameters < ~0.2 μm) have been reported to be immersion freezing ice-active (e.g. Wilson *et al*., 2015), which we consider likely candidates to contribute significantly to the INP population in the marine sample.


There is some commonality between our measurements of INPs and published studies in the literature. The concentrations of INPs we report for the marine sample (Fig. 6) are almost identical to the average concentrations reported for a number of Pacific Ocean samples (Mason *et al*., 2015) and very similar to concentrations reported by Si *et al*. (2018) for the Northern North Atlantic. In Arctic marine samples, lower INP concentrations have been reported (Irish *et al*., 2019). Overall, lower and

higher INP concentrations than what we report have been observed in marine environments.

The continental polluted sample contained elevated levels of combustion aerosol and most likely also of dust and/or fly ash present in the $PM_{10}$-$PM_1$ fraction. However, the INP concentrations are significantly lower than observed for the combustion dominated and mineral dust influenced samples within a wide range of temperatures. That could potentially be associated with

different properties of the combustion/dust components of relevance as INPs.

The rural sample was characterized by very low levels of combustion and dust related components and a moderate level of sea salt. Hence, it may be that the relatively low INP concentrations associated with that sample could be related to a biogenic marine component – most likely present at lower concentrations than in the marine sample. Furthermore, it is possible that

other biogenic components such as PBAPs could play a role as INPs in that sample. The rural continental aerosol sample (0.02 $L^{-1}$ at -15 °C) contained INP concentrations very similar to the 'continental polluted' aerosol (0.03 $L^{-1}$ at -15 °C). The 'continental pristine' aerosol (0.1 $L^{-1}$ at -15 °C) is characterised by a different slope, with relatively low INP concentration in the high temperature range, and relatively high concentrations in the low temperature range.

A statistical analysis of the active INP concentrations measured at -15 °C, which is near the temperature of isothermal experiments below, shows that none of the samples are perfectly unique in their freezing behaviour compared to other samples (Appendix A). Each sample resembles at least one other sample. The continental pristine and continental polluted samples are the least unique because they each resemble the greatest number of other samples, including each other, by this metric. The other four samples are equally unique in terms of the numbers of other samples they resemble in their ice nucleation at that

temperature.

Regarding aerosol composition of samples, the measured levels of PBAPs, combustion and dust components span more than one order of magnitude between the studied samples (Table 1). Hence, it seems highly likely that the relative importance of various INP types varied significantly between these samples. It was not possible to quantify the relative importance of various

INP types with the methods applied. However, based on the information presented in Table 1, it is evident that the studied

samples represent highly different physico-chemical aerosol properties of relevance at least to Northern Europe during different seasons.

So as to assess the robustness of the experiments with respect to repeatability and sample stability over the total experimental time (up to > 24 hours), identical experiments with a constant cooling rate (Sec. 2.4.1) and the same aerosol sample were done at various intervals (Sec. 2.4.3). Such cooling-only experiments were always performed before and after the isothermal experiments for all drop populations from each sample, and also between some of the isothermal experiments. Figure 7 shows that for most samples the difference in freezing temperature (for any given drop population) during repeated constant cooling rate experiments (over 24 hrs) is up to about 0.5K between freezing events at different times for any given value of frozen fraction. However, for mineral dust influenced and combustion dominated samples there was somewhat more variation (up to 1K), indicating that a small minority of the drops may have undergone slight alterations in the nucleating ability of their immersed INPs during the freezing cycles. Most of the variability for any given sample was related to statistical variations in the number of active INPs among different drop populations, where the largest variation between drop populations was also observed for the mineral dust influenced and combustion dominated samples.

Finally, it might be argued that any observed time-dependence might arise for non-stochastic systematic reasons over many hours. For instance, contact of immersed solid INPs with a shrinking drop surface inherently favours freezing by 'inside-out contact-freezing' (Durant and Shaw 2005). However, that did not actually occur. Freezing was measured by video analysis of images of the drops. Inspection of the imagery confirmed that although some of the drops may become smaller by evaporation during the full experimental time of many hours, any shrinking was only slight. Moreover, the constant cooling rate experiments were run regularly between the isothermal experiments, including a run after the last isothermal experiment. This revealed that each drop population behaved consistently from the beginning of the experiment to the last run (Figure 7).

In summary, the instrumental precision, repeatability of our experiments on each drop population and the absence of any measurement changing bias are confirmed by these experiments. Variations of freezing temperature at a given frozen fraction are of the order of < 1 K for repeated cooling experiments with the same drop population, which is much less than the signal from time dependence reported below.

### 3.1.2 Repeatability of freezing for individual drops

All observed drops were indexed and tracked through all freezing cycles, during both the isothermal experiments (Sec. 2.4.2) and the experiments with constant cooling rates (Sec. 2.4.1). It was found that most drops froze in a relatively narrow temperature range during repeated experiments (as also seen in Figure 7). The average temperature of at least four cooling-only experiments (Sec. 2.4.1) on each drop allowed determination of the average freezing temperature, standard deviation and

freezing range for each individual drop. This temperature will henceforth be referred to as the '*normal freezing temperature*' for the drop. The median of the standard deviations for four freezing cycles on each drop was about 0.25-0.34 K for this normal freezing temperature and the median of the freezing ranges were 0.53-0.93 K (Table 2) for the various samples, which is consistent with previous reported observations (Vali 2008).

In Figure 8 the normal freezing temperatures (and range for the observed freezing temperatures) are displayed for those drops that were also observed to freeze during the isothermal experiments. As seen in Figure 8, a majority of these drops that froze during the isothermal experiments had a normal freezing temperature lower than the isothermal temperature for the experiments (marked in the figure by the dotted cyan line). The difference for most drops displayed is about 1-2 K. But a tiny fraction of these drops had frozen at temperatures up to 5 K warmer than for the constant cooling rate cycles.

The practically sigmoidal-like distribution of normal freezing temperatures (Fig. 8) arises because the average probability of any drop freezing per unit time during any isothermal experiment must, when comparing all such drops, decrease with decreasing normal freezing temperature below the isothermal temperature among them. For a given drop, this probability is governed by the immersed surface area of INP material and its composition. These underlying quantities also follow a statistical distribution among drops. Drops with the most depression of the normal freezing temperature below the isothermal temperature would be expected to contain less, or less efficient, INP material than most that freeze, causing these rare drops to freeze only on unusually long time-scales in the isothermal experiment.

In summary, the effect from time-dependence of freezing over 2 hours is to raise the freezing temperature by about 1-2 K in most cases relative to the 'normal' value in cooling-only experiments. However, for a small minority (< 10%) of drops, this temperature shift exceeds about 5 K for the mineral dust influenced and combustion-dominated samples.

## 3.2 Isothermal experiments

### 3.2.1 Isothermal time series and relaxation time

All examined samples showed the same general pattern during the isothermal phase (Sec. 2.4.2). Freezing events are more numerous during the first minutes of the experiment and then their frequency decreases as time progresses (Fig. 9, blue/mauve dots). There was a significant variability between individual experiments and drop populations from the same sample, but this should be expected both from the natural diversity of possible INPs in environmental samples and the stochastic nature of ice activation. The variability among the isothermal experiments is larger than for the constant cooling rate experiments as the dependence on temperature is much larger than that on time. This makes the constant cooling experiments seem more predictable and repeatable and less stochastic than the isothermal experiments.

Additionally, the data in the time series of frozen fraction were binned in logarithmically spaced time intervals and the average frozen fraction was plotted for each bin (Fig. 9, yellow points). Figure 10 shows a relative enhancement of the number of frozen drops during the entire 10-hour period of the isothermal phase, namely $\chi = f_{ice}(t^* = 10 \text{ hrs}) / f_{ice,0}$ (from final and initial averages of frozen fraction for each sample). Here $f_{ice}(t^*)$ is the frozen fraction (total number of drops frozen since the start of cooling at 0 °C divided by the original number of drops at 0 °C) observed at any instant, averaged over all experiments with any given sample. $t^*$ is the time since the start of the isothermal phase. It may be viewed as a dimensionless number of drops. Hypothetically $f_{ice} = 1$ would correspond to complete freezing of all drops observed in the array and $f_{ice} = 0$ to no freezing (e.g. before any cooling at 0 °C). Also, $f_{ice,0}$ is the average initial ice fraction for the sample at the beginning of the first measurements of the isothermal phase ($t^* = 0$). Note that $f_{ice}(t^* = 0) = f_{ice,0} > 0$ because some drops freeze before the isothermal temperature is reached (at $t^* < 0$). The corresponding enhancement over the first 2 hours is also shown in Table 3. The two samples with the most enhancement are the mineral dust influenced and rural continental samples, whereas the two with the least enhancement are continental pristine and combustion dominated samples (Fig. 10).

Figure 11 shows the fractional rate of increase of the number of frozen drops, $f_{ice}(t^*)^{-1} df_{ice}(t^*)/dt^*$, from the isothermal experiment. It is evaluated numerically by a finite difference approximation for the derivative, applied to the smoothed (running mean) time series of frozen fraction for each sample, averaged over all experiments (yellow points in Fig. 9). Also shown is the fractional rate of freezing of unfrozen drops (the 'real freezing rate'), $-(1/f_u) df_u/dt^* = (f_{ice}(t^* = 10 \text{ hrs}) - f_{ice}(t^*))^{-1} df_{ice}(t^*)/dt^*$, considering only those drops that eventually freeze (during 10 hrs) at the isothermal temperature. Here, $f_u(t^*) = f_{ice}(t^* = 10 \text{ hrs}) - f_{ice}(t^*)$ is the number of unfrozen drops that will eventually freeze during the isothermal period divided by the total number of liquid drops at 0 °C. Both fractional rates of change of frozen and unfrozen drops decrease with time (linearly on a log-log plot), following a power-law dependency, from the start of the isothermal phase. This decline is expected from variability of stochastic behaviour among INPs. For example, Wright and Petters (2013, their Figure 7) observed a steady decay of the rate of change of the number of frozen drops with time and fitted their observations with a model based on a modified classical nucleation theory involving a statistical distribution of active sites of a wide range of efficiencies and multiple components (see also Knopf *et al.* 2020). Another interesting feature of both fractional rates of change (Fig. 11) is the similarity among samples at any given time, all sharing the same order of magnitude mostly, despite contrasting chemical composition of the aerosol samples (Sec. 2.2.2) and of likely types of dominant INPs. That similarity is explicable in terms of diverse types of INPs generally sharing similar freezing behaviors governed by the kinetics of the active sites of immersed INPs 'reacting' with incident liquid molecules of suitable energy, by analogy with the chemical kinetic theory for activity of interstitial INPs by DeMott *et al.* (1983).

The reciprocal of the real freezing rate is the natural time scale of the freezing at any instant. If hypothetically all drops contained only one INP each and all their INPs were somehow of identical size, composition and nucleating efficiency, then

the unmodified stochastic model (e.g., Bigg 1953ab; Sec. 1) would predict constancy of the fractional rate of change of the unfrozen drops (the real freezing rate) with each drop having the same probability of freezing per unit time (Fig. 11b). Fig. 11b rules out that hypothesis. The steady decay of the real freezing rate (Fig. 11b) is explicable in terms of each unactivated INP (among those that can eventually activate at the isothermal temperature) having a unique temperature-dependent probability of freezing per unit time that has a statistical distribution among drops. As the isothermal experiment progresses,

first unactivated INPs with higher efficiency at that temperature will nucleate faster on shorter time scales with a higher probability per unit time. Later on progressively less efficient INPs that are slower to freeze remain unactivated and may freeze at long times. Such INPs that are less efficient at nucleating ice could have either less abundance of solid material in each drop, with less chance of an active site on its surface, or a composition that is inherently less efficient.

Similarly, the limited literature of observations show that the unfrozen fraction is often seen to decay steadily with time at constant temperature (Bigg 1953ab, Vali 1994, PK97; Knopf *et al.* 2020). Recently, Knopf *et al.* (2020, their Figure 2a) show observations of the logarithm of the unfrozen fraction of drops containing illite plotted against time, with this logarithm decreasing almost linearly with time until 2 hrs, except with a gradient (rate of decrease of this logarithm) becoming progressively less steep. This is consistent with a quasi-exponential dependency with a relaxation time, $\tau(t)$, (reciprocal of

that gradient) that itself dilates with time ($\propto e^{-t/\tau(t)}$ ).

Thus, regarding our measurements in the isothermal experiments, we hypothesize that the unfrozen fraction of all drops that eventually freeze at the isothermal temperature may be expressed as:

$$f_u \equiv f_{ice,0} + \Delta f_{ice,\infty} - f_{ice}(t^*) = \Delta f_{ice,\infty} \; e^{-\frac{t^*}{\tau}} \quad (1)$$

Here the real (fractional) freezing rate is $(1/f_u) \, df_u/dt^* = -1/\tau(t^*)$, which is just the probability of any drop freezing per unit time among those that eventually freeze at the isothermal temperature. Here $\Delta f_{ice,\infty}$ is the eventual increase of the frozen fraction during the entire period of the isothermal phase. Here $\tau$ is a relaxation time and is the natural time-scale of the freezing.

From inspection of the literature noted above, the relaxation time-scale is allowed to evolve somehow over time ($\tau = \tau(t^*)$).

Consequently, from our isothermal measurements (Fig. 11), the time dependency effects were inferred by re-arranging Eq (1) to yield this empirical isothermal formulation, which was then fitted to the measurements from the isothermal phase:

$$f_{ice}(t^*) = f_{ice,0} + \Delta f_{ice,\infty} \left(1 - e^{-\frac{t^*}{\tau}}\right) \qquad (2)$$

As an empirical isothermal formulation, Eq (2) is not intended as a general model of ice nucleation *per se* and applies only to INPs exposed isothermally to freezing, hence the absence of any temperature-dependence.

Numerically, $\tau$ can be determined from our empirical data by re-arranging Eq (2):

$$\tau(t^*) = -\frac{t^*}{\ln\left(1 - \frac{f_{ice}(t^*) - f_{ice,0}}{\Delta f_{ice,\infty}}\right)} \qquad (3)$$

The fitting of Eq (2) to the measurements was done as follows. First, $f_{ice,0}$ and $\Delta f_{ice,\infty}$ were estimated from the initial and final
averages of frozen fraction during the isothermal phase (Fig. 9, initial and final yellow points). During the isothermal period, from each average of the measured frozen fraction ($f_{ice}(t^*)$; yellow dots in Fig. 9) the relaxation time was inferred using Eq (3), as shown in Figure 12 (blue dots). (Note that an alternative to Eq (3) could have involved $1/\tau(t^*) = (f_{ice,0} + \Delta f_{ice,\infty} - f_{ice}(t^*))^{-1} df_{ice}/dt^*$, with the time-scale being the reciprocal of the fractional rate of change of the number of unfrozen drops among those that can eventually freeze in the isothermal phase.)


These inferred values of $\tau$ were seen to conform to a power law, as shown in Figure 12 (red lines):

$$\hat{\tau}(t^*) = C_i t^{*\alpha} \qquad (4)$$

Note that both in the data and in the fits, as $t^* \to 0$ always $\hat{\tau}$ decreases (Fig. 12). Here $C_i$ is a constant for the $i$-th sample. Also, $\hat{\tau}$ increases monotonically with time throughout each isothermal period. This dilation of the relaxation time-scale with the age of exposure to constant conditions of temperature and humidity is explicable in terms of a statistical distribution of active sites among all the INPs. The most active sites nucleate ice on shorter time-scales and are then 'lost', so the less active sites remain and they activate on longer time-scales, as time progresses, as noted above.


The values for the fit parameters of Eq (4) are given in Table 4. With these, $f_{ice}(t^*)$ was reconstructed by applying the empirically fitted relaxation time, $\hat{\tau}(t^*)$ from Eq (4) for $\tau$ in Eq (2). Figure 9 (red lines) confirms that this empirical isothermal formulation agrees with the experimental data used in its design. All maths symbols are defined in Appendix B.

In summary, the observed isothermal dependence on time of freezing conforms with a simple law of a frozen fraction increasing almost exponentially initially and then approaching an asymptotic value after an extended time of a few hours. The relaxation time increasingly dilates as time progresses throughout the isothermal phase, as expected from the most efficient INPs being steadily depleted by activation and leaving unactivated the less efficient INPs with longer characteristic times for

activation.  The real freezing rate, among unfrozen drops at any instant that will eventually freeze, is just the reciprocal of this relaxation time and tends to decline with time.

### 3.2.2 Time dependent temperature shift for use in empirical parametrization of INP activity

Several parametrizations of heterogeneous ice nucleation are based on observations involving a short time of exposure to constant conditions of humidity and temperature, (DeMott *et al*. 2015, Phillips *et al*. 2008, 2013). This time is about 10 s for a typical continuous flow diffusion chamber (CFDC).  In order to modify such schemes so as to represent the time dependency of INP activity, we propose a temperature shift approach (see also Wright and Petters 2013). The modified active INP concentration, after exposure on longer time scales, may be assumed to  equal  the value from such a scheme for a shifted value of the temperature input, with the shift evolving over time ($\Delta \tilde{T}_X = \Delta \tilde{T}_X(t^*) \leq 0$ ):

$$\tilde{n}_{IN,X,*}(T, S_i, \Omega_X, t^*) = n_{IN,X*}\left(T + \Delta \tilde{T}_X(t^*, \dots), S_i, \Omega_X\right) \qquad (5)$$

Such an approach is supported by findings by Herbert *et al*. (2014) as noted above (Sec. 1).  Here, $\tilde{n}_{IN,X,*}$ is the time-dependent active INP concentration (number per unit mass of air) as a function also of the ambient temperature $T$, the saturation of vapor with regard to ice, $S_i$, and the available surface area, $\Omega_X$, of aerosols of the $X$-th INP species. Also $n_{IN,X*}$ is the corresponding concentration from the reference activity spectrum of the original INP scheme (e.g., Phillips *et al*. 2008, 2013) without any time-dependence.  Eq (5) here is based on our empirical parameterization (EP) of heterogeneous ice nucleation by multiple species of aerosol (Phillips *et al*. 2008, 2013).  Yet the same method is generally applicable to any other INP scheme that neglects time-dependency.

Figure 13 (blue dots) shows the temperature shift inferred for every measurement of frozen fraction (Fig. 9, blue/mauve dots) for all samples during the isothermal phase. This was done by averaging the frozen fraction over the first 10 s and matching this with the prediction of the scheme by adjusting the constant of a proportionality between the frozen fraction and $n_{IN,X*}$ for $t^* = 0$.  So, by definition, $\Delta \tilde{T}_X = 0$ initially. Then at all times subsequently $\Delta \tilde{T}_X$ is numerically solved to satisfy Eq (5).

The shift values for the ambient temperature input to the INP scheme (EP) were seen to conform to a power law, as shown in Figure 13 (red dotted line):

$$\Delta \tilde{T}_X(t^*) = -A_i t^{*\beta} \leq 0 \qquad (6)$$

All samples show a temperature shift of about 3-5 K and 5-8 K over the initial 2 and 10 hours respectively. Here $t^*$ is the time since the start of isothermal conditions and $A_i$ is another constant for the $i$-th sample, to which species $X$ corresponds somehow as the INP type assumed to dominate the observed freezing behaviour of this sample (Sec. 5). Such an ambient temperature shift (downward) may be viewed as arising from corresponding opposite shifts (upward) of comparable absolute magnitude in characteristic freezing temperatures of all INPs. These actual shifts in freezing temperature follow a statistical distribution

in reality for any given INP species, as noted above (Fig. 8).

### 3.3 Constant cooling rates with and without an isothermal phase

Figure 14 (drop populations: cyan lines; averaged for all drops: blue line) shows the frozen fraction as a function of temperature during cooling at a constant rate (2.0 K min$^{-1}$) from -5°C until all samples are fully frozen, in namely the cooling-only

experiments. As noted above for the same data displayed in Figure 7 (various colours), the frozen fraction goes from almost zero to unity between about -10 and -20 °C. The temperatures for the frozen fraction of 0.5 are shown in Table 5 for all six samples.

So as to reveal the influence from time-dependence, other experiments were performed interspersing constant cooling (2.0 K

min$^{-1}$) with an isothermal phase (2 hours) as shown in Figure 14 (drop populations: magenta lines; averaged for all drops: red line), ('hybrid cooling-isothermal' experiments). Before the isothermal temperature was reached, for each sample the frozen fraction evolved identically for these hybrid cooling-isothermal experiments (red line) as for the ordinary cooling-only experiments with a constant cooling rate (blue line). This indicates that the isothermal phase is reached with no "under-shooting" of the target temperature, and that the lower cooling rate used for the last 1K does not influence the result.


Comparison of these hybrid cooling-isothermal experiments (red lines) with the corresponding cooling-only experiments (blue lines), performed subsequently after the isothermal temperature is reached, is a measure of the extent of time dependence for each sample (Fig. 14). Hypothetically, if there were no time dependence (with each INP functioning as a perfect 'switch' with activation exactly at its fixed characteristic freezing temperature), then the red line would simply follow the blue line, and the

blue line would be relatively unchanged. The deviation of the red line from the blue line is a measure of time dependence arising from an extra 2 hours of exposure to constant conditions of temperature.

The eventual impact from time-dependence of freezing after 2 hours is evinced by the maximum difference in freezing temperature, between the red and blue lines, being about 1-2 K for all six samples (Table 5). The pattern of the degree of time-

dependence among the samples is qualitatively consistent with that seen in Fig. 10, which shows the total fractional increase in INP activity after 2 hrs of exposure to constant conditions. Thus, Figures 10 and 14 are consistent about which samples are the most (e.g. mineral dust influenced) or least (e.g. continental pristine) time dependent.

Even after the end of the isothermal phase for some experiments, a deviation persists between the red and blue curves in Fig.

14. Some of the INPs causing this deviation, which normally would have activated at temperatures a few K colder than the isothermal temperature in the cooling-only experiment, actually activated at this temperature during the prolonged isothermal phase of the hybrid experiment because it was so prolonged (2 hrs). Generally, as the temperature cools after the end of the isothermal phase, the hybrid experiment shows a frozen fraction that becomes increasingly similar to the cooling only experiment, as expected from the strong dependency on temperature of INP activity.


In summary, the observed time-dependence of freezing involves a steady increase with time of the freezing temperature, by about 1-2 K after 2 hours for most drops. This further justifies the approach taken above for incorporating the effect of time by means of a time-evolving temperature shift for schemes of heterogeneous ice nucleation in cloud models (Sections 3.2.2, 5).


## 4 Discussion

The present study attempts to fill a gap in knowledge about the role of time in real-world atmospheric ice processes, and how its influence should best be represented. Several previous studies have aimed to provide both a theoretical understanding and empirical data (Vali 1994; Welti *et al*. 2012; Wright and Petters 2013; Budke and Koop 2015; Knopf *et al.* 2020) so as to

represent time dependence more accurately in ice nucleation. However, much of the previous published work studied the effect on idealized systems, and on relatively short time scales. Such studies are invaluable for understanding the basic physics of time dependence in ice nucleation, but may be challenging to apply for practical use in atmospheric modelling.

Pioneering aspects of the present study include the fact that the effects of time dependence on ambient environmental samples

are measured. This is done for far longer time scales (many hours) than observed hitherto in other studies (many mins) and provides more realistic data which can be applied directly to modify INP schemes and cloud models. It is, to the best of our knowledge, the first study to date investigating time dependence on real-world ambient aerosol samples, although real precipitation samples were observed previously (e.g., Wright, 2014).

Generally, the results from the cooling-only experiments (e.g., Figure 7) show that the instrumental setup provides consistent measurements on each individual drop population for all samples, agreeing well before and during, as well as after, the repeated isothermal experiments (Sec. 3.1.1). Together they comprise a total experimental time of up to 24 hours on each drop population. This shows that the measurements are robust, and that the samples do not change significantly during the instrumental time.

The data derived from the isothermal experiments (Figures 9 and 13) show much variability, which may be expected both because of the stochastic nature of time dependence and from statistical variations in the composition of INPs among different drop populations. However, when all frozen fraction data were averaged for a given sample and then fitted by Eq (2), the assumed fit was found to conform to the data (e.g., compare red curve and yellow points in Figure 9). Thus the approach with several repeated isothermal measurements on several drop populations from each sample is likely to give a realistic, albeit approximate, estimate of the effect of time dependence for the different samples. In short, the measurement datasets are sufficient for adequate statistics to describe each sample.

Generally, in such lab measurements of freezing, an effect related to the composition of INPs, which may be a cause of uncertainty, is the risk of 'saturation' of active INPs in the sample solution. As all samples in this study are environmental samples, there is a risk that multiple INPs may be present in each sample drop, and the first INP to activate in any drop will be the one represented in the results. This might introduce a masking effect from the more efficient INPs activating before information can be obtained about any less active INPs in the same drop. For instance, if a drop contains a PBAP INP that nucleates ice at relatively high temperatures, this particle may obscure the presence of other INP in the same drop that would otherwise become active to cause freezing at lower temperatures or after longer times.

However, at the isothermal temperatures of the present study (-14 °C and -16 °C), the cumulative active INP concentrations for the samples were estimated from their freezing spectra (Figure 6) and are of the order of 0.1 to 1 active INPs L$^{-1}$ of air. This would imply that the average number of active INP in each 1 μL sample drop should be less than unity (order of 0.1 to 1 per drop). That in turn confirms the applicability of the drop-freezing measurements for estimation of the atmospheric ice nucleating ability of our samples at such temperatures.

Curiously, as seen in Figure 8 for the rural continental sample, it indeed seems as if the overall active INP concentration is low, and that there may be multiple INP types in the same sample. One INP type activates close to the isothermal temperature (-14 °C); another INP type is less prevalent and activates about 5 K lower than the isothermal temperature (near -19 °C). This is consistent with the observation that this sample, which has relatively few IN, may show a mode of INP activating at warmer temperatures (e.g., possibly PBAPs or mineral dust).

## 5 Implementation of Results for Time-Dependence in Cloud Models

In nature, there are many types of INPs that can be classified as belonging to broader aerosol groups frequently referred to as atmospherically relevant (Kanji *et al*. 2017), as used in some INP schemes (DeMott *et al.* 2015; Phillips *et al.* 2008, 2013). However, in the present project the samples investigated are from the ambient environment and must be assumed to contain a complex composition, where multiple INP species are abundant. Compared to opting for more well-defined artificial samples (e.g. Arizona test dust, Snowmax®) as done in some past studies (Welti *et al*. 2012; Budke *et al*. 2014), the approach of sampling aerosols from the real troposphere entails several challenges (sections 1 and 2.2). In particular, the identity of the INPs dominating the ice initiation in each of our samples is uncertain.

Nevertheless, the time dependence of the temperature shift that we observed (section 3.2) allows preliminary representation of time dependence of INP activity in atmospheric models. Comparison between cooling-only and hybrid cooling-isothermal experiments further illustrates how that the effects from time-dependence may be expressed in terms of a shift in freezing temperature that increases with time (Sec. 3.3).

Eq (6) can be applied in any cloud model to modify such INP schemes for inclusion of time dependence (Table 6), (Sec. 3.2.2). There are several obstacles to overcome regarding implementation of our observed temperature shift. First, there is the issue of which of our samples ($i = 1$ to 6) is most likely to represent each INP species ($X$) in any model. Our statistical analysis of which aerosol samples are most unique in their freezing behaviour would suggest use of either the combustion-dominated, rural continental, mineral dust influenced or marine influenced aerosol samples for major INP types (Appendix A). From the classification of our samples (section 2.2) from the Hyltemossa field station, some likely correspondences may be hypothesized. For example, the mineral dust influenced sample is likely to represent the ice nucleating behaviour of mineral dust INPs, which are known to be active at the isothermal temperature (about -15 ºC), (e.g., Phillips *et al*. 2013); possibly the combustion-dominated and rural continental samples may reflect carbonaceous INPs, perhaps for those that are non-biological and for PBAPs respectively (Table 1), (Sec. 3.1.1). Uncertainty in these correspondences between classified samples ($i$) and INP types ($X$) treated in models may not be a major problem since the temperature shift due to time-dependence varies by less than a factor of about 2 among our contrasting samples (Fig. 13).

Secondly, there is the problem about how to define the time of exposure to constant conditions represented by $\tilde{t}$ in Eq (6) when it is applied in a model. In a natural cloud there are vertical motions as well as long-lived regions of little ascent. For practical implementation in a cloud model, we make the simplification that the above lab results (Sec. 3.2; e.g., Eq (6)) apply to a simulated cloud, where the time since onset of isothermal conditions in a hypothetical lab experiment with the same aerosol population, namely $t^*$, is approximated by the time since the current parcel first entered the glaciating part of the cloud, $\tilde{t}$.

This 'age' of cold parcels ($\tilde{t}$) may be estimated by a passive tracer, $Q$, that decays exponentially with time while they are in cold clouds at subzero levels ($T < 0\,°C$) where the cloud-ice water content (IWC) $> 10^{-6}$ kg m$^{-3}$. The passive 'clock' tracer evolves as follows in the cloud model:

$$\frac{DQ}{Dt} = \begin{cases} \frac{-Q}{\tau_Q} & \forall\ T < 0\,°C \text{ and IWC} > 10^{-6} \text{ kg m}^{-3} \\ 0 \text{ kg}^{-1} & \text{otherwise} \end{cases} \quad (7)$$

Outside of this cold cloud region, $Q = Q_0 = 1$ kg$^{-1}$ is always prescribed everywhere during the entire simulation. Eq (7) is solved numerically by the cloud model during the simulation, predicting the evolution of $Q$ throughout the domain, including tendencies on the right-hand side for its grid-box average from sub-gridscale mixing. $Q$ is unchanged by all microphysical processes, as it is passive.

In an adiabatic parcel, Eq (7) has the analytical solution, $\tilde{Q} = Q_0 \exp(-\tilde{t}/\tau_Q)$, where again the time since entering the cold cloud region is $\tilde{t}$. Then assuming $Q \approx \tilde{Q}$ yields:

$$\tilde{t} \approx -\tau_Q \ln(Q/Q_0) \quad (8)$$

This cold-parcel 'age' is then used in Eq (6) to estimate the temperature shift, $\Delta \tilde{T}_X = -A_i \tilde{t}^{\beta}$ for species $X$, applying the $i$-to-$X$ correspondence assumed above. This shift in turn is added to the temperature input for the INP scheme ($\tilde{n}_{IN,X,*}(T, S_i, \Omega_X, \tilde{t}) = n_{IN,X*}(T + \Delta \tilde{T}_X(\tilde{t}), S_i, \Omega_X)$).

The INP scheme, such as the empirical parameterization (Phillips *et al*. 2008, 2013), is applied as before with $\tilde{n}_{IN,X,*}$ replacing $n_{IN,X*}$. Other temperature-dependent parameters in the EP are modified similarly with their temperature inputs shifted by $\Delta \tilde{T}_X(\tilde{t})$. Likely correspondences are discussed above for the assumption about which sample, $i$, best corresponds to each species, $X$, in the model. (Sec. 5).

This approach may be made to confer time-dependence on other INP schemes lacking it. An advantage of this approach with a clock tracer is that entrainment mixing of fresh INPs from the environment into the cold cloud region is represented during numerical solution of Eq (7). Such mixing tends to increase $Q$ towards unity and reduces $\tilde{t}$ accordingly.

Finally, if very long-lived clouds are being simulated, we recommend applying Eq (6) beyond $\tilde{t} = 10$ hours, providing $\Delta \tilde{T}_X < 10$K and thresholding at 10K otherwise. In view of the experimental limitations of our data, we make a simplifying assumption that the temperature shift is independent of temperature but has a different functional form for each INP species.

In summary, our lab observations with Eqs (5)-(8) provide a simple way to include time dependence in INP schemes commonly applied in atmospheric models. All mathematical symbols are summarised in Appendix B. This enables the glaciation of cold long-lived clouds to be simulated.

**6 Conclusions**

In the present study we present empirical data about the time dependence of heterogeneous ice nucleation for six ambient environmental aerosol samples. Ambient environmental samples, representing a variety of aerosol types expected to be dominated by certain INP species, were investigated. As they were ambient samples, they must be assumed to contain a complex composition, where multiple INP species may be active. Although this approach involves less certainty about the

915 chemical identity of the active INPs observed, it yields results with maximum realism.

The conclusions were as follows:

1. Clear effects from time dependency were observed on a level comparable to previous published works, with a percentage enhancement over 10 mins and 2 hours of about 20–40% and 40–100% respectively (Vali 1994; Welti *et*

*al.* 2012; Wright and Petters 2013; Budke and Koop 2015). There was variation seen among the various samples.

2. The repeatability of freezing of individual drops, which froze during the isothermal phase of the hybrid cooling-isothermal experiments, during the successive cycles of constant cooling (each being about 3 mins in duration for 5 K cooling), was mostly limited to about ± 0.3 K for the freezing temperature for all 6 samples. However, on the much

longer time scale (2 hours) of the original isothermal phase about half of them had frozen at a freezing temperature between 1 and 5 K warmer than for the constant cooling rate cycles. Thus, our observations reveal a minority of active INP with strong time dependence (large shift in freezing temperature) on hourly time scales, which also display only weak time dependence (small shift in freezing temperature) on short time scales of a few minutes.

3. In the isothermal experiments, there was an enhancement of active INP concentrations by about 40% to 100% and by about 70% to 200%, depending on the sample, over 2 and 10 hours respectively. The strongest time-dependence was seen for samples that we have inferred to be representative of mineral dust influenced and rural continental airmasses (Fig. 10), which displayed the lowest initial freezing fractions (0.1-0.2) among all samples (Fig. 9). Conversely, the least time-dependence was seen for samples inferred to represent marine and continental pristine/polluted airmasses,

which have the highest initial freezing fractions (0.4). Similarly, Herbert *et al.* (2014) reviewed lab data (e.g. Wright *et al.* 2013) showing montmorillonite in mineral dust is more time-dependent than other INP types (e.g. soot, soil).

4.  There is a general tendency for the natural time scale of the freezing to dilate increasingly as time progresses during each isothermal experiment:

    a.  The fractional real freezing rate steadily declines with time during the isothermal phase following a power-law.  At any instant, this real freezing rate equals the reciprocal of the natural time-scale of freezing.

    b.  This decline occurs because the 'faster' drops with a shorter natural timescale of freezing will freeze sooner, so the unfrozen drop population increasingly consists of 'slower' drops with longer timescales.   This in turn must be a consequence of variability among drops of the amount, composition and/or nucleating efficiency of immersed INP material.

    c.  A simple empirical isothermal formulation with an exponential dependence on time since the start of the isothermal phase, asymptotically approaching a maximum frozen fraction, is fitted to the average of all drop populations for any sample to represent its freezing at constant temperature over many hours. This time dependence is expressed in terms of the ratio of time to a natural relaxation time, which itself depends on time. This yields a simple power law dependence of the relaxation time on time, again with a steady dilation over time.

5.  Comparison of cooling-only and hybrid cooling-isothermal experiments reveals that exposure to constant conditions for long times causes an upward shift in freezing temperature that increases with time, by about 1-2 K after 2 hours for most drops.  There is a wide variation in the extent of this shift among the individual drops in any drop population, following a sigmoidal-like statistical distribution.

6.  A technique for representation of time dependence is proposed for incorporation into schemes of heterogeneous ice nucleation that currently omit time dependence (e.g., Phillips *et al.* 2008, 2013; DeMott *et al.* 2015; Patade *et al.* 2019, 2021), as are commonly used in cloud models (Eqs (5)-(8), Table 6, Sections 3.2.2 and 5). This involves a shift that depends on time for the ambient temperature input for these schemes when determining the active number of IN. Our observations reveal a simple power-law dependence of this ambient temperature shift with time, reaching about 3 K and 5 K of cooling over the initial 2 and 10 hours respectively.

In point 4, this natural time-scale of the freezing is defined here as the (time-dependent) relaxation time of the exponential factor for the approach of numbers of drops frozen towards an eventual asymptotic upper limit, and equals that for the decay of numbers of unfrozen drops towards their lower limit. We approximated both limits by observations at 10 hours.

Regarding point 3, the degree of time-dependence is somewhat weaker than seen by previous studies, which generally were not oriented towards sampling of ambient aerosol.  During the isothermal phase, Vali (1994) observed a doubling of the number

of frozen drops after only about 10 mins (-18 °C, freezing of distilled water drops on aluminium foil).  After 10 hrs, Wright and Petters (2013) saw an approximate quadrupling of the number of frozen drops at constant temperature (-22 °C, 1% wt of Arizona Test Dust [ATD] in each drop).  By contrast, for our mineral-dust influenced sample an increase by only about 150% was seen after 10 hrs, with much less change for most other samples.  This discrepancy reflects the need to study ambient aerosol samples for any lab experiments intended to be pertinent for real clouds.  A similar rationale was the basis for creation of the EP (Phillips *et al.* 2008, 2013).

This weakness of the degree of time-dependence observed here (point 1) makes the hypothesis about stochastic INPs from Westbrook and Illingworth (2013) seem less plausible.  From an observed flux of ice crystals falling out from a thin layer-cloud over UK, they estimated that the active INPs detectable by a CFDC would be removed after 3000 s whereas they observed persistence of crystal production over 24 hours.  If time-dependence were to account for this, it would need to boost INP activity by more than a factor of 30 in one day, which seems inconsistent with our observations.    On the other hand, the weakness of ice precipitation in that layer-cloud would imply weakness of SIP, allowing any time-dependence of INPs to affect the ice concentration.  A detailed modeling study is needed that treats SIP and vertical mixing by weak convective cells to quantify any role of time-dependence for the microphysical persistence of the cloud.

A striking implication from our results is that the instrumental uncertainties arising in field probes measuring active IN are at least as large as the time-dependence of ambient aerosol sampled, which is less than about a factor of two at -15 °C over 2 hours (point 1).  For example, when different field probes were compared for the same Saharan dust sample by DeMott *et al.* (2011, Figure 6 therein), a spread of up to a factor of 4 or 5 among active fractions was measured during the ICIS-2007 workshop.  Consequently, the limited extent of time-dependence that we report highlights the utility of field measurements of INPs with probes that have short residence times (e.g., about 10 sec for a CFDC), even when their data is applied in a time-independent (singular hypothesis) manner without classical nucleation theory in atmospheric models.

Any purely stochastic model of INP activity, assuming that the fractional freezing rate of all unfrozen drops is constant (e.g. Bigg 1953; reviewed by PK97), would predict very high frozen fractions after a certain time, which would be inconsistent with our measurements. Instead, the statistical variability of efficiencies among INPs must be accounted for with any application of stochastic theory.   Moreover, the invariance of our measured freezing spectra after repeated freezing cycles indicates that non-stochastic effects on time-dependence are minimal.

The singular model (e.g., the original EP; Sec. 1) commonly applied in atmospheric models would clearly be an adequate approximation for our observations since the degree of time-dependence seen is quite limited.  As is evident from Fig. 7, there is very high reproducibility of the frozen fraction as a function of temperature for repeated constant cooling ramps carried out on the same droplet population.  It is also evident from Fig. 8 (red error-bars), that the vast majority of studied droplets froze

at almost identical temperatures between repeated cooling ramps.   The impact of exposure to isothermal conditions for the extreme duration of 10 hours is observed to be merely a change of freezing temperature by 2 or 3 K for most drops monitored individually (the modelled temperature shift inferred in Eq (6) for treatment of time-dependence in INP schemes is comparable but about twice as large; c.f. Fig. 13).  Some pronounced variability in freezing temperature is measured for a minor subset of droplets, however.


Finally, in any future similar lab experiments with ambient aerosol samples, it will be beneficial to target major INP types using recently established methods (e.g., Testa *et al.*, 2021). These could allow samples to be focused with more certainty on mineral dust or inorganic carbonaceous INPs such as black carbon.

In summary, the time dependence of INP activity of ambient aerosols sampled from the planetary boundary layer was characterized. This reveals a steady dilation of the natural time scale of freezing during exposure to constant conditions of temperature as the more efficient INPs are depleted.   A simple empirical approach is provided for INP schemes of atmospheric models by introducing a time-dependent temperature shift (Eqs (5)-(8)). This enables simulations to assess whether there is any significant  impact from time-dependence on the glaciation and precipitation of cold long-lived clouds, observed in field

campaigns (e.g. Westbrook and Illingworth 2013; Fridlind *et al.* 2017).

### 7 Acknowledgements

The project was funded by a research grant to VTJP, from the Swedish Research Council for Sustainable Development (''FORMAS'' award 2018-01795). The topic of this award concerns the effects on clouds and climate arising from the time-
dependence of heterogeneous ice nucleation.    VTJP planned and directed the present study. Also, we thank the Royal Physiographic Society of Lund, the Walter Gyllenberg Foundation, the Crafoord Foundation (grant 20190964) and the strategic research area, MERGE, for financial support. The research leading to the supportive aerosol results from Hyltemossa have received funding from the European Union's Horizon 2020 research and innovation programme under grant agreement No 654109. We thank Jan Pallon and Mikael Elfman for carrying out the PIXE measurements and the associated data analysis.
We thank Madeleine Peterson Sjögren for providing Bio Trak data. We thank John Falk for carrying out cold stage measurements. We thank Patrik Nilsson, Marcin Jackowicz-Korczynski, Erik Ahlberg and Adam Kristensson for supportive aerosol data, for support with filter sampling and operation of the Hyltemossa field station. We thank ACTRIS for support and access to data. We thank EMEP for access to PM data.

**8 Author Contributions**

JJ carried out the experimental work with support from TBK. JJ did most of the data analysis with support from VTJP and minor contributions from TBK. DW performed the statistical analysis. All authors were involved in the experimental design and the interpretation of results. JJ and VTJP drafted the first manuscript version together, and all authors contributed to the manuscript writing.


**9 Competing Interests**

The authors declare that they have no conflict of interest.

**10 Data availability**

All data from the experiments documented are to be archived on a link from a public web-page describing the wider project (FORMAS) that supported the work.

 **Appendix A: Statistical analysis of INP concentration at -15 ºC cold stage temperature**

The measurements of INP activity among all six aerosol samples were analysed with a statistical test to detect if they differ from one another. Results are summarised in Tables A1. Figure A1 schematically illustrates the differences in freezing behaviour among the six aerosol classes based on the INP concentration at a cold stage temperature of -15ºC from two-sample

statistical F test for each pair in possible permutations. This test assumes that measurements of this concentration are normally distributed in each sample.

Some of the aerosol classes show similarity. Considering results from the statistical analysis, almost all aerosol classes are different from each other except these pairs:


- combustion-dominated vs mineral dust-influenced
- rural continental-continental polluted
- continental polluted-continental pristine
- continental pristine-marine samples.


Finally, the various aerosol classes can be ranked based on their '*freezing uniqueness*' (Tables A2 and A3). Continental pristine and continental polluted samples are the least unique. The rest are equally unique. None of the samples are perfectly unique and each is statistically similar to at least one other sample.

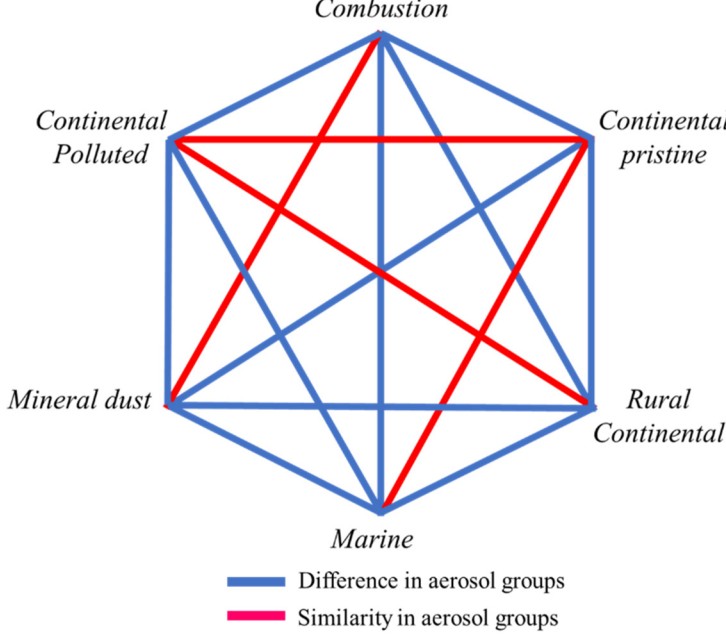

**Figure A1:** Web-chart showing difference and similarity in INP concentrations among six aerosol classes at -15°C cold stage temperature. Red lines show no statistically significant difference. Statistically significant differences for pairs of aerosol samples are shown by blue lines.

**Table A1:** Statistical 'F' test analysis of INP concentration at -15°C cold stage temperature for pairs of samples. Estimated and critical values of the sample test statistic, $F_{critical}$ and $F_{estimated}$, for each pair of aerosol classes. When the sample test statistic is less than the critical value, then the pair does not differ significantly.

| Aerosol group | $F_{critical}$ ($\alpha = 0.05$) | $F_{estimated}$ |
|---|---|---|
| Combustion-rural continental | 2.02 | 145.314 |
| Combustion-mineral dust | 1.81 | 1.3687 |
| Combustion-polluted | 1.83 | 134.6907 |
| Combustion-pristine | 1.5 | 6.5217 |
| Combustion-marine | 1.67 | 9.4526 |
| Rural continental-mineral dust | 2.09 | 106.1692 |
| Rural continental-continental polluted | 2.09 | 1.0788 |
| Rural continental-continental pristine | 1.94 | 22.2815 |
| Rural continental-marine | 1.85 | 15.3729 |
| Mineral dust-continental polluted | 1.9 | 98.4075 |
| Mineral dust-continental pristine | 1.59 | 4.7648 |
| Mineral dust-marine | 1.76 | 6.9062 |
| Continental polluted-continental pristine | 1.74 | 0.0484 |
| Continental polluted-marine | 1.83 | 14.2496 |
| Continental pristine - marine | 1.58 | 1.4494 |

**Table A2:** Freezing uniqueness score of aerosol groups.  Higher values of the score imply more uniqueness.

| Aerosol group | Freezing uniqueness score |
|---|---|
| Marine dominated | 3 |
| Mineral dust influenced | 3 |
| Continental pristine | 1 |
| Continental polluted | 1 |
| Combustion dominated | 3 |
| Rural continental | 3 |

**Table A3:** Uniqueness of aerosol samples based on statistical 'F' test of INP concentration of each aerosol group at -15°C cold stage temperature.

| Freezing uniqueness ranking | Aerosol class |
|---|---|
| 1 | Combustion, Rural continental, Mineral dust, Marine |
| 2 | Continental Pristine, Continental polluted |

## Appendix B: List of Symbols

**Table B1**: List of symbols.

| Symbol | Meaning | Units |
|---|---|---|
| | | |
| $A_i$ | Constant of proportionality for power-law dependency of temperature shift on time | K s$^{-\beta}$ |
| $C_i$ | Constant of proportionality for power-law dependency of relaxation time-scale on time | s$^{1-\alpha}$ |
| $i$ | Label for sample of ambient aerosol ($i$ = 1 to 6) | - |
| $n_{IN,X*}$ | Number mixing ratio of reference activity spectrum of INPs from aerosol group $X$ at water saturation in the background-troposphere scenario of EP (Phillips *et al.* 2008, 2013) | kg[air]$^{-1}$ |
| $\tilde{n}_{IN,X*}$ | Modified value of $n_{IN,X*}$ accounting for time-dependence | kg[air]$^{-1}$ |
| $f_{ice}$ | The frozen fraction in the isothermal phase (either -14 or -16 ºC), being the number of drops frozen since 0 ºC divided by initial number of liquid drops at 0 ºC ('frozen fraction') | - |
| $f_{ice,0}$ | Value of $f_{ice}$ at start of isothermal phase ($t^*$ = 0) | - |
| $\Delta f_{ice,\infty}$ | Eventual maximum increase in $f_{ice}$ at the longest times observed | - |
| $f_u$ | Number of drops not yet unfrozen at any instant among those that will eventually freeze during long isothermal experiments (10 hrs), divided by the initial number of drops at 0 ºC | - |
| $F_{estimated}$ and $F_{critical}$ | Sample statistic for the F-test and its critical value | - |
| $S_i$ | Saturation ratio of vapour with respect to ice | - |
| $Q$ | Passive 'clock' tracer | kg[air]$^{-1}$ |
| $Q_0$ | Value of $Q$ (set to unity) outside the cold-cloud region (subzero temperatures with appreciable IWC) | kg[air]$^{-1}$ |
| $\tilde{Q}$ | Value of $Q$ from analytical solution for hypothetical adiabatic parcel | kg[air]$^{-1}$ |

| | | |
|---|---|---|
| $t$ | Time | s |
| $t^*$ | Time since start of the isothermal phase of lab experiment | s |
| $\tilde{t}$ | Age of in-cloud parcel since first entering the subzero region of the cloud | s |
| $T$ | Physical temperature of ambient air | ºC |
| $\Delta\tilde{T}_X$ | Temperature shift applied to temperature input of scheme for INPs in aerosol group $X$ | K |
| $X$ | Label for group of insoluble aerosol in scheme for heterogeneous ice nucleation | - |
| $\alpha$ | Exponent in power-law dependency of relaxation time-scale on time | - |
| $\beta$ | Exponent in power-law dependency of temperature shift on time | - |
| $\chi$ | Fractional change in number of frozen drops after 10 hrs during the isothermal experiment | - |
| $\tau$ | Relaxation time-scale in isothermal formulation of $N_{ice}$ | s |
| $\tau_Q$ | Relaxation time-scale of $\tilde{Q}$ | s |
| $\hat{\tau}$ | Power-law fit to inferred value of $\tau$ for $i$-th sample, as a function of $t^*$ | s |
| $\Omega_X$ | Total surface area of all aerosols larger than 0.1 microns in diameter from group $X$ (not depleted by ice nucleation while inside cloud) | [aerosol] $m^2$ [air] $kg^{-1}$ |

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

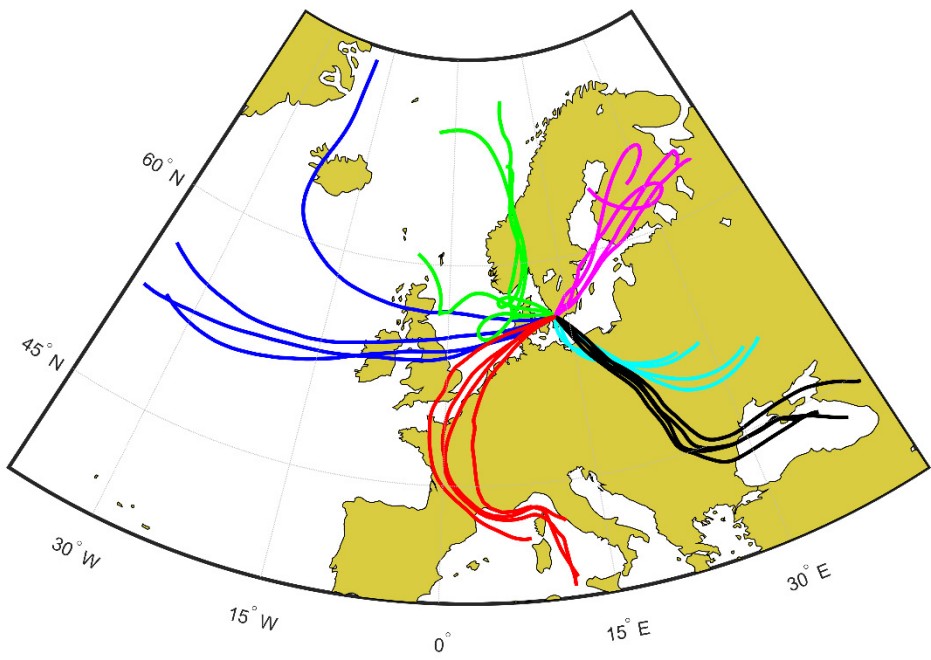

**Figure 1:** Back trajectories (120 hr) for the 6 aerosol samples from the HYSPLIT model (Stein *et al*. 2015). The displayed samples are marked in various colours (blue = Marine, red = Mineral dust influenced, magenta = Continental pristine, cyan = Continental polluted, black = Combustion dominated and green = rural continental). The individual lines for each sample show back trajectories for the airmass arriving at the Hyltemossa station initiated every 6 hours during the sampling dates, 500 meters above ground level. Similar wind directions were observed for the back trajectories arriving at 50 and 2000 m above ground level.

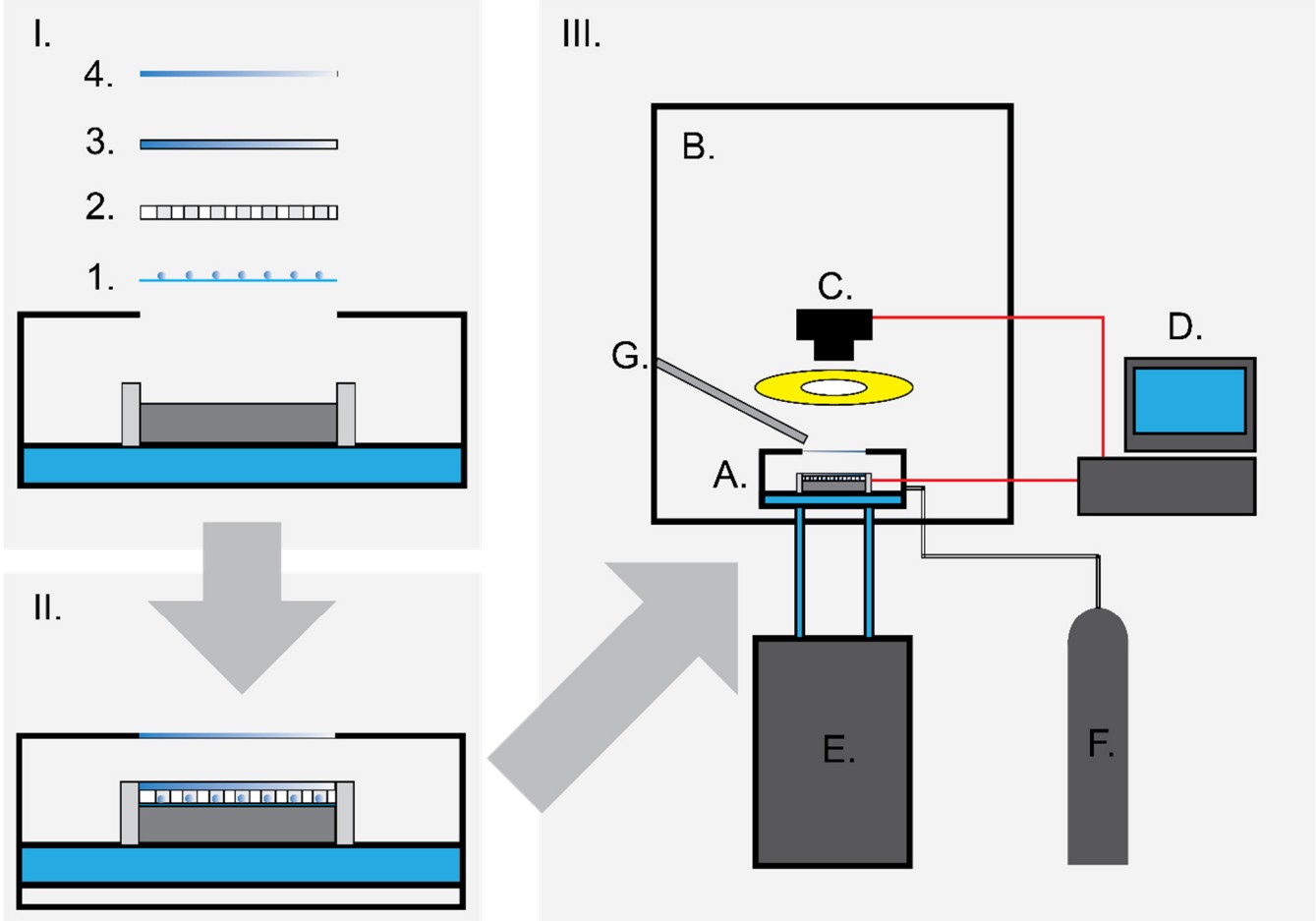

**Figure 2:** The Lund University Cold-stage (LUCS). **Upper left panel, I:** The freezing array design. The sample is placed on siliconized hydrophobic glass substrates (1) which are placed on the cold-stage. The sample drops are separated by a silicone grid (2) confining each drop to an individual cell, between the glass substrate and a polycarbonate lid (3). The freezing array is in turn encased in a small environmental chamber, with a quartz viewing window (4). **Lower left panel, II:** The sample mounted in the assembled freezing array. **Right panel, III:** Overview of the LUCS system. The cold-stage (A) is placed in a laminar airflow cabinet (B) to avoid any possible contamination. A camera system with a circular continuous LED light source is fixed above the cold-stage's viewing window and focused on the cold-stage. The camera is controlled by a computer (D), and a cryogenic water circulator (E) provides the cold-stage with 8.5 °C cooling water, acting as a heat sink for the device. The environmental chamber is continuously purged with dry nitrogen gas, and a flow of dry, HEPA-filtered air (G) is directed at the viewing window to eliminate potential fogging at low temperatures.

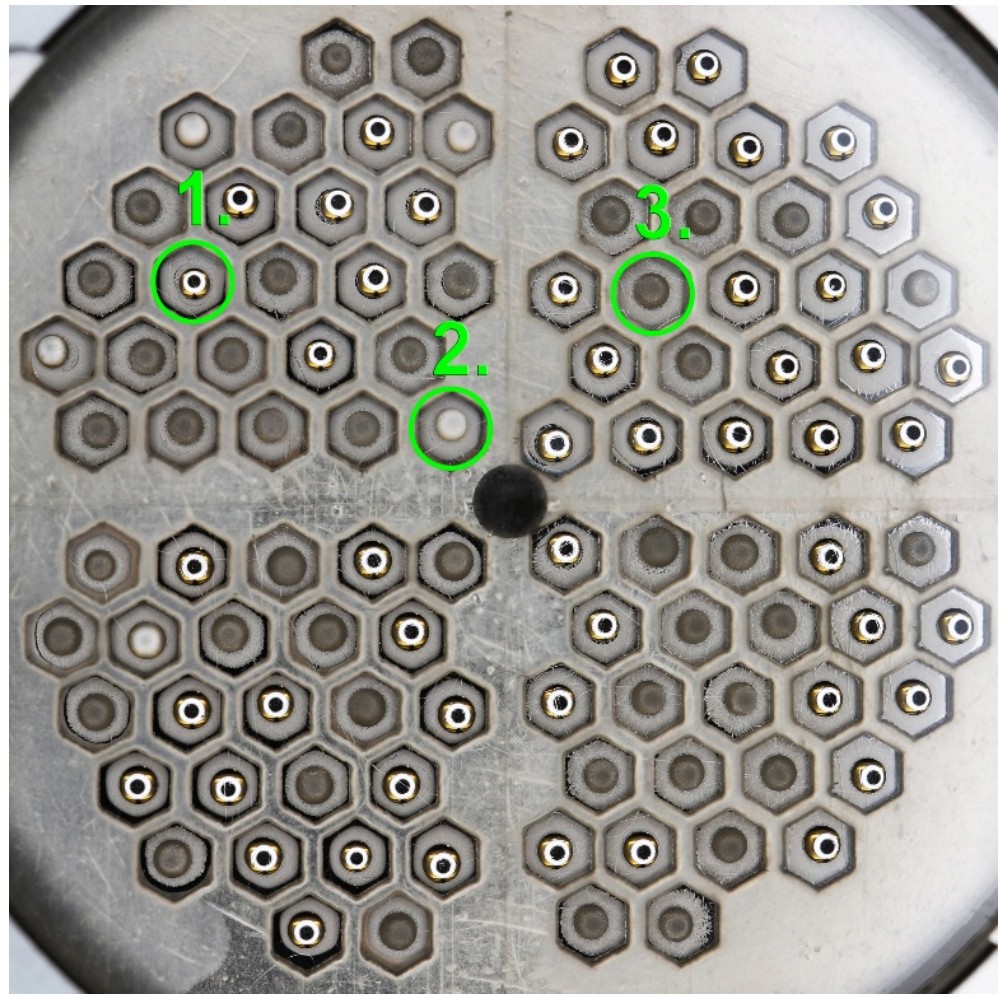

**Figure 3:** An image generated by LUCS, used as the input data to determine drop freezing events. The image shows the outlay of the 100 sample drops on the 40 x 40 mm stage; each drop confined to a sealed cell. The image shows drops that are still in the liquid phase and reflect the light from the circular light source (1), drops that are just undergoing freezing (2) and drops that are fully frozen (3).

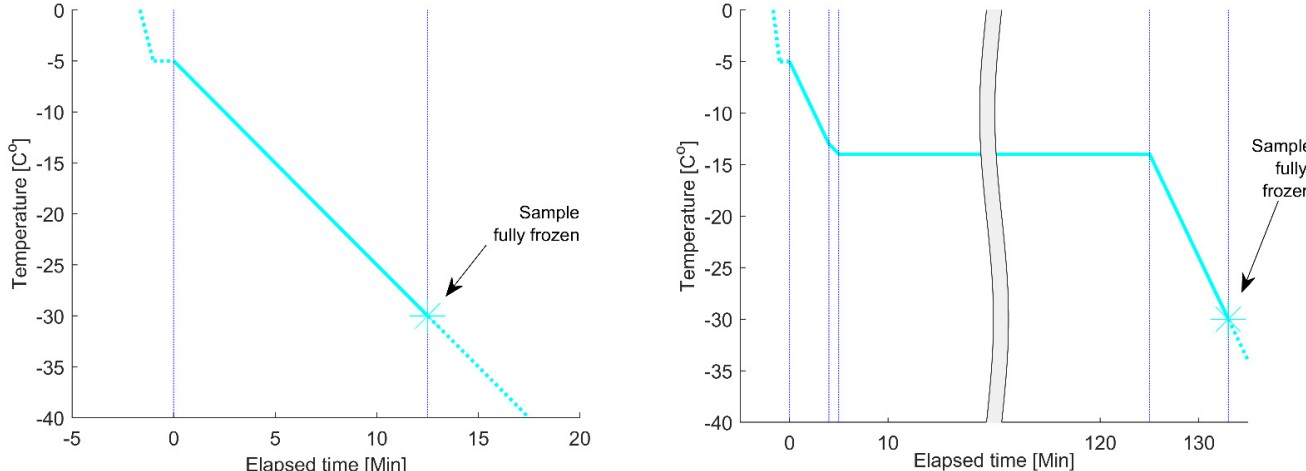

**Figure 4:** The cooling programs used in this study. The cyan lines display the evolution of temperature as a function of time, the blue dotted lines indicate where the cooling rate was changed and where the experiment was completed (when the sample was fully frozen). Left panel: the cooling program for the constant cooling rate experiments. Right panel: The cooling program used for the 2-hour isothermal experiments. One isothermal experiment with a longer isothermal period (11 – 16 hours) was

also done on each drop population (not shown). In this figure, the timescale starts (t = 0) at the first cooling ramp after the 1 min pause at -5 °C, all times referred to in the analysis for the isothermal experiments are from the start of the isothermal phase, if nothing else is specifically stated.

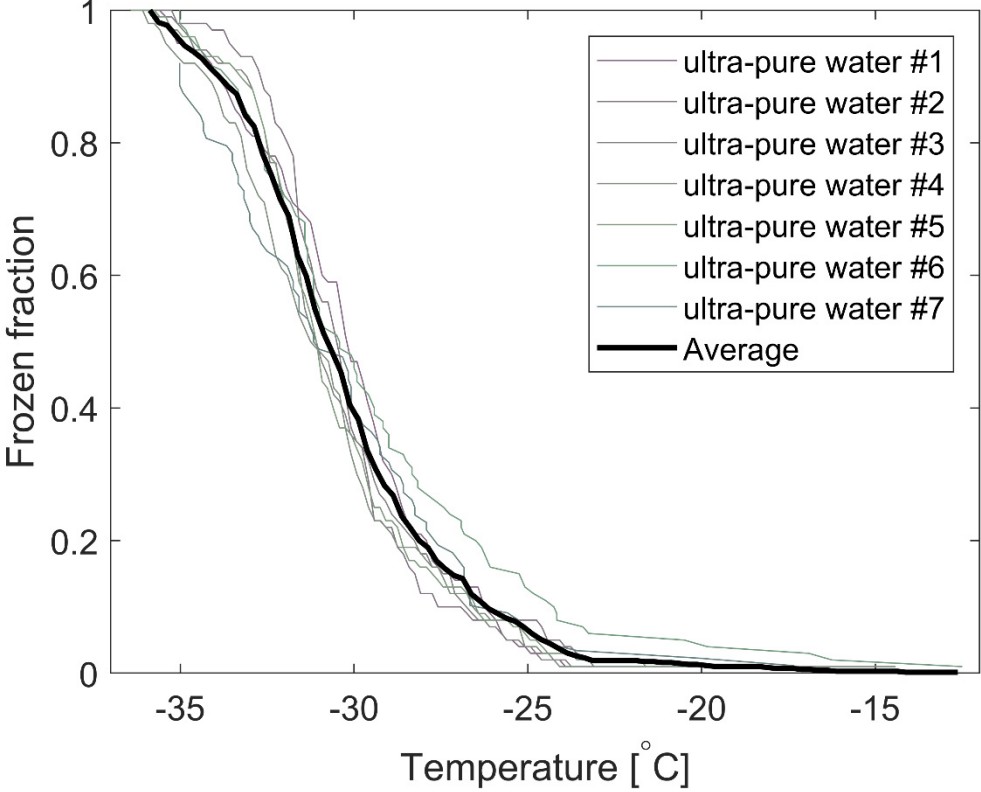

**Figure 5:** The frozen fraction of ultra-pure water droplets (volume 1 μL) for seven different experiments. The average for all runs is included. In the vast majority of such runs, the first droplet typically freezes for a temperature slightly below -20°C. Within the presented ensemble, one sample shows an elevated and unusually high number of droplets frozen for temperatures above -20°C. Those observations were most likely associated with unusually poor quality of the glass slides used in that

experiment.

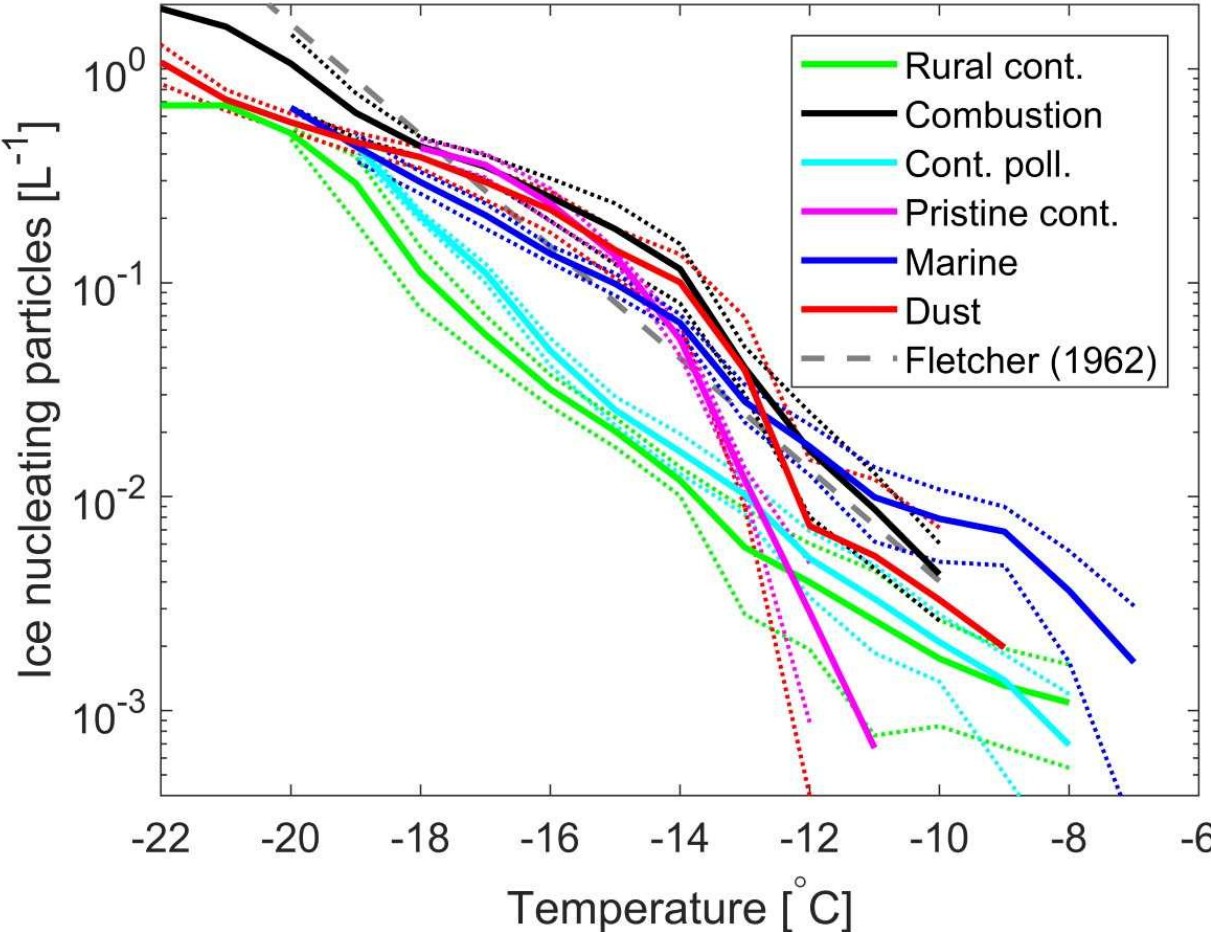

**Figure 6:** Ambient concentrations (per Litre of air) of ice nucleating particles versus temperature for the six samples of aerosol material studied (dark blue for the marine dominated sample, red for the mineral dust influenced sample, magenta for the continental pristine sample, cyan for the continental polluted sample, black for the combustion dominated sample and green for rural continental sample). The dotted lines indicate the random error range corresponding to ±one standard deviation. The Fletcher (1962) parameterisation is included for comparison. The averaged INP spectra are based on the first measurements for five different droplet populations (in total 500 droplets, each 1 μL) per sample with a constant cooling rate (2 K min$^{-1}$). The reproducibility was generally very high for a given sample at temperatures below -14°C. Variability was more pronounced in the high temperature range of the spectra, where fewer freezing incidents were observed.

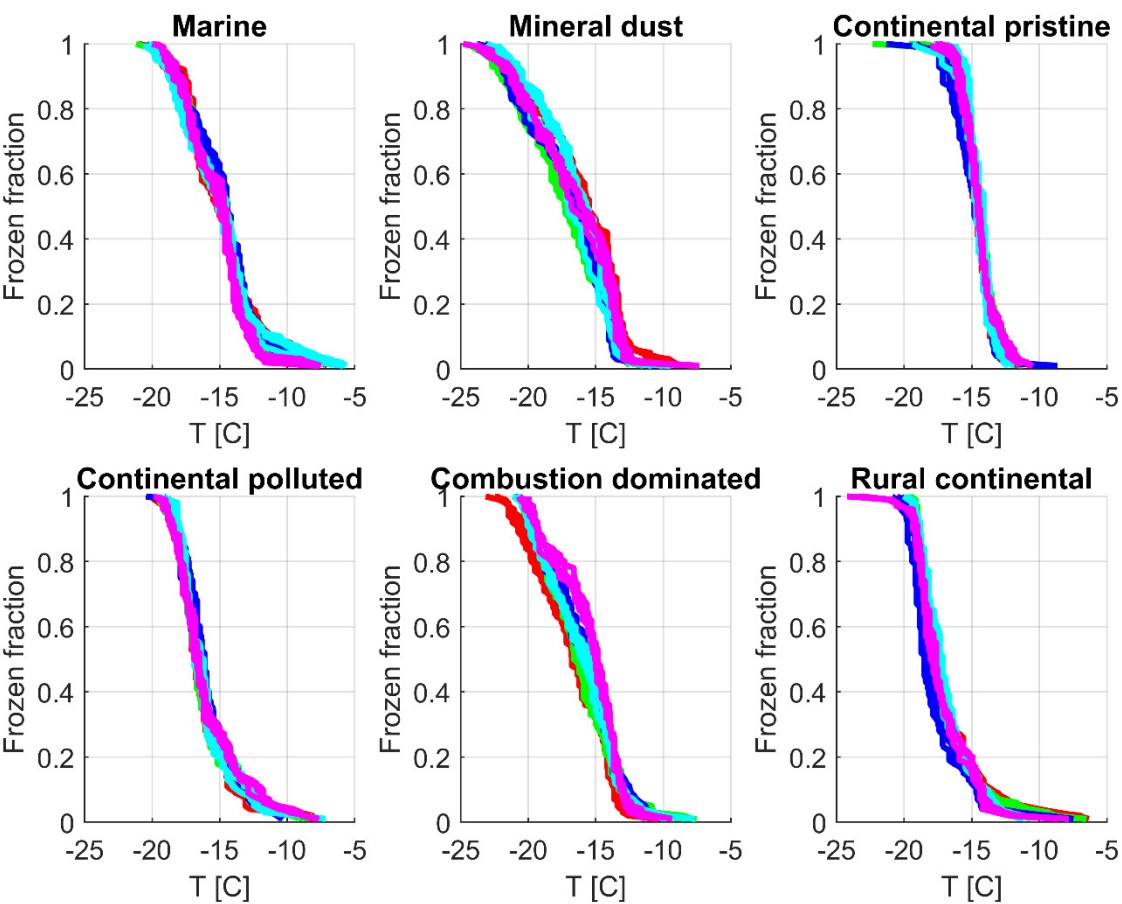

**Figure 7:** Frozen fraction as a function of cold-stage temperature for the 6 samples measured at constant cooling rate (2 K min$^{-1}$), (6 panels: Marine, Mineral dust influenced, Continental pristine, Continental polluted, Combustion dominated and rural continental samples). Each colour represents one of five different drop populations from any given sample exposed to the same cooling cycle starting at -5°C. For each drop population, at least 4 curves of the same colour are shown, each representing experiments at different times before, between and after the isothermal periods (2 -16 hours) regarding each panel. The time resolution of the imagery was 5 s which limits the accuracy of the determination of the freezing temperature (±0.1K).

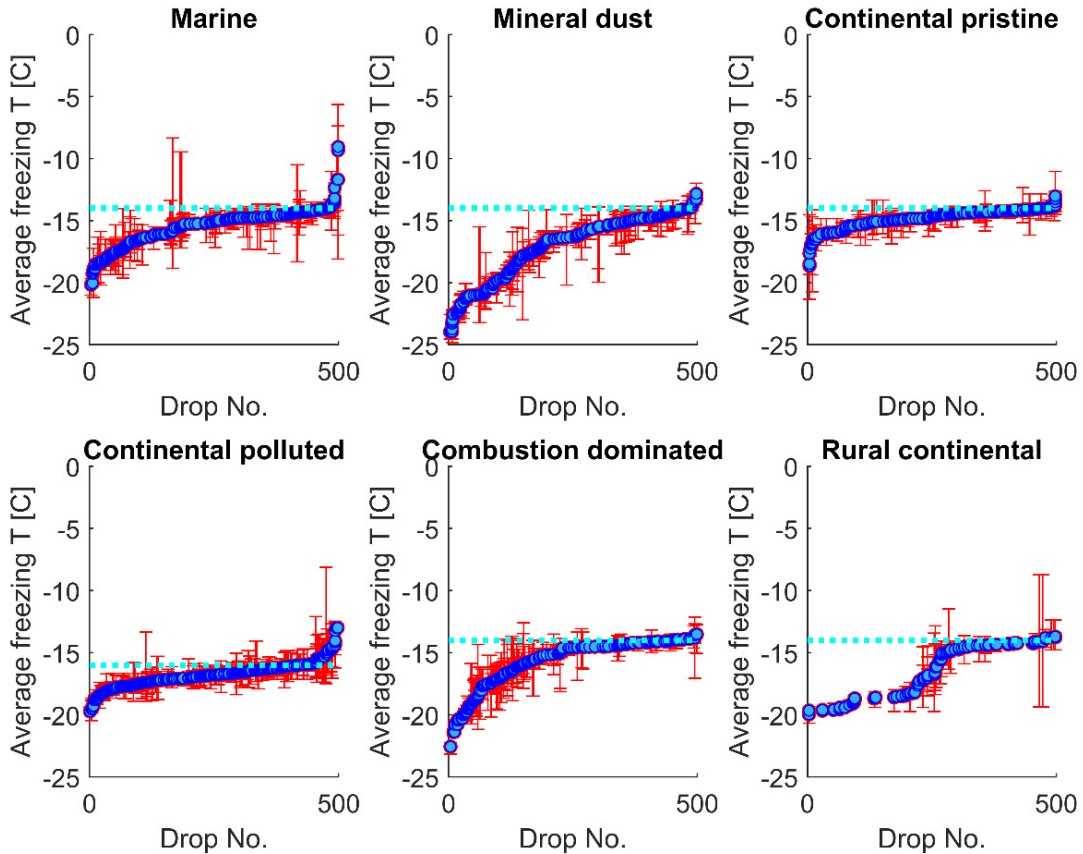

**Figure 8**: The average freezing temperature of each individual drop (dark blue circles), and the freezing range (from minimum to maximum) for each drop (red error-bars), measured during the same constant cooling rate experiments depicted in Fig. 6 (minimum of 4 cooling cycles), for the marine dominated, mineral dust influenced, continental pristine, continental polluted, combustion dominated and rural continental samples of aerosol material (panels from top left to lower right). All drops displayed also froze during the isothermal experiments (up to 10 hours). The temperature for the isothermal experiments is marked in each panel (cyan dotted line). The median value of the standard deviation for the normal freezing temperature of the individual drops were in the range between 0.25-0.34K. The median of the temperature ranges (absolute difference between minimum and maximum values) for the freezing temperature of the individual drops were 0.53-0.93K.

(a)

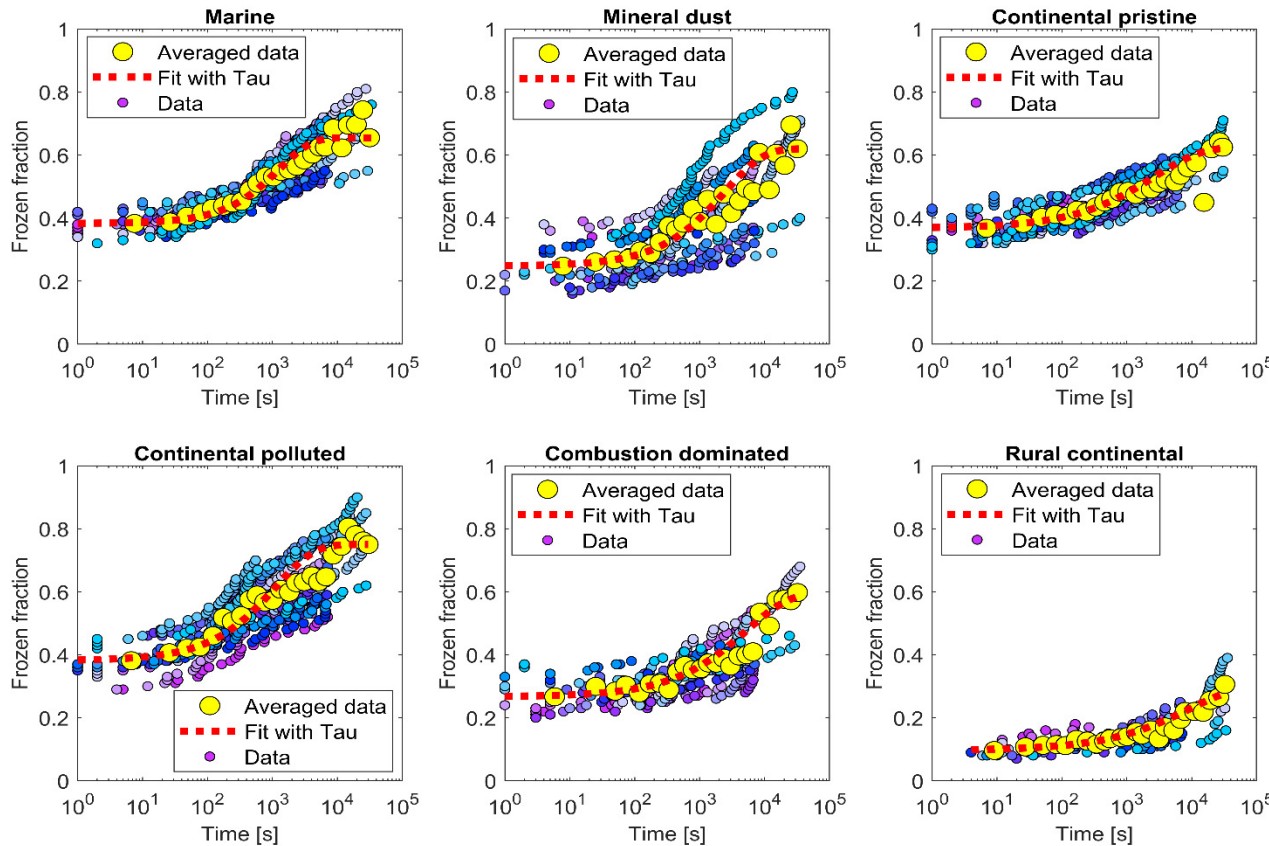

**(b)**

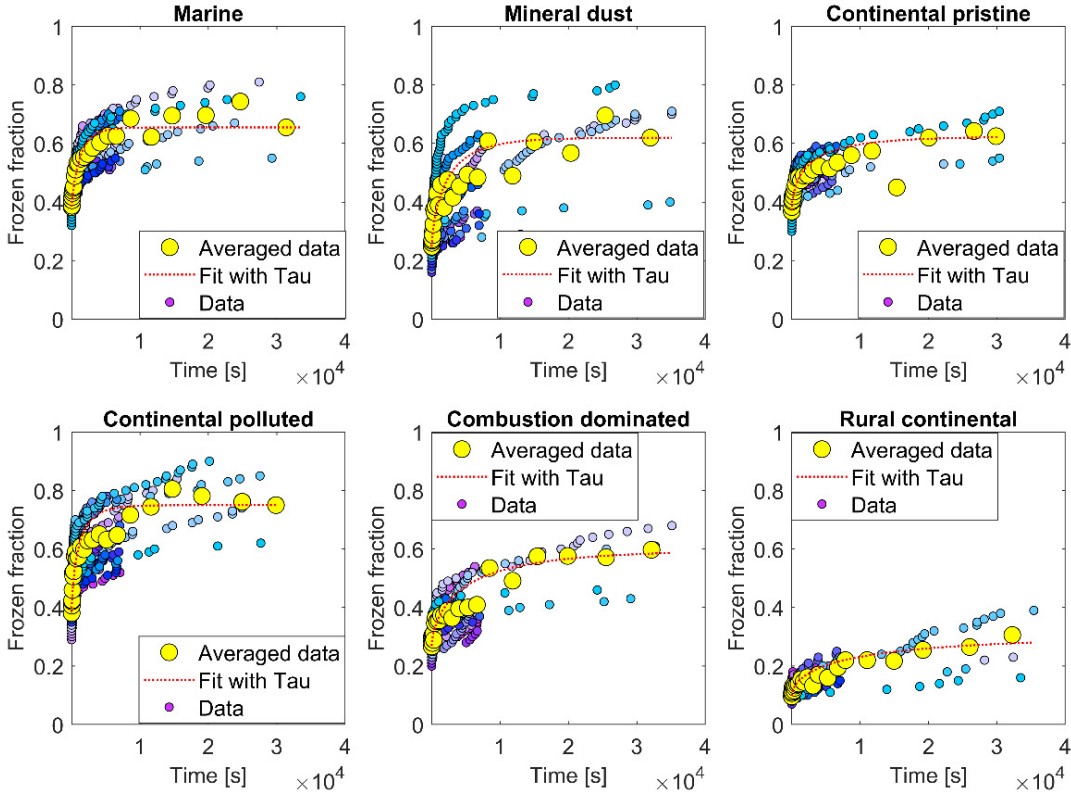

**Figure 9:** (a) The measured frozen fractions ($f_{ice}(t^*)$) as a function of time since the start of each isothermal experiment (dots of various shades of blue and mauve, one shade for each drop population). The measurements are averaged over all drop populations (yellow dots), for each of the marine dominated, mineral dust influenced, continental pristine, continental polluted,

combustion dominated and rural continental samples of aerosol material (panels from top left to lower right). Also shown are the empirical fits using Eq (2) and the empirically determined relaxation time ($\hat{\tau}(t^*)$) from the fits (Eq (4)) for the averaged data for the samples (red line). Also shown is (b) the same plots displayed with a linear axis for time.

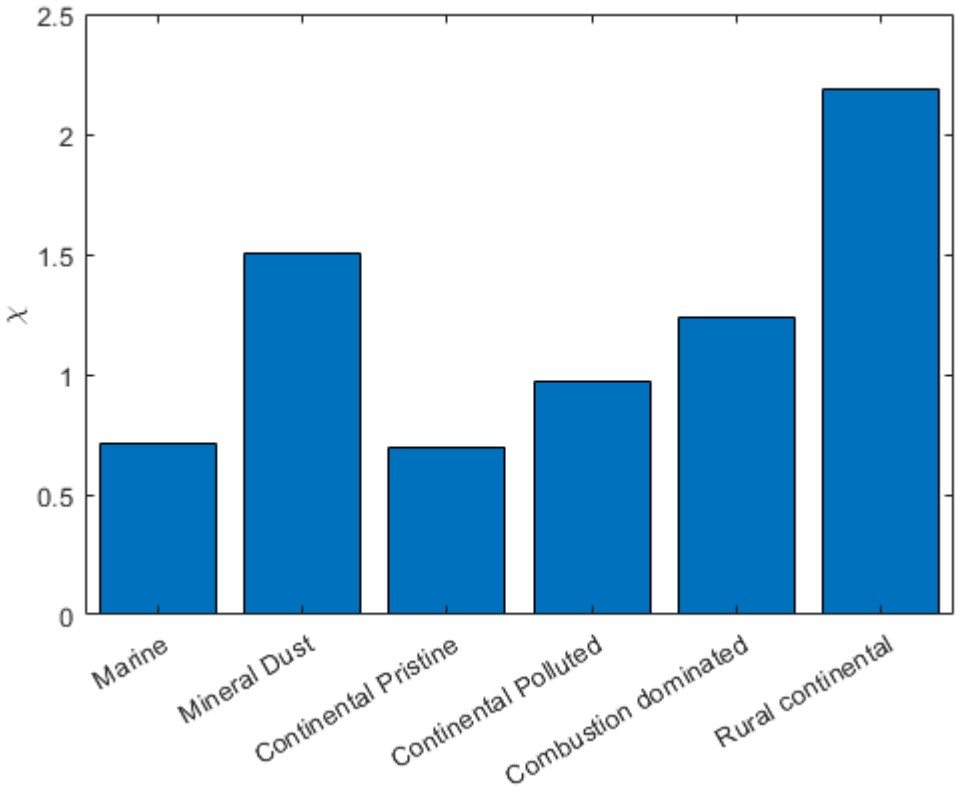

**Figure 10:** Fractional increase, $\chi$, after 10 hours of the isothermal experiment in frozen fraction averaged as in Fig. 9 (initial and final yellow points), for the marine dominated, mineral dust influenced, continental pristine, continental polluted, combustion dominated and rural continental samples of aerosol material (left to right).

 (a)

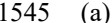

(b)

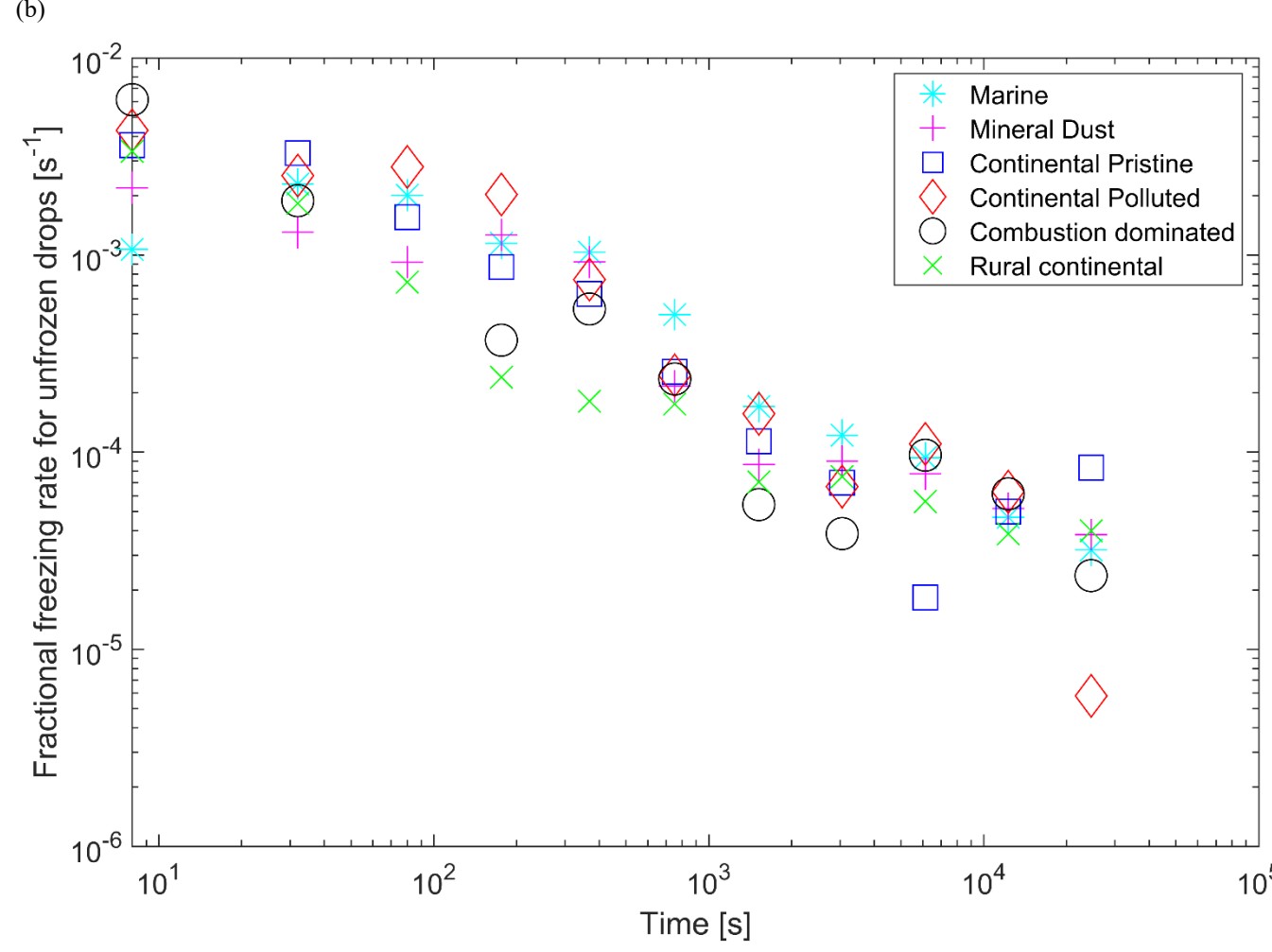

**Figure 11:** (a) Fractional rate of change of the frozen fraction, $f_{ice}(t^*)^{-1} \, df_{ice}(t^*)/dt^*$, plotted as a function of time since the

1555 start of the isothermal phase for the marine dominated, mineral dust influenced, continental pristine, continental polluted, combustion dominated and rural continental samples. This was estimated by a finite difference approximation to the derivative for consecutive values in the time series of the data. Only positive estimates of these rates are plotted. (b) Also shown is the corresponding fractional rate of change of the number of unfrozen drops, $(f_{ice}(t^* = 10 \text{ hrs}) - f(t^*))^{-1} \, df_{ice}/dt^*$, considering only the drops that eventually freeze after 10 hours in the isothermal experiment.

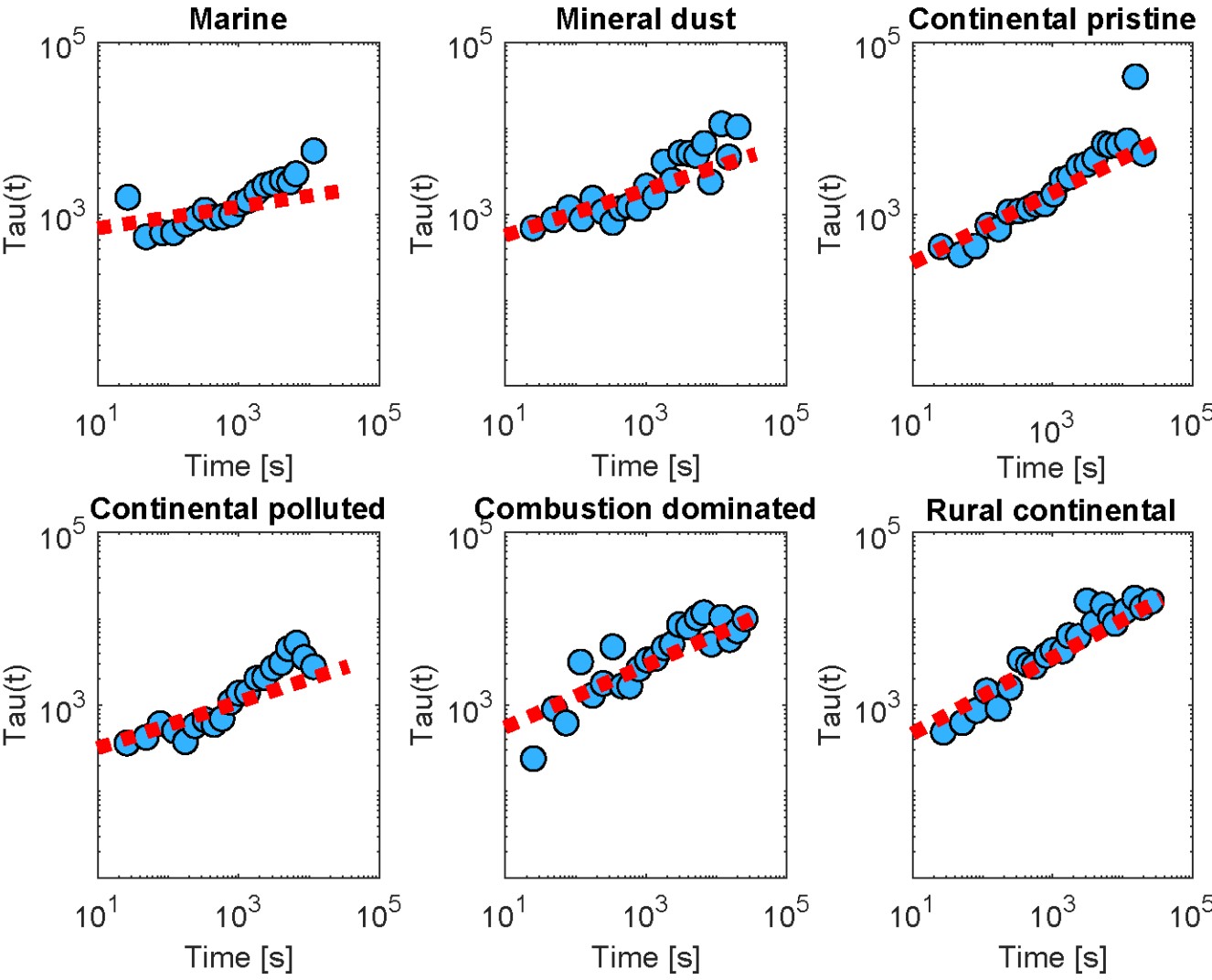

**Figure 12:** The relaxation time τ inferred from the isothermal experiments (up to 10 hours) using the averaged data (yellow dots in Figure 9), with Eq (3) as a function of time since the start of the isothermal phase (blue dots) for the marine dominated, mineral dust influenced, continental pristine, continental polluted, combustion dominated and rural continental samples of aerosol material (panels from top left to lower right). Also shown are empirical fits for all measurements of each sample using Eq (4), (red dotted line).

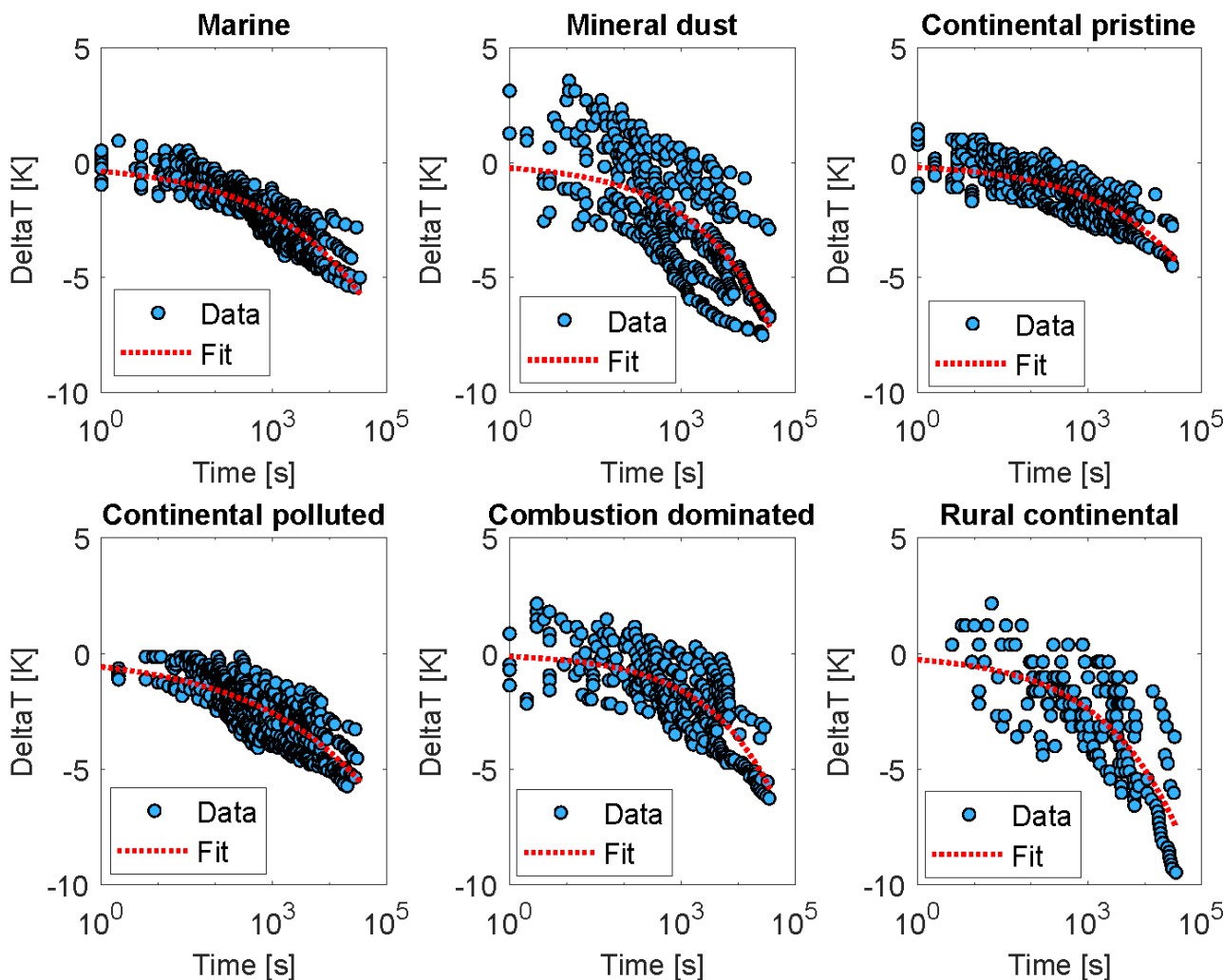

**Figure 13:** The apparent temperature shift, $\Delta T$, required to match the empirical parameterization of the INP activity (Phillips *et al.* 2013) with the measurements of frozen fraction from the isothermal experiments (blue dots), for the marine dominated, mineral dust influenced, continental pristine, continental polluted, combustion dominated and rural continental samples of aerosol material (panels from top left to lower right). Evolution over time since the start of the isothermal phase is displayed. Also shown is the fit (Eq (6)) assuming a power law dependent on time (red dotted line).

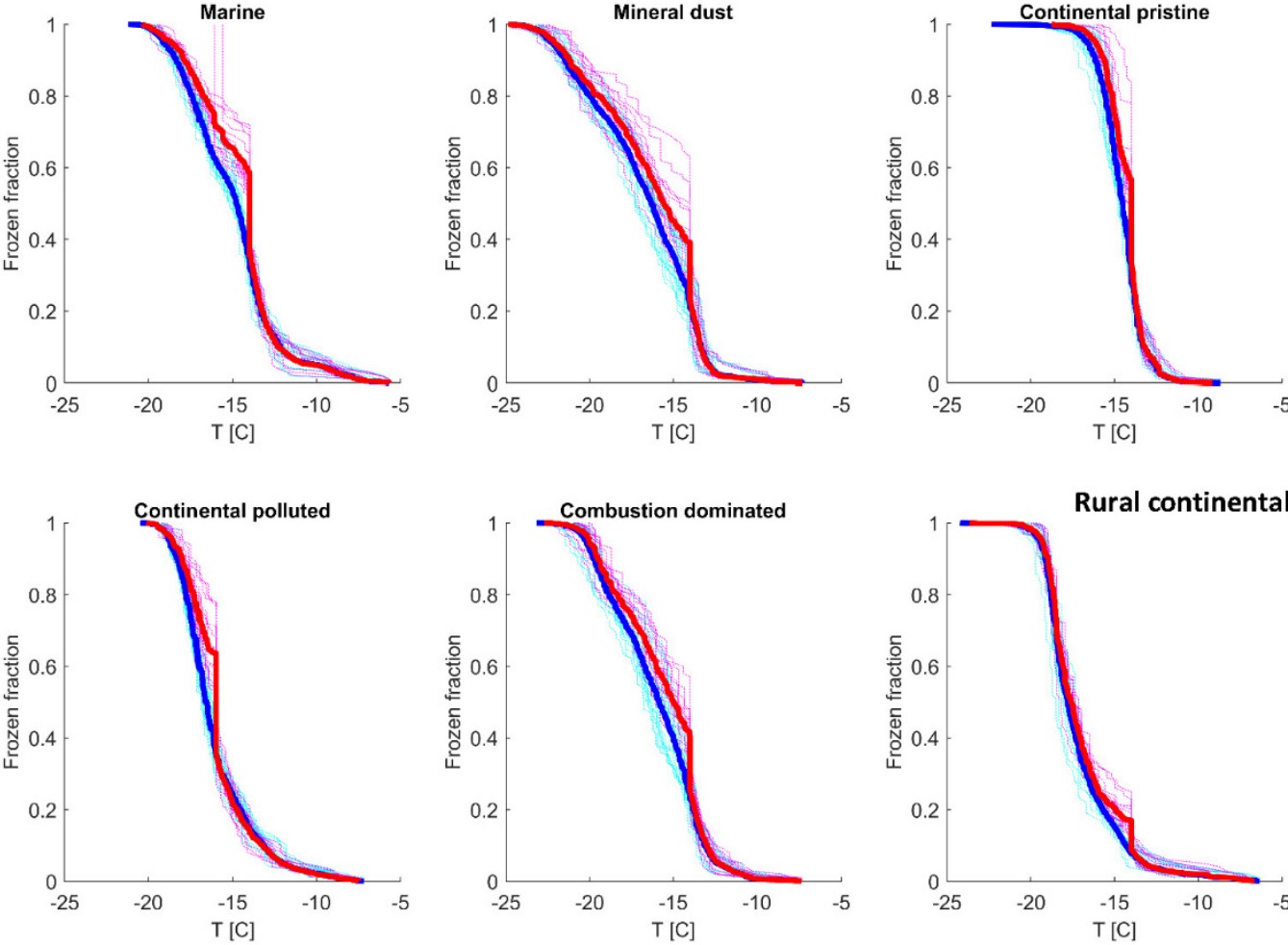

**Figure 14:** Frozen fraction raw data for the constant cooling rate experiments (cyan dotted lines) and for isothermal experiments of 2 hours (magenta dotted lines) for the marine dominated, mineral dust influenced, continental pristine, continental polluted, combustion dominated and rural continental samples of aerosol material (panels from top left to lower right). The large steps for two isothermal experiments displayed for the marine sample is an effect of a camera lag, not a physical effect, as they occur after the isothermal phase this does not influence the result. The averaged data for constant cooling rate experiments (blue lines) and isothermal experiments (red lines) are also shown. The offset seen for the isothermal experiments at the isothermal temperature is the effect of the 2-hour isothermal phase of the experiment. As seen in the figure, the samples behave almost identically until the isothermal phase is reached, and the isothermal data approaches the constant cooling rate data after the isothermal phase has elapsed, and constant cooling is resumed.

# Tables

**Table 1:** Aerosol particle properties for the six studied samples during the respective sampling periods. An additional dust dominated sample has been included to present information about the PM composition of relevance to the studied dust sample. The concentrations of particulate matter (PM) were measured at the nearby Hallahus station. The black carbon (BC) concentrations were inferred from the aethalometer (AE33) at Hyltemossa. Non-refractory $PM_1$ components were measured with the ACSM at Hyltemossa. The same filter samples were used for PIXE and the cold stage measurements, with the exception of the dust sample. Supermicron primary biological aerosol particles (PBAPs) were detected with a Bio Trak instrument at Hyltemossa, but mineral dust particles may also fluoresce. N/A refers to the data not being available due to malfunctioning instruments or used up samples. Meteorological seasons are indicated.

| | | | | | OPC | | | AE33 | ACSM | | | | | PIXE | | | | | Bio Trak |
|---|---|---|---|---|---|---|---|---|---|---|---|---|---|---|---|---|---|---|---|
| Date | Category | T °C | RH % | Vol $m^3$ | $PM_{10}$ µg/m³ | $PM_1$ µg/m³ | $PM_{10}$-$PM_1$ µg/m³ | BC µg/m³ | Org µg/m³ | $NH_4$ µg/m³ | Cl µg/m³ | $NO_3$ µg/m³ | $SO_4$ µg/m³ | Si µg/m³ | Ca µg/m³ | K µg/m³ | S µg/m³ | Cl µg/m³ | Fluoresing particles (e.g., PBAPs) $L^{-1}$ |
| 2020-03-26 (early Spring) | Continental Polluted | 3.8 | 52 | 23.3 | 24.3 | 16.5 | 7.8 | 0.72 | 5.1 | 2.37 | 0.12 | 5.41 | 0.87 | N/A | N/A | N/A | N/A | N/A | N/A |
| 2020-09-05 (early Autumn) | Marine | 13.6 | 79 | 16.8 | 9.24 | 2.3 | 7.1 | 0.11 | 1.7 | 0.34 | 0.07 | 0.74 | 1.3 | 0.20 | 0.08 | 0.09 | 0.23 | 2.04 | N/A |
| 2020-09-18 (early Autumn) | Rural continental | 11.2 | 70 | 18.4 | 5.84 | 0.9 | 4.7 | 0.04 | 1.1 | 0.07 | 0.01 | 0.03 | 0.15 | 0.11 | 0.06 | 0.08 | 0.06 | 0.28 | N/A |
| 2020-11-29 (late Autumn) | Pristine continental | -0.9 | 99 | 24.0 | 3.41 | 1.7 | 1.6 | 0.10 | 0.9 | 0.04 | 0.08 | 0.50 | 0.3 | 0.17 | 0.01 | 0.03 | 0.19 | 0.03 | 0.2 |
| 2020-12-07 (early Winter) | Combustion-dominated | 6.4 | 88 | 6.5 | 20.85 | 18 | 3.0 | 0.76 | 8.0 | 3.64 | 0.23 | 7.92 | 4.7 | 0.71 | 0.11 | 0.15 | 2.80 | 0.15 | 3.5 |
| 2021-02-23 (late Winter) | Mineral dust influenced (Saharan dust) | 5.8 | 87 | 6.1 | N/A | N/A | 11[a] | 1.32[b] | 6.7 | 1.88 | 0.06 | 4.72 | 1.02 | N/A | N/A | N/A | N/A | N/A | 9.2 |
| 2021-02-25 (late Winter) | | 9.0 | 89 | 5.4 | N/A | N/A | 14[a] | 1.33[b] | 6.2 | 3.72 | 0.07 | 11.0 | 1.8 | 1.40 | 0.52 | 0.32 | 0.37 | 0.01 | 6.8 |

[a] Estimated from the BioTrak OPC data.

[b] Likely to be biased high due to elevated dust levels.

**Table 2:** The median of the standard deviations for four freezing cycles on each drop for the normal freezing temperature in cooling-only experiments. Also the median of the freezing temperature range (difference between highest and lowest observed freezing temperature) for the individual drops is given for all samples.

| Sample | Median of standard deviation [K] | Minimum standard deviation [K] | Maximum standard deviation [K] | Median of freezing T range [K] | Min freezing T range [K] | Max freezing T range [K] |
|---|---|---|---|---|---|---|
| Marine dominated | 0.34 | 0.02 | 6.05 | 0.87 | 0.10 | 12.47 |
| Mineral dust influenced | 0.31 | 0.03 | 4.20 | 0.93 | 0.07 | 7.73 |
| Continental pristine | 0.25 | 0.02 | 3.25 | 0.53 | 0.07 | 6.80 |
| Continental polluted | 0.31 | 0.02 | 4.66 | 0.77 | 0.03 | 9.37 |
| Combustion dominated | 0.32 | 0.02 | 3.10 | 0.90 | 0.07 | 6.83 |
| Rural continental | 0.26 | 0.03 | 6.00 | 0.58 | 0.10 | 10.63 |

1610

**Table 3:** The averaged initial ice fractions for the samples at t = 0, and the observed average increase in ice fraction after 2- and 10-hours isothermal time. It should be noted that the experimental data for the isothermal time between 2 and 10 hours of exposure to constant conditions are much scarcer than the data for the first 2 hours since so few drops freeze then. This can also be seen in Figure 9. $\chi$ is the fractional increase in frozen fraction (FF) after 10 hours, also shown in Figure 10.

| Sample type | Initial FF | Change in FF (2 hours) | Change in FF (10 hours) | Unfrozen fraction after (2h) | Unfrozen Fraction after (10 h) | $\chi$ |
|---|---|---|---|---|---|---|
| Marine | 0.38 | 0.25 | 0.27 | 0.37 | 0.35 | 0.71 |
| Mineral dust influenced | 0.25 | 0.23 | 0.37 | 0.52 | 0.38 | 1.50 |
| Continental pristine | 0.37 | 0.15 | 0.26 | 0.48 | 0.37 | 0.70 |
| Continental polluted | 0.38 | 0.25 | 0.37 | 0.37 | 0.25 | 0.97 |
| Combustion dominated | 0.27 | 0.11 | 0.33 | 0.62 | 0.40 | 1.24 |
| Rural continental | 0.096 | 0.097 | 0.21 | 0.81 | 0.69 | 2.19 |

1620

**Table 4:** Empirically fitted parameters for Eq (4), $\hat{\tau}(t) = C_i t^{\alpha}$, with 95 % confidence bounds for the fitting parameters.

| Sample type | $C_i$ Fit | 95% conf. bound lower | upper | α Fit | 95% conf. bound lower | upper |
|---|---|---|---|---|---|---|
| Marine | 515 | 231 | 1152 | 0.125 | 0.018 | 0.233 |
| Mineral dust influenced | 307 | 107 | 875 | 0.266 | 0.126 | 0.406 |
| Continental pristine | 107 | 32 | 362 | 0.406 | 0.243 | 0.568 |
| Continental polluted | 173 | 67 | 446 | 0.266 | 0.140 | 0.392 |
| Combustion dominated | 243 | 81 | 734 | 0.358 | 0.211 | 0.503 |
| Rural continental | 172 | 57 | 521 | 0.438 | 0.290 | 0.585 |

**Table 5:** Temperatures with frozen fractions of 50% during constant cooling-only experiments. Also shown are maximum differences in in freezing temperature between hybrid cooling-isothermal and cooling-only experiments after 2 hrs.

| Sample type | Freezing temperature ( °C) | Maximum differences (K) in freezing temperature between hybrid cooling-isothermal and cooling-only experiments |
|---|---|---|
| Marine | -14.8 | 1.6 |
| Mineral dust influenced | -16.3 | 1.4 |
| Continental pristine | -14.5 | 0.7 |
| Continental polluted | -16.6 | 1.1 |
| Combustion dominated | -15.8 | 1.0 |
| Rural continental | -17.8 | 1.2 |

**Table 6**: Empirically fitted parameters of Eq (6), $\Delta T(t) = -A_i t^\beta$, with 95% confidence bounds for the fitting parameters.

| Sample type | $A_i$ Fit | 95% conf. bound | | $\beta$ Fit | 95% conf. bound | |
|---|---|---|---|---|---|---|
| | | lower | upper | | lower | upper |
| Marine | 0.376 | 0.328 | 0.425 | 0.261 | 0.245 | 0.277 |
| Mineral dust influenced | 0.240 | 0.148 | 0.332 | 0.323 | 0.280 | 0.367 |
| Continental pristine | 0.210 | 0.165 | 0.254 | 0.288 | 0.262 | 0.314 |
| Continental polluted | 0.575 | 0.502 | 0.648 | 0.217 | 0.201 | 0.233 |
| Combustion dominated | 0.131 | 0.082 | 0.180 | 0.362 | 0.320 | 0.405 |
| Rural continental | 0.267 | 0.147 | 0.386 | 0.319 | 0.269 | 0.368 |