# Peer review of "Time-dependence of Heterogeneous Ice Nucleation by Ambient Aerosols: Laboratory Observations and a Formulation for Models"

_Atmospheric Chemistry and Physics, 2021_

## Community Comment (CC1)

**Comment on acp-2021-830**

Gabor Vali

December 15, 2021

This work is a welcome addition to a rather sparse set of experiments in which freezing of a population of water drops is
observed at fixed temperatures. This comment refers principally to that part of the paper. The constant cooling and repeat
freezing data are of good quality and are well analyzed.

Experiments at fixed temperatures need to have large sample sizes in order to meaningfully determine the time dependence
of freezing and to separate the influences of the distribution of different INPs in the individual samples (drops) from the
stochastic time dependence of nucleation[1]. Previous constant-temperature experiments that can be used for the purpose are
those by Vonnegut (1948; Vt48), Vali and Stansbury (1966; VS66), Vali (1994; V94), Vali (2008; V08), Wright and Petters
(2013,WP13a) and Wright et al. (2013; WP13b). There is no quantitative comparison with these earlier works in acp-2021-830;
this comment is aimed at remedying that and to critique what is reported in the paper.

In all the experiments to be examined, sets of sample drops were cooled from above 0°C to the test temperature $T_s$, held at
that temperature for a period time and then cooled again until all sample drops were frozen. In Vt48, the sample was plunged
to the test temperature in an uncontrolled way. In subsequent works the rate of cooling was controlled[2].

The parameter best suited to discuss the results of the experiments is the rate of freezing, $R_T$, originally given as Eq. 3 in
V94, and defined as

$$R_T(t) = -\frac{1}{N_{uf}} * \frac{\delta N_{uf}}{\delta t} \tag{1}$$

where $N_{uf}$ is the number (or fraction) of the sample that is not frozen at time $t$. This function is a more explicit representation
of observations than the fraction frozen versus time curves, though some trends can also be seen qualitatively in those curves.

Freezing rates as functions of time are shown in Fig. 1 for four constant temperature data sets. Panel (a) shows the freezing
rate extracted from Fig. 5 of Vt48 with data from experiments with 64 water drops on a treated metal plate. Values of $R_T$ were
derived by reading numbers off the published graph and then using Eq. 1. Panel (b) is derived from Fig. 4 of Pound (1952)
which shows the percent of tin droplets remaining liquid as they are held in a dilatometer at various temperatures for up to 250
minutes. The dilatometer measures the change in volume associated with the solidification. The published graph of percent
unfrozen was used to read off changes over various time intervals and then Eq. 1 applied substituting percent unfrozen for
$N_{uf}$. This is not totally valid as the tin droplets were not uniform in size, but the focus here is on the temporal change of $R_T$
which will be relatively insensitive to this simplification. Panel (c) is based on the same data as Fig. 3 in V94, but for only one
temperature and not normalized to the rate of freezing at the moment cooling stopped. Panel (d) is based on data in WP13a
shown in their Fig. 5. The original data were kindly provided by the authors. Freezing rates were calculated using Eq 1 again
making the simplification that the dispersion of droplet sizes in the emulsion used in these experiments doesn't alter the points
to be made here.

The four data sets shown in Fig, 1 derive from rather diverse methods and are based on limited sample sizes. Yet, there are
common features worth noting:

1. The freezing rate decreases with time in all cases. This is significant because it contradicts the prediction of a stochastic
   model of freezing of a population of drops assumed to have the same INP content. With that model the rate would remain
   constant while the temperature is constant.

2. The freezing rate is a function of temperature, as seen in panels (a) and (b) of Fig. 1. This is an expression of the general
   trend for INP numbers to increase with decreasing temperature.
* * *
[1]Another approach is repeated freezing of the same drop and observing the time it takes for freezing to occur.

[2]To simplify this comment, it is assumed that the reader is familiar with the fact that freezing temperatures of drops in a set are spread over a range of
temperatures. Also for the sake of brevity, and in view of the relatively minor influence of the rate of cooling on freezing temperatures the impact of the history
of the sample prior to arriving at $T_s$ is ignored.

[Figure]

**Figure 1.** Freezing rate, $R_T(t)$ for four different data sets. (a) Vonnegut (1948), (b) Pound (1952), Vali (1994) and Wright and Petters (2013). See text for details.

3. The function best describing the time evolution of $R_T$ is not necessarily an exponential. A roughly exponential decrease was suggested in V94 from a limited data set and for a water sample that was shown in cooling experiments to have an exponential dependence of the INP concentration on temperature. The data shown in Fig. 1 is insufficient to judge to what extent power functions might be better in panels (a) and (d). There is no *a priori* reason to expect a simple algebraic relationship. Because of the lack of a quantitative theoretical explanation, any algebraic expression must be considered at this time a parameterization of the empirical data.

Results in Fig. 8 and Fig. 10 of acp-2021-830 support the first point in the list above but references to some "natural time scale" and to fast and slow nucleation are not explained. Point 2 is not tested with the current data as only one isothermal temperature was used for each sample.

Regarding point 3, acp-2021-830 employs nested equations requiring three parameters ($\Delta N_{ice,\infty}$, $C_i$ and $\alpha$) to describe the fraction of drops frozen as a function of time. It is unclear what physical model is there to justify this approach. It also may be noted that the overall trend in Fig. 10, neglecting differences among samples, appears to be a power function similar to two of the panels in Fig. 1. It does confuse matters somewhat, that those examples were obtained for polidisperse drop populations. Incidentally, the large range of three orders of magnitude in $R_T$ in Fig. 10 is somewhat surprising given what is seen in Fig. 8.

In general terms, the results in acp-2021-830 confirm that time is less important than temperature in heterogeneous freezing nucleation. The important issue is how to explain this (secondary) time dependence and to what extent the new data support

earlier conclusions. At this point only a qualitative evaluation seems possible but at that level the current results are compatible with the three points listed above. It is less clear to what extent the authors agree with this interpretation of their data, since their analyses are not well explained.

Briefly, the three points listed above summarize aspects of the view of heterogeneous freezing nucleation developed in VS66, V94 and WP2013b. In these papers the freezing rate $R_T$ is shown to be determined by two processes, one related to the stochastic time dependence of embryo growth, the other to the random distribution of INPs of different character in the sample drops. The first process leads to the site nucleation rate function with the characteristic temperature that anchors that function. The allocation of INPs of different characteristic temperatures in the sample drops is described by the nucleus spectrum[3].

The analyses given in ACP-2021-830 may be based on similar thoughts to those summarized in the preceding paragraph. If a closer agreement between the new data and those in earlier publication could be shown that would be a welcome reinforcement of the VS66/V94/WP2013b formulation. This is a crucial point to be clear about, as the interpretation of experiments as well as any effort to construct models of ice nucleation in complex systems like clouds or plants depend on this understanding.

The scheme presented in the paper for incorporating time dependence in cloud models follows the temperature shift approach suggested in V94 but without accounting for the rate of cooling. Regrettably, no comparison is offered with the TDFR parcel model of Vali and Snider (2015).

In all, the excellent device described in acp-2021-830 can certainly yield important inputs to studies of immersion freezing. Results in Section 3.1 show this; the presentation there suffers somewhat from the lack of quantitative error analyses. Only one test temperature in the isothermal experiments (Section 3.2) is a limitation but what data has been generated already is valuable and deserves a clearer presentation and more thoughtful analyses. The value of the parameterization in 3.2.2 could be evaluated by comparison with other, possibly simpler, solutions to the problem. Hopefully, these shortcomings will be corrected in a revised version of the paper and it will become clearer to what extent the results reinforce, or demand re-consideration, of earlier results.

(Dr. Markus Petters is thanked for data and for helpful discussion.)

**References:**

Pound, G. M.: Liquid and crystal nucleation, Ind. Eng. Chem., 44, 1278-1283, 1952.

Vali, G.: Freezing rate due to heterogeneous nucleation, J. Atmos. Sci., 51, 1843-1856, 10.1175/1520-0469, 1994.

Vali, G.: Repeatability and randomness in heterogeneous freezing nucleation, Atmospheric Chemistry and Physics, 8, 5017-5031, 10.5194/acp-8-5017-2008, 2008.

Vali, G., DeMott, P. J., Möhler, O., and Whale, T. F.: Technical Note: A proposal for ice nucleation terminology, Atmos. Chem. Phys., 15, 10263-10270, 10.5194/acp-15-10263-2015, 2015.

Vali, G., and Snider, J. R.: Time-dependent freezing rate parcel model, Atmos. Chem. Phys., 15, 2071-2079, 10.5194/acp-15-2071-2015, 2015.

Vali, G., and Stansbury, E. J.: Time dependent characteristics of the heterogeneous nucleation of ice, Can. J. Phys., 44, 477-502, 10.1139/p66-044, 1966.

Vonnegut, B., 1948: Variation with temperature of the nucleation rate of supercooled liquid tin and water drops. J. Colloid Sci., 3, 563-569.

Wright, T. P., and Petters, M. D.: The role of time in heterogeneous freezing nucleation, Journal of Geophysical Research: Atmospheres, 118, 3731-3743, 10.1002/jgrd.50365, 2013a.
* * *
[3]Definitions are given in Section 4.7.2 of Vali et al. (2015) for the site nucleation rate, and in Section 4.3 for the nucleus spectrum. The combined process is described in Section 4.7.3.

Wright, T. P., Petters, M. D., Hader, J. D., Morton, T., and Holder, A. L.: Minimal cooling rate dependence of ice nuclei activity in the immersion mode, Journal of Geophysical Research: Atmospheres, 118, 1-9, 10.1002/jgrd.50810, 2013b.

---

## Author Comment (AC3)

**Reply to Community Comment by Alexei Kiselev**

Kiselev: Dr. Gabor Vali has very recently attracted my attention to the excellent experimental data set on isothermal freezing of droplets containing ambient INPs, presented in the manuscript by Jonas Jakobsson, Vaughan Phillips, and Thomas Bjerring-Kristensen. I have read it with great interest. I fully agree with the authors that the isothermal freezing experiments are scarce and the data sets containing such experimental data are valuable.

Response:  We appreciate the comment here and thank Kiselev for the compliment.

Reviewer:  For this reason, I would like to draw the authors' attention to the manuscript we have published in 2016, where we have investigated the freezing behavior of several feldspar specimens in immersion freezing in a droplet freezing assay setup (Peckhaus et al., ACP 2016). Owing to the large number of droplets in our droplet freezing assay setup, we could observe freezing of several hundreds of nL-sized droplets at constant temperature for an hour. Our observations have, in general, confirmed the conclusions of this manuscript: in a simple system containing only one type of ice nucleating active site, the freezing follows a strict exponential pattern, whereas in a heterogeneous system featuring broad or even multimodal distribution of IN active sites, a steady decrease of freezing rate over time is observed. Interestingly, we could account for all observed effects (time dilation of freezing rate, freezing behavior at constant cooling rate, and cooling rate dependency) by using a consistent set of fit parameters within a CNT-based model equation framework (the so-called Soccer-Ball Model, SBM, Niedermeier et al., 2014 and 2015).

Response:  I struggle to see any observations published by Peckhaus et al. of a strict exponential decay of the liquid fraction for a simple system.   Figure 7 of their paper shows a linear decrease on a log-log plot of liquid fraction and time.  But an exponential decrease would need to display a linear decrease on a plot of log of liquid fraction vs time plotted linearly (ie. a straight line on a semilog plot).  Their Figure 7, for feldspar samples, shows what we see, a power law (liquid fraction proportional to time to the power of a negative constant) an exponential decay with a relaxation time that increases with time (time dilation).  The time for decrease by a certain fraction increases with time.

We agree that variability of amounts of INP material in each drop and of its nucleating efficiency of most active sites would cause such time dilation.

Reviewer:  Given the size of our sample and relatively high level of control over the experimental conditions, the authors of this manuscript might be interested in applying their parameterization to our experimental data set, which we will be happy to share. In any case, a mention of the (Peckhaus et al., 2016) in the introduction would make the overview of the previous research more complete.

Response:  Thank you for the idea.  Yes, this seems to be worth trying, perhaps in a future paper.

---

## Author Comment (AC4)

**Reply to Community Comment by Gabor Vali**

Vali: This work is a welcome addition to a rather sparse set of experiments in which freezing of a population of water drops is observed at fixed temperatures. This comment refers principally to that part of the paper. The constant cooling and repeat freezing data are of good quality and are well analyzed.

Response:  We thank Vali for this lucid comment.

Vali:  Experiments at fixed temperatures need to have large sample sizes in order to meaningfully determine the time dependence of freezing and to separate the influences of the distribution of different INPs in the individual samples (drops) from the stochastic time dependence of nucleation.

Response:  We used many hundreds of drops to study each sample.

Reviewer:  Previous constant-temperature experiments that can be used for the purpose are those by Vonnegut (1948; Vt48), Vali and Stansbury (1966; VS66), Vali (1994; V94), Vali (2008; V08), Wright and Petters (2013,WP13a) and Wright et al. (2013; WP13b). There is no quantitative comparison with these earlier works in acp-2021-830; this comment is aimed at remedying that and to critique what is reported in the paper.

Response:  During the isothermal phase, Vali (1994) observed a doubling of the number of frozen drops after 10 mins (-18 degC, drops on aluminium foil).  After 10 hrs, Wright and Petter (2013) saw an approximate quadrupling of the number of froze drops at constant temperature (-22 degC, 1% wt of ATD in each drop).

These results show stronger time-dependence than seen by our paper.  After 10 hrs we saw for our mineral-dust influenced sample, an increase by about 150%.

This underscores the importance of realism in choice of aerosol material if one wishes results to be representative of the real troposphere.  Such previous studies were seemingly oriented more towards theoretical perspectives on ice nucleation, hence their idealized design.

Reviewer:  In all the experiments to be examined, sets of sample drops were cooled from above $0\_C$ to the test temperature Ts, held at that temperature for a period time and then cooled again until all sample drops were frozen. In Vt48, the sample was plunged to the test temperature in an uncontrolled way. In subsequent works the rate of cooling was controlled.  The parameter best suited to discuss the results of the experiments is the rate of freezing, RT , originally given as Eq. 3 in  V94, and defined as

$R\_T (t) = (1/Nuf)\ dNuf /dt$  (1)

where Nuf is the number (or fraction) of the sample that is not frozen at time t. This function is a more explicit representation of observations than the fraction frozen versus time curves, though some trends can also be seen qualitatively in those curves.

Freezing rates as functions of time are shown in Fig. 1 for four constant temperature data sets. Panel (a) shows the freezing rate extracted from Fig. 5 of Vt48 with data from experiments with 64 water drops on a treated metal plate. Values of R_T were derived by reading numbers off the published graph and then using Eq. 1. Panel (b) is derived from Fig. 4 of Pound (1952) which shows the percent of tin droplets remaining liquid as they are held in a dilatometer at various temperatures for up to 250 minutes. The dilatometer measures the change in volume associated with the solidification. The published graph of percent unfrozen was used to read off changes over various time intervals and then Eq. 1 applied substituting percent unfrozen for Nuf . This is not totally valid as the tin droplets were not uniform in size, but the focus here is on the temporal change of R_T which will be relatively insensitive to this simplification. Panel (c) is based on the same data as Fig. 3 in V94, but for only one temperature and not normalized to the rate of freezing at the moment cooling stopped. Panel (d) is based on data in WP13a shown in their Fig. 5. The original data were kindly provided by the authors. Freezing rates were calculated using Eq 1 again making the simplification that the dispersion of droplet sizes in the emulsion used in these experiments doesn't alter the points to be made here.

The four data sets shown in Fig, 1 derive from rather diverse methods and are based on limited sample sizes. Yet, there are common features worth noting:

1.  The freezing rate decreases with time in all cases. This is significant because it contradicts the prediction of a stochastic model of freezing of a population of drops assumed to have the same INP content.  With that model the rate would remain constant while the temperature is constant.
2.  The freezing rate is a function of temperature, as seen in panels (a) and (b) of Fig. 1. This is an expression of the general trend for INP numbers to increase with decreasing temperature.
3.  The function best describing the time evolution of RT is not necessarily an exponential. A roughly exponential decrease was suggested in V94 from a limited data set and for a water sample that was shown in cooling experiments to have an exponential dependence of the INP concentration on temperature. The data shown in Fig. 1 is insufficient to judge to what extent power functions might be better in panels (a) and (d). There is no a priori reason to expect a simple algebraic relationship. Because of the lack of a quantitative theoretical explanation, any algebraic expression must be considered at this time a parameterization of the empirical data.

Results in Fig. 8 and Fig. 10 of acp-2021-830 support the first point in the list above but references to some "natural time scale" and to fast and slow nucleation are not explained. Point 2 is not tested with the current data as only one isothermal temperature was used for each sample.

Response:   We agree with the three points.

We agree that the fractional change in the number of unfrozen drops is the important quantity, since it is related to the probability of any drop freezing per unit time.    So we have included in the paper a new

Figure 11b to show this (R_T):

[Figure]

However, we think that some of the drops are irrelevant to the computation of R_T in an isothermal experiment, as they will never freeze at the isothermal temperature even at the longest conceivable times. So when we computer the number of unfrozen drops, we consider only those drops remaining that will freeze at the longest time (10 hrs). We reach the same conclusion that the fractional rate of change of the number of unfrozen drops falls somehow, perhaps by a power law.

Our paper observes the natural time-scale of ice nucleation, which is defined as the relaxation time in the exponential time factor (effectively the reciprocal of the fractional freezing rate).

It is reassuring to know from the Fig. 1 shown by Vali here that the time dilation of the freezing timescale during isothermal experiments (reducing fractional freezing rate) that we have seen is also present in the previously published studies.

If one could somehow ensure that each drop had the same INP composition and size and only one INP per drop, would one then see the stochastic hypothesis hold true with constancy of R_T ?

Response: Regarding point 3, acp-2021-830 employs nested equations requiring three parameters (_Nice;1, Ci and _) to describe the fraction of drops frozen as a function of time. It is unclear what physical model is there to justify this approach.

Response:   What we write is that "*the limited literature of observations show that the unfrozen fraction is often seen to decay exponentially with time (Bigg 1953ab, Vali 1994, PK97; Knopf et al. 2020). Consequently, from our isothermal measurements (Fig. 11), the time dependency effects were inferred by fitting the observations with this empirical isothermal formulation*:

$$N_{ice}(t^*) = N_{ice,0} + \Delta N_{ice,\infty}\left(1 - e^{-\frac{t^*}{\tau}}\right) \qquad\qquad (1)$$

".

This justification of our isothermal formulation (Eq (1)) suffices for the present paper.   We follow an empirical approach, so there is no need to justify it with any model of underlying physics.

Response:  It also may be noted that the overall trend in Fig. 10, neglecting differences among samples, appears to be a power function similar to two of the panels in Fig. 1. It does confuse matters somewhat, that those examples were obtained for polidisperse drop populations.

Incidentally, the large range of three orders of magnitude in RT in Fig. 10 is somewhat surprising given what is seen in Fig. 8.

Response:  What is shown in Figure 11a (previously Fig. 10) is evidence of time dilation of the natural time-scale of the observed freezing (the time-scale is the reciprocal of the fractional freezing rate).  That may be seen in Fig. 9 (previously Fig. 8) if it is appreciated that the time axis is plotted logarithmically.

Reviewer: In general terms, the results in acp-2021-830 confirm that time is less important than temperature in heterogeneous freezing nucleation. The important issue is how to explain this (secondary) time dependence and to what extent the new data support earlier conclusions. At this point only a qualitative evaluation seems possible but at that level the current results are compatible with the three points listed above. It is less clear to what extent the authors agree with this interpretation of their data, since their analyses are not well explained.

Response:  As found by Knopf et al. (2020), the evolution with time of the fractional rate of freezing observed in drop freezing experiments is governed by variability among drops of the amount of IN material per drop.  So it might be apt for the nucleation community to transition towards more realism of lab experiments, with perhaps isothermal observations of aerosol suspended in air that is somehow humidified so that there is only one INP per droplet (E.g. at the AIDA cloud chamber).

Reviewer:  Briefly, the three points listed above summarize aspects of the view of heterogeneous freezing nucleation developed in VS66, V94 and WP2013b. In these 5 papers the freezing rate RT is

shown to be determined by two processes, one related to the stochastic time dependence of embryo growth, the other to the random distribution of INPs of different character in the sample drops. The first process leads to the site nucleation rate function with the characteristic temperature that anchors that function.

The allocation of INPs of different characteristic temperatures in the sample drops is described by the nucleus spectrum. The analyses given in ACP-2021-830 may be based on similar thoughts to those summarized in the preceding paragraph. If a closer agreement between the new data and those in earlier publication could be shown that would be a welcome reinforcement of the VS66/V94/WP2013b formulation. This is a crucial point to be clear about, as the interpretation of experiments as well as any effort to construct models of ice nucleation in complex systems like clouds or plants depend on this understanding.

Response:  We have heeded the reviewer's suggestion to relate the present results to those from V94 and WP2013.  We now include a paragraph with comparison in the concluding section.

Reviewer: The scheme presented in the paper for incorporating time dependence in cloud models follows the temperature shift approach suggested in V94 but without accounting for the rate of cooling. Regrettably, no comparison is offered with the TDFR parcel model of Vali and Snider (2015).

Response:  We looked at the TDFR model of Vali and Snider (2015).   Their Eq (6) connotes a natural time-scale of the freezing (1/q) that is constant with time.  We get a drastic steady increase with time of this time-scale.  Thus, if we were to somehow apply the TDFR model to our observations we would see a discrepancy at long times presumably.

Reviewer: In all, the excellent device described in acp-2021-830 can certainly yield important inputs to studies of immersion freezing.

Results in Section 3.1 show this; the presentation there suffers somewhat from the lack of quantitative error analyses. Only one test temperature in the isothermal experiments (Section 3.2) is a limitation but what data has been generated already is valuable and deserves a clearer presentation and more thoughtful analyses. The value of the parameterization in 3.2.2 could be evaluated by comparison with other, possibly simpler, solutions to the problem. Hopefully, these shortcomings will be corrected in a revised version of the paper and it will become clearer to what extent the results reinforce, or demand re-consideration, of earlier results.

Response:  We thank the reviewer for the encouragement.

---

## Author Response (AR1)

**Summary of Author Responses to Reviewers**

In summary, we have heeded the warnings of both reviewers.  A chief concern was about relating the aerosol samples to major INP types in the model.  We have always been open about the uncertainties inherent in assuming particular correspondences.  To address this issue, we have now included more information characterizing the aerosol samples (lines 218-330), with fresh measurements in Sec. 2.2.2 with a new Table 2.  We also compare in more detail the INP concentrations seen in the samples with a new Figure 6 in Sec. 3.1.1 (lines 450-580) and provide the statistical tests required in a new Appendix A (lines 1005-1035).

Specifically, the physico-chemical aerosol properties presented in the reviewed manuscript were considered too limited by both reviewers. In the revised manuscript we present many new measurements of additional relevant aerosol properties (see the renewed Table 1, lines 233-330).  All aethalometer data from our sampling period had to be re-analysed, which has led to slightly different and more robust concentrations of black carbon. In addition, we have included (i) elemental concentrations inferred from particle induced X-ray emission studies of the remaining filter pieces, (ii) concentrations of supermicron primary biological particles when available and (iii) the chemical composition of the non-refractory particulate matter with diameters below 1 µm.

The added aerosol data support the initial classification of the samples and aid the discussion of potential differences in dominant types of ice nucleating particles within the sample ensemble.

Another concern was the need for more clarity in the explanations of the formulations applied.  We now have done that by defining symbols more carefully and by giving a list of symbols in a new Table B1 in the Appendix B.

We have revised the figures for more clarity (new Figures 6-11).  We have provided new Figures (e.g. Figures 5 and A1) to elucidate better the stochastic behavior, for instance with a new Figure 11b showing the fractional rate of change of the numbers of unfrozen drops.

With these changes, we hope that the paper is almost ready for publication.

**Reply to Reviewer 1**

**Author Response**

We are grateful to the reviewer for their comments on the manuscript, which have helped to improve it.

We have tried to improve the explanation of the mathematics with a new Appendix B with a list of symbols (lines 1045-1050).

**Point-by-Point Comments**

Reviewer:  This study analyses the time-dependence of freezing exhibited by ambient aerosol samples collected in southern Sweden. Constant cooling and isothermal experiments were performed with a recently developed cold-stage. The time dependence was found to be comparable to that seen in previous studies. A representation of time dependence for incorporation into schemes of heterogeneous ice nucleation, which currently omit time dependence, is proposed. The relevance of time-dependence in heterogeneous ice nucleation and its implementation in freezing schemes of cloud models is a timely and important topic. Yet, the study has major weaknesses that need to be addressed before publication. Moreover, the manuscript is not written carefully. The language and formulations are often imprecise and unclear, which hampers the understanding.

Response:  The reviewer's warning is well taken.

In response we have improved the language and formulations, with a more detailed definition of the mathematical symbols.  These are listed in a new Appendix B (lines 1045-1050).

Reviewer:  All the samples were collected at the Hyltemossa research station located in southern Sweden. The investigated samples were assigned to the following aerosol classes: marine dominated, mineral dust influenced, continental pristine, continental polluted, combustion dominated, and rural continental based on wind directions. In addition, BC content, PM1 and PM10 were determined. No attempt was made to further characterize the samples to confirm the assignments. The frozen fraction as a function of temperature shown in Fig. 6 and the INP concentration (Fig. 5) are all very similar and do not show the diversity found for INP samples collected at different locations. Also, the IN activity exhibited by the different samples are often opposite to expectations based on the class they were assigned to; e.g. the marine and the mineral dust samples were found to exhibit very similar INP concentrations, yet marine samples typically exhibit much lower INP concentrations than mineral dust samples.

Thus, the claim that the collected samples cover the major relevant INP classes needs to be abandoned, unless it were to be supported through chemical characterization (e.g. elemental analysis).

Response:  **We never claimed to cover all "the major relevant IN classes", so there is nothing to be abandoned.**

What we actually wrote in the reviewed manuscript was:

> *"Six ambient aerosol samples were collected representing aerosol conditions likely influenced by these types of INPs: marine, mineral dust, continental pristine, continental polluted, combustion-related and rural continental aerosol".*

> *"A representation of time dependence for incorporation into schemes of heterogeneous ice nucleation that currently omit time dependence is proposed".*

> *"In this study, we aimed at selecting samples likely to be dominated by different INP types at least of relevance to Northern Europe, and most likely of wider spatial-temporal relevance".*

> *"In the present study we present empirical data about the time dependence of heterogeneous ice nucleation for six ambient environmental aerosol samples. Ambient environmental samples, representing a variety of aerosol types expected to be dominated by certain INP species, were investigated. As they were natural samples, they must be assumed to contain a complex composition, where multiple INP species may be active".*

The language here has always been cautious and we never claimed that our samples definitely characterise **all** major types of INPs.

**The concentrations of INPs we report (the new Figure 6) for the marine sample are almost identical to the average concentrations reported for a number of Pacific Ocean samples (Mason et al., 2015) and very similar to concentrations reported by Si et al. (2018) for the Northern North Atlantic. In Arctic marine samples, lower INP concentrations have been reported (Irish et al., 2019). Overall, lower and higher INP concentrations than what we report have been observed in marine environments. So judging from the INP concentrations alone, there is nothing speaking against that sample potentially being dominated by marine INPs.   There is no evidence of any problem.**

**This is all discussed with fresh text (lines 450-552).**

It is likely that dust particles contribute significantly to the immersion freezing INP population at many different locations and in various environments where other types of INPs may be absent. Hence, dust particles may be the dominant immersion freezing INP type although present at low concentrations in remote pristine regions during some seasons. In our study, it is highly likely that a significant fraction of the PM in the dust dominated sample was comprised of Saharan dust, which had been transported for about a week in the atmosphere before collection in the boundary layer. Hence, we would not expect dust loadings or INP concentrations as high as could be expected in the dust plume and/or closer to the source region.

We have extended the supportive aerosol chemical analysis as suggested, and the results reported in the revised version of the manuscript do confirm elevated dust concentrations in the dust sample. Also,

the additional analysis confirmed that a significant fraction of the supermicron PM present in the marine sample could be associated with sea salt. In general, we find that the added aerosol properties support the sample classification.

**In summary, the reviewer's warning is well taken. During revision of the manuscript we have added more analysis of the aerosol composition for each sample** (lines 220-330) **and have included more cautious wording (e.g., line 18).**

Reviewer:  As it seems, the cold stage used in this study has not been described before. Therefore, its performance needs to be characterized properly. What is the precision and accuracy of the temperature measurement? Is there a freezing bias depending on the location of the drop on the substrate? What is the freezing curve of pure water?

Response:  We have not been able to work with our cold stage since November 2021, due to major construction work in our lab. Hence, we are limited to data already obtained with the cold stage in this context.

The temperature of the cold stage has been measured with external thermocouples and a very good agreement was observed relative to the cold stage read out. So we do not expect any significant temperature biases, in addition to the one described in more detail below.

All data obtained during constant cooling ramps of 2.0 K/min (> 10 000 freezing incidents) have been analysed. We compared the freezing temperature of the outer 50% of droplets to the inner 50%, regarding the placement of drops in the array on the cold stage. We found that the subset of droplets present closer to the outer edge of the array on average appeared to be exposed to temperatures 0.20 K higher than the centrally placed droplets for these cooling ramps and temperatures within the range -20 to -15°C. The INP concentrations presented in the revised version of the manuscript has been corrected for that offset, which had only a minor influence on the reported results. We do not have sufficient cold stage data to tell to which extent the temperature offset would be reduced if a slower cooling rate (e.g., of 1.0 K/min) were applied, but it is likely. Also, we consider it likely that a temperature off-set for the isothermal operation would be less than 0.20 K for the droplets closer to the edge. Hence, it is likely that a fraction of the droplets during isothermal experiments had temperatures 0.1 to 0.2 K higher than was we report. We consider such minor offsets acceptable for this type of experimental work.

**A fresh figure with freezing curves for ultra-pure water has been included in the revised manuscript. This new Figure 5 shows the frozen fraction observed for ultra-pure water during the constant cooling rate experiment (lines 441-443).  It confirms the accuracy of our device, as the freezing then occurs mainly at temperatures colder than about -25 degC with a freezing temperature for 50% of drops frozen colder than -30 degC.   This is near the homogeneous freezing temperature.**

Reviewer:  A time dependence is inferred without specifying whether it complies with the assumption of stochastic ice nucleation. The reason of the time dependence should be discussed. It should be analyzed to what degree the time dependence is indeed stochastic, i.e. stemming from identical INPs freezing at a

certain rate, or whether distinct nucleation sites exhibit gradual shifts or sudden jumps in freezing temperature as e.g. shown in Vali (2008), Wright and Petters (2013) or Kaufmann et al. (2017).

Response: "Stochastic" means that chance governs whether any active site on the solid aerosol particle nucleates ice so that there is a time-dependence of the population of such INPs. Our observations of time-dependence are consistent with that role of chance acting on active sites, each with a fixed freezing rate at any given temperature.

**The goal of our paper is to measure the overall time-dependence of ambient aerosol samples and to provide an empirical framework for representing time-dependence in cloud models. Our aim is not chiefly to explain the underlying reason for the time-dependence, which would probably require additional experiments.**

However, we can make some reasonable inferences about likely explanations. Regarding the freezing fraction spectrum among drops as a function of temperature during constant cooling experiments, a qualitatively similar spectrum was observed by Wright and Petters (2013, their Figure 5) as shown in our paper (our new Figure 6). They fitted their observed spectrum with a model based on a modified classical nucleation theory involving a statistical distribution of active sites of a wide range of efficiencies and multiple components. Moreover, observations by Wright and Petters (2013, their Figure 7) suggest an exponential decay of the freezing rate with time, which we also observe (our Figure 11). This all suggests that our results could be simulated by the same type of model based on multi-component multiple-component stochastic model of heterogeneous freezing nucleation, or even simply by assuming a variable amount of INP material per drop with a single component (Knopf et al. 2020).

It is difficult completely to rule out time-dependence arising from systematic changes in the freezing temperature of individual INPs over time. This could arise perhaps from morphological changes in the immersed solid material or immersed particles migrating with time towards the drop surface, where freezing is more likely.

**However, as can be observed from Fig. 7, there is very high reproducibility of the frozen fraction vs temperature for repeated constant cooling ramps carried out on the same droplet population. It is also evident from Fig. 8, that the vast majority of studied droplets froze at almost identical temperatures between repeated cooling ramps (see the red error-bars).**

The singular model would be a good approximation for our observations since the degree of time-dependence seen is quite limited. We were aware of potential changes in freezing patterns over time and after performed freezing cycles. The impact of exposure to isothermal conditions for 10 hours is merely a change of freezing temperature by 2 or 3 K for most drops.

We do observe some pronounced variability in freezing temperature for a minor subset of droplets. In future work, it would be interesting to check when and how that variability occurs. Is it a systematic change/jump in freezing temperature – which shows reproducibility – or did we observe more 'random' variability in such cases?

**In summary, our observations are qualitatively consistent with the stochastic theory from Marcolli et al. (2007) or Wright and Petters (2013) or Knopf et al. (2020). However, the degree of time-dependence we see is weaker. It is beyond the scope of our paper to analyse the theoretical reasons for the time-dependence we observed. Text has been added to clarify (lines 610-635).**

Reviewer: The relevant literature is not sufficiently taken into consideration in the introduction and in the discussion of the results (see specific comments).

Response: In the revised paper, the literature is now discussed more fully with inclusion of the references suggested in the reviews (lines 63-70, 103-108, and 126-130).

Reviewer: Line 41: the "many possible pathways" should be specified in the text.

Response: This has been done (lines 46-49).

Reviewer: Field and Heymsfield (2015) is not listed in the reference list. Moreover, more references should be given to support this statement, e.g. DeMott et al. (2010); Mülmenstädt et al. (2015).

Response: This has been done (line 43).

Reviewer: Lines 53-58: The discussion of the different types of atmospherically relevant INPs includes only two references. This is not sufficient.

Response: Many further references have been added (lines 60-70). Also, the description of INPs potentially associated with combustion emissions has been extended for support of the more extensive chemical aerosol analysis presented in the revised manuscript.

Reviewer: Lines 61–62: this statement is too general.

Response: We have added text to that sentence so as to explain what is meant (lines 73-76).

Reviewer: Line 93: here again, more than just one study should be referenced, e.g. add Vali (2008; 2014).

Response: Done, we have also cited Vali (1994) here (line 107).

Reviewer: There have not been many studies on temperature dependence but more than mentioned here. Older studies have been reviewed in Vali (2008) and Westbrook and Illingworth (2013). More recent laboratory studies have been performed by Herbert et al. (2014), Beydoun et al. (2016), Alpert and Knopf (2016), and Kaufmann et al. (2017). Moreover, there have also been recent modeling studies on the time dependence of immersion freezing, namely by Vali and Snider (2019) and Fan et al. (2019). These references should be included and discussed.

Response:    We presume here that the reviewer meant to write "time dependence" instead of "temperature dependence".   These references have been added as required (lines 126-130 and 155-158).

Reviewer:  Lines 178–179: Do you mean the particle size range between PM1 and PM10?

Response:   Yes, that follows from the definition of PM1 and PM10.

Reviewer:  Line 305: should it be "arise" instead of "rise"?

Response:  Yes. It is now corrected.

Reviewer:  Lines 314–326: Here, the INP concentrations are just compared with Fletcher (1962), without mentioning where the samples from Fletcher (1962) were collected. Typical INP concentrations of the claimed aerosol classes should be added and used for comparison.

Response:    The parameterization from Fletcher (1962) represents and average over different types of airmasses. The Fletcher parameterization is extremely often used as reference when these types of data are presented also in several recent publications. Hence, we find it useful to include here as a reference as well.

As stated above, we do not agree with the reviewer, that any given type of sample (e.g., marine or dust dominated) is associated with a specific INP concentration.  In reality, the associated INP concentrations of a given type of sample typically spans several orders of magnitude in concentration.

**Finally, we have included references (e.g., Si et al., Irish et al., Mason et al.) in the text to indicate that the observed concentrations of INPs are comparable to what has been reported in the literature (lines 495-500).**

Reviewer:  Lines 331–332: statistical tests should be performed to analyze whether the investigated samples are statistically different

Response:   Statistical tests have been performed as required for the freezing behaviour at -15 degC for all possible permutations of pairs (two-sample F tests) among the six samples.  Some pairs of samples statistically differ from each other; other pairs are similar.

A new Appendix A in the paper now summarises the results.

Furthermore, random errors are included in the new Figure 6 showing the INP spectra.

Reviewer:  Lines 347–349: do these variations in freezing temperature refer to the instrumental precision or characterize the samples?

Response:   These variations of 0.3 K refer to repeated experiments of freezing for any given population of drops.  The text now clarifies this (line 551), (see also lines 560, with a new Table 2).

Reviewer:  Lines 357–360: The values given here should become part of a table, in which also the largest and smallest standard deviations could be listed.

Response:  This has been done with a new Table 2.

Reviewer:  Line 360: Vali et al. (2008) is not in the reference list.

Response:   This has been added.

Reviewer: Lines 362–369: It should be stated which fraction of the droplets remains unfrozen, e.g. as an additional column in Table 4. The difference for most drops was stated to be "about 1–2 K". How was this value calculated?

Response:  This extra column in the new Table 3 (previously Table 4) has been included.

The statement that for most drops the difference was "about 1–2 K" is a comment on Figure 8 and not a formal calculation.  It should be regarded as data commentary (lines 566 and 578).

Reviewer:  Lines 376–378: The differences between individual isothermal experiments cannot be seen properly in Fig. 8, because all the isothermal experiments are shown as blue data points. Please choose different colors for different isothermal experiments. Moreover, the larger diversity between isothermal compared to constant cooling experiments should be discussed/explained.

Response:    Figure 9 (previously Fig. 8) has been replotted as required.

The diversity between the isothermal experiments is larger compared to the constant cooling rate experiments as the dependence on temperature is much larger than that on time. This makes the constant cooling rate experiments appear much more predictable and thus repeatable than the isothermal experiments, which show a more stochastic behavior and more diversity.

Text has been added to clarify (lines 587-588).

Reviewer:  Line 381 and throughout the manuscript: There seems to be a confusion between "freezing fraction" and "frozen fraction", which seem to be used synonymously. Yet, the frozen fraction means the fraction frozen at a given time, while the freezing fraction designates the fraction of drops that froze within a set time interval. As it seems, the authors mean "frozen fraction" most of the time.

Response:   The terminology is now changed everywhere as required.

Reviewer: Line 381–385, Fig. 9 and Table 4: The information provided in Fig. 9 is given more precisely as part of Table 4. This figure can therefore be removed. Moreover, the formula to calculate the data of Fig. 9 should be explicitly given.

Response: Thank you for the suggestion.

We are inclined to keep the figure as it is, since it encapsulates the essence of the measured time-dependence in one simple picture. The purpose of this Table 3 is to provide many more measurements for extra information. Table 3 includes much more information than what is shown in the new Figure 10.

The formula defining chi is included in the text as required (line 593).

Reviewer: Line 386 and Fig. 10: The analysis is unclear, also because freezing and frozen fraction are mixed up. The quantities in the formula should be properly defined. Did you really take the derivative or not just evaluate time intervals?

Response: The plotted fractional freezing rate (now Figure 11) was evaluated numerically with a finite difference scheme to approximate the derivative, using consecutive values of freezing fraction in each time series of measurements. The text is modified to clarify (line 604).

Reviewer: Lines 391–392: What is meant in this sentence by more and less active INPs? Typically, the ice nucleation rate of an INP increases with decreasing temperature. Yet, this sentence does not mention any temperature dependence and seems to imply that there are fast and slow nucleating INPs independent of temperature. The concept of slow and fast INPs needs to be clarified.

Response: We now add text to clarify (lines 624-627).

By "active", we really meant more efficient (a higher nucleation efficiency).

We now realise it may have been better to plot the fractional change of the unfrozen fraction for analysis of the stochasticity. So we have added a new Figure 11b showing this.

This sentence is about isothermal experiments, so the temperature dependence of the efficiency of INPs is not so relevant here. Anyway we now include mention of this temperature and also clarify the concept of slow and fast INPs.

Reviewer: Lines 394–405: Here, a time dependence of INP activation is proposed without taking the temperature dependence into account. Yet, models need to combine both, and cover also situations of temperature fluctuations:

Response: We agree that temperature dependence is crucially important generally.

But this Section 3.2.1 is entitled "**Isothermal time series and relaxation time**". This is not the point in the paper where we consider temperature-dependence.

The model (Eq (1)) provided at this specific point in the paper is not intended as a model of ice nucleation generally for all temperatures. Of course, later in the paper we will show how to create such a model. Perhaps the word "model" has connotations in the experimental community that we never intended.

**To avoid confusion, this Eq (1) is renamed as an "empirical isothermal formulation" (instead of "model"), (line 641), to convey the fact that it treats the measurements in isothermal experiments at a single fixed temperature. There are no temperature fluctuations of any significance during each isothermal experiment.**

Reviewer: e.g., what would be the time dependence of freezing in an air parcel that was supercooled by 1 K before reaching the isothermal period? Vali (1994) found that this depletes the INPs that are active at the isothermal temperature. The proposed approach should also be discussed in view of the findings of Vali and Snider (2015).

Response: See the preceding comment. We will propose a general model of atmospheric ice nucleation with time- and temperature-dependence later in the paper, not here.

**Here, with Eq (1), we are only trying to understand the observations at constant temperature.**

Reviewer: Line 419–421: Again, this argumentation insinuates that less active sites activate more slowly than the more active ones.

Response: That is what Herbert et al. (2014) found: INPs that are less efficient display more time-dependence and activate more slowly.

Reviewer: Yet, the nucleation rates of sites are highly temperature dependent.

Response: Yes, absolutely. We never denied that. Right here, with Eq (1), our focus is not on temperature-dependence. This section 3.2.1 is about the isothermal experiments.

Reviewer: Line 440, Eq. 4: How is the time dependence of the INP concentration calculated?

Response: There was a typing error in Eq (4). On the RHS the only time-dependence represented is via DeltaT, the temperature shift, which is informed by our lab data empirically.

Reviewer How can the temperature shift approach be combined with temperature fluctuations observed in air parcels?

Response: Those temperature fluctuations in air parcels are predicted by any cloud model that resolves the parcels. Our approach is designed to be implemented in the IN scheme of such a cloud model.

The dependence of IN activity on the ambient air temperature is already represented in the unmodified empirical parameterization (Phillips *et al.* 2008, 2013). The time-dependence of INPs is represented as a higher order perturbation, with the time-dependence of the temperature-shift being assumed to be the same at all temperatures.

The lack of sensitivity of this time-dependence of the temperature shift with respect to temperature is suggested by results from Herbert et al. (2014, their Figure 2).

Reviewer: Line 467–468: This listing of temperature information should be put in a table.

Response: Done (Table 5, line 716).

Reviewer: Lines 478–480: This sentence needs to be formulated better.

Response: The grammar has been improved (line 726).

Reviewer:  Line 486: Again, a table would be more appropriate.

Response: Done (Table 5, line 734).

Reviewer:  Lines 488–489: How did you establish the consistency?

Response: We were referring only to the qualitative pattern of the degree of time-dependence among the six samples. The text is now clarified (line 736).

Reviewer:  Lines 493–495: this sentence should be formulated better.

Response:  Done (line 740).  This sentence was intended to explain the previous sentence.

Reviewer:  Lines 499–500: this sentence should be formulated better.

Response:  Done (line 746).

Reviewer:  Line 507: it is Budke and Koop, 2015.

Response:  Corrected.

Reviewer: Line 524–525: A further explanation of the time dependence would be non-stochastic changes in IN activity that have been found e.g. in refreeze experiments by Vali (2008), Wright and

Petters (2013) or Kaufmann et al. (2017). An estimate of the contribution of such changes compared to stochastic freezing would be interesting.

Response:  We do not completely agree with this notion.

In reality, repeatability of measured freezing spectra before and after the isothermal experiments, as reported in the paper, implies that there is little systematic shift in freezing temperatures of individual INPs.  We do not see signs of such non-stochastic changes in IN activity in our data for the vast majority of drops we observe.

The weaker repeatability of freezing for a minority of drops requires further investigation (see the few wider error-bars in the new Fig. 8).

**Our observation here further reinforces the need for the experimental community to study ambient aerosol samples, instead of surrogate samples from the ground or artificially manufactured aerosol material, when seeking conclusions about atmospheric ice nucleation.**

**A change of focus is needed we believe in the community.  We have added text to convey this in the concluding section (lines 931 and 967).**

Reviewer:  Lines 532–537: Here, the possibility of several INPs present in the same drop is discussed as a risk. Yet, it is a fact that there are multiple INPs present in microliter drops, albeit with different characteristic freezing temperatures. Also in cooling experiments, several INPs compete in ice nucleation. To judge how many INPs have similar characteristic ice nucleation temperatures and might compete within a drop, samples with different degrees of dilution should be compared. The authors should consider performing experiments with more diluted samples for comparison.

Response:  The reviewer is correct in this comment, and we have considered experiments with diluted samples to further investigate this phenomenon. Unfortunately, under the scope of this study we were limited both in experimental time and in the amount of available sample to perform these experiments. Thus, we have prioritized to keep the measurement procedure, including sample preparation, consistent for all samples in this study.

Also, as stated in lines 787-790, we have made calculations which imply that further dilution of the samples may have a significant impact on the number of observed freezing events during the isothermal phase of the experiments. This could potentially further weaken the counting statistics which is already a challenging.

**Studying more dilute samples would involve isothermal experiments at colder temperatures with the potential to investigate the INPs active at lower temperatures. With our apparatus – we may not be able to carry out isothermal experiments at much lower temperatures – before the background may start to bias observations.   Also, these isothermal experiments are extremely time consuming – already representing months of full time lab-work.**

Reviewer: Lines 541–552: Here again the temperature range of activity needs to be specified. This discussion does not make sense without specifying the temperature.

Response:   Done (lines 787-790).

Reviewer: Lines 549–552: CCN are mostly liquid and do not contain any INP. Thus, having several INPs in one cloud droplet is highly unlikely.

Response:  We agree.  The sentence is now deleted.

Reviewer: Lines 564–644: this needs to be explained better.  Lines 574–590: The use of Q needs to be explained better.

Response:  This has been done (lines 830, 840-861).  The definition of "cold cloud" is delineated precisely.

We have re-written some of the maths so as to make clearer the numerical technique for including the time-dependence in a cloud model.  We have now included a new Appendix B with a list of symbols for extra clarity.

Reviewer: Line 714: Knopf et al. (2021) has been published in the meantime.

Response:  This change is now made.

Reviewer: Line 831: do you mean "quartz" instead of "quarts"?

Response:  Yes, this is now corrected.

Reviewer: Figure 5: The formula that was used to calculate the INP concentrations should be stated or referenced. The different aerosol classes exhibit quite similar INP concentrations as a function of temperature. Therefore, statistical tests need to be performed to test whether the aerosol classes are statistically different. Moreover, each droplet population could be shown as separate line in Fig. 5, as it is done in the freezing spectra in Fig. 6, to judge visually whether the different aerosol classes are different. Finally, the INP concentrations should be compared with typical INP concentrations of the aerosol classes they should represent.

Response:  The INP concentrations were obtained as described by Vali (1971), and that has been specified in the text.  The new Figure 6 (previously figure 5) showing the INP spectra easily becomes hard to read. In the revised manuscript, we have only made use of the first constant cooling ramps for

each of the five different droplet populations investigated for each sample (in total 500 droplets). The random errors representing the variability between these data have been included in the revised figure.

As described in more detail above statistical tests have been performed and we do not agree to the idea that a well-defined INP concentration is linked specific aerosol types. The limited data set we present in the revised manuscript with several additional aerosol properties included clearly shows that similar INP spectra can be observed for significantly different aerosol classes.

Also, we do not find evidence supporting the reviewer's point of view here in the literature, as described in more detail above.

Reviewer: Figure 6: The differences in frozen fraction between aerosol classes are small and difficult to judge the way the frozen fraction is plotted. The figure could be improved by narrowing the temperature range to -5°C to - 25°C (as it is done in Fig. 13) and by adding gridlines to the panels.

Response:  This is now done in the new Figure 7.

Reviewer: Figure 7: It might be helpful to add dots for the drops that did not freeze during the isothermal experiments. They could be put at the bottom of the panel (at -25°C).

Response:   It is not possible to do this since the temperatures at which those drops froze were not observed.  They generally did not freeze at -25 degC.

Line 884: what is meant by "minimum of 4 cooling cycles"?

Response:  Some of the experiments involved multiple cooling and heating cycles.  For the new Figure 8 we used at least 4 such cycles per sample.

Reviewer: Figure 8: These plots are again difficult to read. The frozen fraction for each experiment should increase continuously but the blue data points just scatter, most probably because they stem from isothermal experiments performed with different droplet populations. In this case, they should be shown in different colors or symbols so that different experiments can be discriminated. Were the data points taken at defined time intervals?

 Response:  Corrected in the new Figure 9 with different shades of dots for different drop populations.

Reviewer: Figure 9: the information provided by this figure is also given in Table 4. It can be removed.

Response:  We prefer to keep this new Figure 10 as it encapsulates the essence of the time-dependence in a vivid way.

Reviewer:  Table 1: The line numbers are shifted to the right.

Response:  Corrected.

Line 982: what do you mean by "much more limited"?

Response:  We meant "much scarcer".  We have now clarified the caption of the new Table 3 to explain this.

**Replies to Reviewer 2**

**Author response**

We are grateful to the reviewer for their effort in scrutinizing the manuscript.

We have included the extra statistical analysis required with a new Appendix A.  We now characterize the chemical composition of the aerosol samples in more detail (lines 218-326), justifying the classification we have given.  Table 1 and Sec. 2.2.2 have been greatly extended.

**Point-by-point Responses**

Reviewer:  This is quite a nice study, using a limited number of samples to study the time dependence of ambient ice nucleating particles freezing in the immersion freezing mode. In contrast to what I read in another review, I find the details about the experimental device (LUCS) and methods to be very good (and the authors responsible for it are to be lauded). The writing is also fairly clear, excepting poor introduction/definition of terms used in equations. The results demonstrate a relatively weak time dependence to freezing that is nevertheless consistent with prior studies using soil samples and cloud water. The consequent impact can be described by a temperature adjustment of say 2K in order to describe freezing at longer time scales.

Response:   We are grateful to the reviewer for the encouragement and we agree that the time-dependence observed is rather limited.

We have improved the definition of the maths symbols to make the paper more accessible to the modeling community.  There is now a list of all maths symbols in a new Appendix B.

Reviewer: One does wonder to what extent temperature control of the drops and where an INP may be floating in individual drops may influence these results. This is not discussed.

Response:  These are key questions and we now include discussion of possible systematic reasons for time-dependence (lines 541-547).

However, as we point out in the fresh text, the high degree of repeatability of the freezing temperature of each drop indicates that such systematic reasons for time-dependence are not significant.

Reviewer: In any case, the corrections in comparison to very short time scales range up to at most about a factor of 2. It is interesting that this is well within the bounds of the agreement of many immersion freezing methods when compared together. This is not spoken about either, but should be mentioned, the reason being that it emphasizes the utility of such measurements, regardless of whether used in a deterministic manner or with an approach as suggested here to describe the modest time dependence.

Response:  Yes, we agree that the degree of time-dependence we detect is so limited that its effect, when represented in cloud models by adapting IN schemes derived from field probes (e.g. CFDC), may be dwarfed by the instrumental errors of those probes.

**We agree that our study confirms the utility of measurements of IN with the CFDC and other similar field probes. We add text to stress this with a new paragraph in the concluding section (lines 942-950).**

Reviewer: While much effort is expended on analyzing cooling ramps and isothermal data on six samples, the least convincing aspect of the study is that these six cases can be clearly identified and taken as sufficiently representative and attributable to the types of aerosols identified for comparison. There are reasons that there should be variability amongst those types, and season could matter for different types as well. I recognize that numerous caveats were added in regard to the inability to know INP composition, but they are ultimately ignored in fashioning a parameterization that differs for the different types.

Response: We agree that this is a limitation of our study. However, there are reasons for believing that there is little effect on the time-dependent temperature shift arising from the type of INP composition. Herbert et al. (2014) found that this temperature shift was invariant when comparing contrasting INP types with high and low nucleation efficiencies. This invariance was explained theoretically.

If the degree of time-dependence is not dramatically sensitive to INP type (e.g. Figure 10), then there is no problem arising from uncertainty in the assumptions of which INP type are represented by which of our aerosol samples when the scheme is applied in a model.

**In this revision of the paper we have substantially extended the amount of presented physico-chemical aerosol properties in the revised Table 1 (lines 220-330). So there is more evidence provided now for the contrasting aerosol conditions represented by sampling with much extra text (especially in Sec. 2.2.2).**

Reviewer: Consequently, in suggesting that these results could be used as representative of INP types present in the noted aerosol scenarios (e.g., mineral, or organics) moving forward, when in fact the differences between them are modest (note Fig. 5 and Fig. 10, with insignificant differences apparent), is questionable. In reality, it seems unnecessary, unless one is only intent on using the referenced parameterizations instead of simply pointing out how deeper insights could be gained in the future using these methods in places where certain aerosol scenarios are clearly more dominant. This is not meant as a severe judgment on a study that has been needed for a long time.

Response: Well, we think there are some differences in the degree of time-dependence between the samples. "Time will tell" which INP types these differences reflect, future research will reveal this.

Reviewer: Needed and useful, especially for pointing out that corrections to INP data for time dependence is small, and results do not change greatly in repeated experiments, challenging some other recent studies (not noted, but oddly referenced at one point for the exponential decay of freezing rates – which those authors seem to attribute to experimental artifacts) that suggest that immersion freezing nucleation is largely purely stochastic for ambient INPs. It should be emphasized more that the present results appear to reject that hypothesis.

Response: There seems to be the potential for some confusion here about terminology.

If by "stochastic" one means that the fractional freezing rate of unfrozen drops remains constant with time such that the unfrozen fraction declines exponentially, then yes we disagree with that. But also Knopf et al. (2020) and Wright and Petter (2013) failed to observe such "pure" simple stochasticity.

If by "stochastic" one means that there is a wide PDF of nucleating efficiencies and of amounts of immersed solid material among the INPs of drops (e.g. Marcolli et al. 2007; Knopf et al. 2020), then we find qualitative agreement with that perspective.

By the way, we have more closely inspected our own published results and those of Knopf et al. and Wright and Petters, and we now find that the fractional freezing rate declines steadily, but not exactly exponentially over a constant relaxation time-scale. The decline is only exponential in the sense of a relaxation time-scale that dilates with time. We are more cautious the use of the term "exponential" now.

Reviewer: One other factor that I felt needed to be brought out in discussing Westbrook and Illingworth is the extreme population (extraordinarily high INP concentrations) required by that study to exist for their hypothesis of long freezing time constants to explain ice formation in clouds. Considering all other existing measurements of INP concentrations in the ambient atmosphere, and results such as presented in this paper on time dependence of freezing, the numbers required by that conjecture are not within the realm of possibility. I kept expecting the discussion to come back to this point, but clearly the authors have in mind to do full model simulations to invalidate the earlier hypothesis. That is a bit disappointing, because it leaves the readers hanging. In the end, the study is demonstrative of what could be done, with great effort obviously, if many more cases are identified or if done in environments that are more clearly dominated by certain INP types. I have an assortment of related and other specific comments added to this, which I do below in order of appearance. My recommendation is that this paper needs revision in places before being accepted for publication.

Response: **We agree with the apparent inconsistency with the results from Westbrook and Illingworth (2013) and have added text to explain (lines 942-950).**

The processes occurring in natural long-lived layer-clouds are complex. There are fluctuations of temperature and humidity from in-cloud turbulence and there may be secondary ice production that may amplify or damp the effects from extra activity of INPs.

There may have been cells of weak embedded convection with outflow feeding the layer-cloud observed by Westbrook and Illingworth. Such ascent may have replenished the INPs continuously. They report there were some such cells present away from their target area, but it is difficult to see how they could prove that the cloudy region they studied were unaffected by these far upshear.

**So it is difficult a priori to be completely certain whether the Westbrook-Illingworth hypothesis is inconsistent with our results. Another paper may investigate this.**

Reviewer: **Abstract**

Reviewer:  Line 13: It should read "six" ambient samples, to be explicit.

Response:  Done.

Reviewer: **Introduction**

Line 53: The first ice in any mixed phase cloud does not have to be from activation of INPs if sedimentation occurs from higher levels that may reflect homogeneous freezing conditions.

Response:  Agreed.  We have qualified what we wrote and now say:  *"The first ice in any mixed phase cloud is from activation of INPs, if its top is below the level of homogeneous-freezing (about -36 $^o$C depending on drop size; Pruppacher and Klett 1997)"*.

Reviewer: Lines 61: Only spot I saw where ice nuclei is used in preference to ice nucleating particles.

Response:  Corrected.

Reviewer: Lines 72-73: There is a fine point here that is not stated with regard to Westbrook and Illingworth's argument. This is that the action of a stochastic process over many hours would require an INP population unlike any ever measured. It already seems a nonstarter, but this study provides insight.

Response:  **We now mention this with new text (lines 942-950).**

The purpose of the paper is not to prove or disprove the Westbrook and Illingworth argument.  In view of the complexity of cloud-microphysical processes, potentially with secondary ice production (sublimational breakup), a detailed simulation of such a cloud is needed to do that.

Since they claimed the cloud was precipitating, then secondary ice production is possible.   What they wrote is "*The only established ice multiplication mechanism is rime-splintering which occurs exclusively between −3 and −8 ◦C (Pruppacher and Klett, 1997). Since our supercooled cloud did not span this temperature range, and since riming of the crystals themselves was minimal …, we do not believe that multiplication occurred in this cloud layer.*"

We now know that ice multiplication by ice-ice collisions or by fragmentation of freezing drizzle or by sublimational breakup is possible.  Rime-splintering is not the only possibility.

Since their cloud was observed over land and there were efforts to exclude embedded convection from the analysis, there might have been undetected pockets of ascent that drove turbulence mixing fresh IN into the cloud.

**There are many possible explanations other than time-dependence for the Westbrook and Illingworth observations of the long-lived layer-cloud.  Theirs was only a hypothesis and it awaits investigation with a detailed model.  Our paper provides a tool enabling this.**

Reviewer: Lines 93-94: A reference seems appropriate to support this point

Response:  References are now included (line 114).

Reviewer: Line 115: This is a curious reference for a paper that ultimately finds results in complete disagreement with single parameter CNT. Is it meant to point out that this is the case for certain INPs, such as illite?

Response:  Knopf *et al.* (2020) write that "*We demonstrate that IF* [immersion freezing] *can be consistently described by a stochastic nucleation process accounting for uncertainties in the INP surface area*" immersed in each drop.   They get adequate agreement in isothermal experiments with illite for up to half an hour or so.

We have clarified the citation in the text (lines 140-142).

Reviewer: Line 130: "…there is an inevitable cost from lack of identification of the precise chemical species initiating the ice in observed samples." **I appreciated these caveats, so then I wondered why the selected samples were not treated only as examples, rather than suggesting they are meaningfully representative of specific aerosol types.** There are ways to get at INP composition, even via immersion freezing methods, they simply are not used herein (see below).

Response: The reason that we do not treat our samples merely as randomly selected "examples" is that the goal of the paper is to provide a way to include time-dependence in the INP schemes of cloud models that are based on field probe measurements.  That is why we took great care to select the samples in such as way as to typify the airmass types as representatively as we could.

**We initially decided which airmass types we would target, then we selected a subset of measurement days that conformed in terms of physico-chemistry and back-trajectories with the corresponding classification criteria of each target.**

Reviewer: Lines 159-160: I am curious about the selection of filter pore size. I understand that larger pores allow high flow. Was face velocity and collection efficiency considered to estimate if there were undercollection of particles at 0.4 microns and smaller?

Response: We used the minimal pore size that was deemed possible to achieve 24 h sampling.

Studies of sampling efficiency have been carried out by others. The types of filters we used generally collect >90% of the particles present in the mobility size range from 20 to 300 nm - often with a collection efficiency >95% by number (Zikova et al., 2015). Hence, we cannot exclude that a tiny fraction of small INPs could had made it through the filter pores during sampling. However, the aim of our study was not to report accurate ambient INP concentrations, so we do not consider this an issue with respect to any of the main results reported.

This is now clarified in the text (line 203).

Reviewer: Line 161: What does it literally mean that not all filters were able to achieve a full 24-hour sampling? This is an unusual statement. The pump stopped because of overloading of the filter? The flow rate changed and you did not record it over time to get an accurate volume? If flow rates were not recorded, then this should be stated as an uncertainty for INP concentrations.

Response: The flowrates and sampled volume were recorded in 5 min intervals by the sequential sampler. The sequential sampler stopped automatically when the pressure drop exceeded the instruments capacity to maintain the set flowrate of 1 $m^3$/h. So yes, in a sense, the instrument stopped because of overloading in some cases. But this was quite rare.

Reviewer: Line 168: There is a difference between marking the samples to reflect different aerosol types and what will dominate as INPs, right? Sometimes the dominant composition is irrelevant if one particular type acts with higher efficiency. I think you aimed to select episodes that represented potentially different dominant aerosol types, assuming that these might reflect different abundances of INPs of different types. Ideally, you need a single type that is not influenced by trace amounts of another type, but there is literature to show that a little mineral dust sometimes overwhelms a marine INP population. Hence, the approach has a great deal of uncertainty associated with it. This of course is the nature of ambient sampling, and why some attempt to parse out influences of the different aerosol types present through more detailed approaches.

Response: We fully agree to those considerations.

We find it highly useful to classify the samples based on their physico-chemical aerosol characteristics – which are presented in more detail in the revised manuscript (lines 220-320). We can tell, that the relative ratios between potential ambient INPs vary significantly within the sample ensemble. We believe that we already have made it clear, that in our study, it is not possible to directly identify the types of active INPs.

However, it is highly unlikely that the active INP populations were identical between these samples, which is discussed in more detail in the revised manuscript (a new Appendix A compares the samples statistically for ice nucleating properties).

Reviewer: Line 174: You need to say more about how the HYSPLIT model was set up, and it should be referenced appropriately. I especially did not understand why the trajectories were set to end at 500 m, instead of somewhere closer to the surface site. Did you test different levels for this end point location?

Response: Additional information about the HYSPLIT operation has been included in the revised manuscript (lines 227-232, and a new Figure 1).

It is evident from the aerosol data presented in Table 1, that regional to long-range atmospheric transport has a huge impact on the aerosol properties at the Hyltemossa site. It is well known that friction in the very lowest parts of the planetary boundary layer (PBL) inhibits efficient horizontal transport. Horizontal transportation is more pronounced at somewhat higher altitudes in the PBL. Hence, a location 500 m above ground level is very often chosen for airmass back-trajectory modeling related to ground based aerosol measurements (e.g. Waked et al., (2018) and references therein).

It cannot be excluded that atmospheric circulation in the free troposphere can be of relevance to some aerosol parameters detected close to the ground level. Therefore we have also analysed airmass back trajectories arriving over Hyltemossa at an altitude of 2000 m AGL (above ground level).

In the revised Figure 1, we now present HYSPLIT 120h air mass back trajectories arriving at Hyltemossa 50 m (dashed lines) and 2000 m (dotted lines) above the ground. As can be observed from this Figure, the atmospheric circulation did not appear to vary greatly with altitude in the lower troposphere during the sampling periods.

Reviewer: Lines 188-189: But can you say that soil dust does not dominate also in the "combustion-dominated" sample, or any particular continental sample for that matter? You are a bit blind without knowing anything about the nature of the INPs contained in the air at any time.

Response: As stated further above, it was not possible for us to identify specific active INPs in these samples.

In the revised manuscript (lines 220-320), we present more extensive physico-chemical aerosol properties of more direct relevance to the potential INP population. Based on those supportive measurements, it appears highly likely that the relative abundance of different potential INP classes vary significantly between the samples. That is discussed in more detail in the revised version of the manuscript.

Reviewer: Section 2.2.2 overall: I will say that I otherwise appreciated the honesty and accuracy in statements made in this section about how certain (not very) one could be about the assumed total aerosol composition as representing INPs. Then why title it "Sample classification according to likely dominant composition of INPs"? Again, you are referring to what you think is the dominant aerosol type. There is no guarantee that the total aerosol type abundance will be reflected by a dominant INP of that type. It depends on individual efficiencies and what all types are there, which I think the authors understand. I suggest that in the future it could be beneficial to analyze for general INP types using methods in the literature (e.g., Testa et al., 2021, J. Geophys. Res, doi: 10.1029/2021JD035186). There are ways to get at inorganics that would include minerals and black carbon, for example.

Response: We are grateful to the reviewer for this excellent suggestion for future work and we now cite this reference by Testa et al. (2021) at the end of the concluding section (line 968).

We have included data related to minerals, BC, organics and supermicron primary biological particles when available in the revised version of Table 1. We have modified the discussion about potential INPs accordingly.

Reviewer: Section 2.2.2 also: Have you considered testing your assumption using aerosol reanalyses, such as MERRA-2?

Response: We thank the reviewer for this helpful suggestion and will use MERRA-2 in any similar future observations.

Reviewer: Section 2.2.3: What is a sterile cryogenic vial? That is, what do you mean by sterile? Was it tested for INPs released by pure water?

Response: The sterile cryogenic vials are small, sterilized polypropylene vials designed for the storage of biological material, human or animal cells, etc. at temperatures down to $-196$ °C. They are delivered sterilized by gamma radiation and are certified RNase-, DNase-, pyrogen- and DNA-free. The vials are flushed with ultrapure water before use and never opened outside of the laminar airflow benches. The vials were used during the "ultra clean water"- background measurements that are now included in the manuscript as Figure 5, and are thus included in these background measurements. (The vials are supplied by VWR, product number: 479-1237).

Reviewer: Lines 306-308: No freezing, meaning zero wells frozen? In general, I felt that the testing of water, field blank filters rinsed in water, and any other handling protocols need a little better documentation, especially for temperature ramps. Surely there is a background in the device. There were no frozen wells for DI in similar ramps, nor for the field blanks?

Response: We have included a new figure (Figure 5) in the manuscript to show the instrument freezing background for the instrument of ultrapure water, using the same procedure as the constant cooling rate experiments, including the transfer of water "sample" by the sterile cryogenic vials. During the experiments mentioned in Sec. 2.4.2 no freezing events were observed during 2h experiments at the isothermal temperatures (see also lines 432-437).

Reviewer: Lines 319-326: These generalizations are fine. At such a sampling site, even these characterized types must have seasonality, no? Perhaps in the conclusions you should note that using these 6 samples to characterize different source types might be a stretch until an annual cycle is explored or means are derived to more carefully distinguish influences and assured impacts on INPs.

Response: Yes, and that discussion has been modified in the revised manuscript accordingly. There is now a reference to Schneider et al. (2021) so as to depict seasonality of aerosol loadings (line 471).

Reviewer: Lines 331-332: Given this, I do not think that you can make the statement that it is "highly likely" that these six identified types differ significantly. You have not proven that. They all look quite similar within some bounds (again, differences in both Fig. 5 and Fig. 10 are minimal).

Response: We have addressed this issue in Appendix A with statistical tests for the difference in active INPs at -15 degC. None of the six samples are perfectly unique, each is statistically indistinguishable from at least one other sample. But most pairs of samples are statistically different from each other.

Reviewer: Are they representative of INP in general for the region? That seems likely. If you have the aerosol data and can make such calculations, could you not normalize all of these events (except the dust one where there is no data) by total aerosol surface area to see if that separates them at all? I understand that what you would want is speciated surface area, but total could be informative.

Response: Additional aerosol data are presented in the revised manuscript (lines 220-330, a new Table 1), and the discussion of likely INPs has been modified accordingly. With those changes, we do not see the purpose of normalizing the INP spectra to an estimated total aerosol particle surface.

Reviewer: Figure 5: Should you not actually show the variability you are referring to in the caption, e.g., with error bars?

Response: This is now done in the new Figure 6.

Reviewer: Line 340. I became confused already earlier in the paper as to whether or not repeated cycling involving heating and cooling were used. This fact should be moved forward in the methods.

Response: Done. It is now moved to section 2.4.3.

Reviewer: Lines 366-367: "…because the probability of any drop freezing during any isothermal experiment decreases with decreasing normal freezing temperature below the isothermal temperature." I did not understand this at all. This is not intuitive without some additional explanation.

Response: Extra explanation with more precise language is now provided:

> "The practically sigmoidal-like distribution of normal freezing temperatures (Fig. 8) arises because the average probability of any drop freezing per unit time during any isothermal experiment must, when comparing all such drops, decrease with decreasing normal freezing temperature below the isothermal temperature among them. For a given drop, this probability is governed by the immersed surface area of INP material and its composition. These underlying

*quantities also follow a statistical distribution among drops. Drops with the most depression of the normal freezing temperature below the isothermal temperature would be expected to contain less, or less efficient, INP material than most that freeze, causing these rare drops to freeze only on unusually long time-scales in the isothermal experiment."*

Reviewer: Figure 8: Question of clarification. The "freezing fraction" here is on the basis of the droplet population, correct? Or on the basis of the final number frozen? This figure is difficult to read due to the use of a logarithmic scale on the x-axis. What do these look like with time on a linear scale of say hours starting from time zero? That would seem to be a starting point, before plotting them this way.

Response: This new Figure 9 has been modified to include a linear time scale as required.

Reviewer: Figure 9: I especially cannot understand this figure. Should not the end total ice fraction be larger than the initial ice fraction in all cases? Why would this ratio be less than 1? Or does 0.5 mean a 50% increase and so on? If so, the y-axis needs redefinition.

Response: Yes, this was a typing error. The label has been corrected on the new Figure 10. It should have been "eventual fractional increase in frozen fraction" or just "chi".

Reviewer: Figure 10 and discussion around it: This is an interesting figure that suggests to me that the INPs generally have similar freezing behaviors that are describable in nearly a chemical kinetic fashion (e.g., DeMott et al., 1983, J. Clim. Appl. Meteor., doi:10.1175/1520 0450(1983)022<1190:AAOCKT>2.0.CO;2).

Response: This is an excellent point and we now cite the DeMott paper for this in the discussion around the new Fig. 11 (line 615).

Reviewer: I wondered though what N and Nice exactly are. They are the same? These are not defined anywhere, either in the manuscript or the caption. Are they the total number of drops? Or the total number frozen after xx hours? If looking at the change in freezing rate, it seems like the reference should be the total INP population, not the drop number that may or may not reflect an INP per drop. I think that the relevant value is Nice,infinity, in Eq. (1), but it is unclear how this is determined or estimated. I think this is finally stated later, perhaps at line 408. Hence, the introduction of these things is a bit out of order.

Response: $N_{ice}$ was the number of drops frozen as a function of time, t, since the start of the experiment at 0 degC normalized by the total number of drops. We have corrected the introduction of these definitions. We have now changed the terminology with "f" now used instead of "N" for frozen fractions throughout the paper.

Reviewer: Figure 10 caption: "Occasional negative rates are not plotted." How do you get a fractional freezing rate that is negative?

Response:  What we meant was that our numerical estimate using a finite difference scheme occasionally yielded unrealistic negative values of the freezing rate.

Reviewer:  Line 399: Now Nice(t) is a frozen fraction? This is very confusing. Frozen fraction or number terms need careful definition before they are used. N normally refers to number, but I sense it is being used for both number and fraction in this paper.

Response: No, N_ice(t) was always a frozen fraction.

To clarify this, we have changed the terminology to "f" for fraction everywhere and now define all symbols in a new Appendix B.

Reviewer: Line 534: That there are multiple INPs in each drop is a risk? This is a fact of the method, at least for cooling ramps that extend over the mixed-phase regime. It is accounted for in most immersion freezing analyses, ala Vali (1971).

Response: Yes, the reviewer is correct that this was poorly phrased. As we are using environmental samples, all drops will of course contain an ensemble of different  INPs. 'Risk' here refers to the risk that we have a masking effect of more effective INPs activating before we can get information about less active INPs.

**We have updated the manuscript to clarify this (line 782).  In the next paragraph (line 787) we explain why we do not think this is a real problem for our experiments.**

Reviewer: Lines 571 to 572: I do not trust that you can make such correspondences at all. The study is not sufficiently detailed to do so. You do not even know if sources are organic or inorganic, for sure.

Response:  We never claimed to make any such correspondences with certainty.

What we wrote was fine:

> "*From the classification of our samples (section 2.2) from the Hyltemossa field station, some likely correspondences may be hypothesized*".

The paper's goal is chiefly to advance models in light of fresh lab results.  Modellers do not have the luxury of waiting until perfectly representative observations become available before representing a potentially critical process in the model.  When imperfect or incomplete data exists that allows a rudimentary representation of the target process in a model, then to ignore this may introduce more bias into the simulations than to include it.  We think that is the situation here.

**Regarding Figure 6, there is no evidence of any problem with the INP concentrations we measured in each type of airmass, as we argue (Sec. 3.1.1).   There is commonality with some other published**

**studies (e.g. Si et al., Irish et al.) showing observations of INPs in marine environment, as we explain in the results section (e.g. lines 470-500).**

Reviewer: Line 590: Possibly. It remains to be seen how useful the approach will be. But this is where we are left hanging. If the total INP number is not so much greater than measured deterministically with a small temperature shift, aren't the conjectures of Westbrook and Illingworth invalid already?

Response:  See the above comment on this hypothesis.  To be sure, we need to simulate with a cloud model since there are so many nonlinear feedbacks and other processes occurring.  We include new text to discuss this hypothesis in the concluding section (lines 935-940).

Reviewer: Line 644-645: It may enable it, but odd to highlight a single study without proving it. Can we expect that robust simulations will be achieved? I suggest that the emphasis on saying that single events can be used to target certain aerosol types be removed from this paper, and postulated instead in the next one that seems in preparation.

Response:  To avoid emphasizing a single event unduly we have added an extra reference to studies simulating long-lived cold clouds (line 976).

**Data availability:** No statement was made. Will the data be made available somewhere? This is important.

Response:  Yes, the availability of data is now stated (line 1001).   We will archive data on public *'ftp-servers'.*

**Editorial notes**

Reviewer: Line 24: decline, rather than declines.

Response: Corrected as required.

Reviewer: Line 124: do you really need the word "Background", which is not really quantifiable. Just say you collected aerosol data?

Response:  Corrected.

Reviewer: Line 137: I would omit "assumed to be likely". It does not help qualify that there is no way to be certain about influences, considering limited information on aerosol compositions.

Response:  Done.

During revision, we have augmented the information on aerosol composition (see Table 1).

Reviewer: Line 616: "for what we have inferred to be representative of mineral…"

Response:  Corrected as required.

---

## Referee Report (RR1)

**General Comments**

The authors have done a commendable job of addressing my points and those of other reviewers. I still very much appreciate this study overall, and wish to see it formally in the literature. I also appreciate the effort made to more fully characterize aerosol compositions involved in the classification of cases, to clean up the equations and make more precise explanation, and to add methodological details. While the aerosol chemistry does clearly indicate differences between the cases, and the statistical differences of some of the INP populations is supported, my opinion is that it is no clearer that such categorizations tag reliably specific INPs or makes them any more meaningful for categorization in models. This is acknowledged in the paper. Nevertheless, I think it is entirely possible that in follow-ups to this study, the authors or other authors will use these categories or consider the polluted or high black carbon or high dust cases as representative of more specific INPs categories in numerical models. I think that would be in error, as more effort is needed in future research to get to the point of a true "closure" study. This regardless of the apparently noble efforts modelers are making to deal with imperfect observational data, to paraphrase one of the more ludicrous responses I have seen (i.e., modelers not having the "luxury" to wait). It is well known that biological/biogenic INPs play a significant role in the atmosphere at temperatures >-20C, and also that they are the most difficult category to quantify and pin down. Over a land location especially, these must be present at some level always, and they cannot simply be characterized by total biological particle concentrations from a given sensor. INPs are always more specific, even within some broad categories, and this is likely especially true for the most efficient ice nucleators that are active at modest to moderate supercooling. In the end, I think that readers will focus most on the quantification and possible generalization of time dependence for ambient INPs (i.e., the focus of the title and what is in the abstract), and will focus less on the classifications of cases that represent some unknown mix of different INP types in all cases. I list a few specific points below for possible attention.

**Specific Comments**

1) In the new discussion circa line 942, is it necessary only to point out the utility of findings only for INP measurements that have short residence times (i.e., real-time measurements like a CFDC)? One imagines that it means that slow cooling or isothermal measurements, while helpful for validating results such as presented in this paper, are overall not necessary even for classical immersion freezing measurements.

2) The authors note that they found weaker time dependence across cases than found in the literature, and they reference Herbert et al. (2014) as supporting that different INP types should show little difference. That is not a result highlighted in Herbert et al., as far as I read that paper, and it is also the case that it was heavily focused on inorganic materials, especially minerals. Also, Wright et al. (2013; doi:10.1002/jgrd.50365), Fig. 3, shows that some types of INPs vary in time dependent character when isolated. But this all leaves me to wonder, again, if there is any reason to think that the different cases are representative for future application to specific INP scenarios, versus a more generalized time dependence to use for any INP category (at least to the extent that data at -15C characterizes things across temperature of relevance to mixed-phase clouds)? I know that categorization to fit model categories was an imagined goal of this paper, but it remains the one that has the least clear support. I would even say that the careful effort put

in to attempting to categorize aerosols and air mass characteristics as related to INPs stands as testament to how extremely difficult it is to characterize INP scenarios, since INPs are but a limited fraction of the entire aerosol population. The INP data are certainly representative for the region, but parsing them out to different sources is not possible from the data collected. The story on time dependence is the true feature result here.

3) Regarding new Fig. 5 and discussion around it (much appreciated), background is important whether in the temperature regime where it is "rare" or where the frozen fraction gets very high. The question I have, and which needs a clear statement, is if any corrections are actually applied, and if the rare occurrences at higher temperatures are simply ignored. I am not judging, just saying that it needs to be said. Some would average all background cases and do corrections, but one at least needs to say what was decided.

4) The data availability statement is not up to current standards and expectations, in my opinion.

---

## Author Response (AR2)

**Summary of Author Responses to Reviewers**

**Reply to Reviewer 1**

**Author Response**

We are grateful to the reviewer for their comments on the manuscript, which have helped to improve it.

We have corrected some problems with the original Figure 11. Also, we have explained our isothermal formulation (now Eq (2)) in terms of a new Eq (1) expressed in terms of the number of unfrozen drops remaining.

**Point-by-Point Comments**

Reviewer:  General comment: The revised manuscript is clearly improved compared with the initial submission. The authors have taken the concerns of the anonymous reviewers seriously and modified the manuscript accordingly. The writing of the manuscript has been improved and imprecise formulations have been reformulated more clearly. The reasoning of the authors can now be followed throughout the mnuscript and the aims and purpose of the study have become clearer. Specifically, the following improvements have been made:

- The relevant literature is discussed more fully in the revised manuscript.

- A detailed discussion of the composition of the different aerosol classes has been added. Although the physicochemical characterization given in the revised Table 1 exhibits some gaps, it helps to better classify and compare the different samples. It is more clearly discussed to what extent the analyzed sample types are able to represent the proposed aerosol classes.

- Statistical tests have been added to analyze whether the samples exhibit significantly different freezing spectra.

- The addition of a pure water-freezing curve helps to judge that there is no relevant influence of the background signal on the freezing spectra of the samples.

- A discussion of the origin of time dependence was added and it was concluded that the time dependence is mostly stochastic.

- The empirical approach to treat the time dependence is now explained better. An appendix with a table containing the definition of the mathematical symbols has been added that helps to understand the equations.

- The conclusions have been extended. Most importantly it is explicitly stated that the stochastic component of immersion freezing is minor compared with the temperature dependence of INP freezing.

This is an important finding of this study, in view of a purely stochastic representation of immersion freezing that has been proposed in a recent study (Knopf et al., 2020).

Response:  We are glad to see the reviewer appreciates our modifications.

Reviewer:  Yet, the valuable input from Gabor Vali has only partially been considered in the manuscript revisions. The suggestion to show the results in terms of the freezing rate, i.e. fraction of unfrozen droplets that freeze per time, has been taken up in Fig. 11 by adding a panel (b) displaying the freezing rates of the samples. Unfortunately, throughout the rest of the manuscript, the authors stuck to an analysis in terms of their fractional freezing rate, which has much less physical meaning. Yet, even if the authors do not want to convert their fractional freezing rate to the real freezing rate, the revised manuscript can be published after the following minor revisions:

Response:  Yes and no.

In fact, our isothermal freezing formulation, now Eq (2), is derived from an assumption that the unfrozen fraction decays quasi-exponentially (with a relaxation time that depends on time), as is apparent from our inspection of observations by Knopf et al. (2020).

We now make this derivation explicit in the paper by inclusion of a new Eq (1) about the time-dependency of the unfrozen fraction.   However, we introduce two new features here:  (1) an assumption that the relaxation time depends on time, as we infer from that inspection of the literature, and (2) the notion that drops which can *never* freeze during the isothermal experiment must be irrelevant to the time evolution of the unfrozen fraction, which should therefore exclude them in its definition.

We have added fresh text to clarify (new Eq (1) and lines 667-675).

Reviewer:  Abstract: The abstract could be extended by a sentence that the authors have added to the conclusion section in the revised manuscript: "Any purely stochastic model of INP activity, assuming that the fractional freezing rate of all unfrozen drops is constant, would predict very high frozen fractions after a certain time, which would be inconsistent with our measurements. Instead, the statistical variability of efficiencies among INPs must be accounted for with any application of stochastic theory." Or a similar statement.

Response:  Done (lines 28-30).

Reviewer:  Line 18: it should be mentioned that all samples were collected at the same station. The name of the sampling station could also be given.

Response:  Done (line 18).

Reviewer:  Line 75: "since INPs influence the ice concentration observed" instead of "since INPs determine the ice concentration observed" would be more precise since updraft velocity is also a major determinant of ice crystal number density in clouds.

Response:  Done (line 77).

Reviewer: Line 76: do you mean here secondary ice production? If yes, it should be stated explicitly.

Response:  This includes both SIP and homogeneous freezing.

Reviewer: Line 77: It is more than "beneficial" to simulate the first ice in mixed-phase clouds accurately. Consider to replace "beneficial" by "crucial" or something similar.

Response:   Yes and no.

Naturally, it is necessary to predict correctly whether there is *any* onset of first ice.   In some clouds (E.g. Sassen et al. 2003), subzero temperatures are too warm for any heterogeneous ice nucleation.

However, deep precipitating clouds will have an ice concentration that is determined by the inter-play between ice multiplication (defined as the positive feedbacks of microphysical processes involving SIP) and thermodynamic limits on the ice concentration (e.g. onset of subsaturation).   So, an order of magnitude error in the concentration of the first ice may be irrelevant to overall accuracy of the eventual ice concentration for such clouds.

Some clouds are too thin for ice precipitation and hence for SIP too.  For these, one needs to simulate the first ice accurately as it is the only ice.

**We leave the text unchanged.   "Beneficial" is nuanced and a fair compromise (line 79).**

Reviewer: Lines 104–105: do you mean here that also the freezing rate should decline exponentially? Yet, for stochastic freezing, the freezing rate, i.e. the fraction of unfrozen droplets that freeze per time, remains constant. Just the absolute number of droplets that freeze per time decline together with the number of unfrozen droplets. In your terminology, the freezing rate seems to be termed a fractional rate. Then. it is absolutely unclear what you mean by "freezing rate". As your terminology differs from the common terminology, there are many sources for confusion. If you want to stick to your terminology, it might be best to just remove "freezing rate" from the sentence.

Response:  We agree.

There was confusion created by us using our own terminology that deviated from what appears to the common terminology.

**For consistency with the common terminology, we now use "freezing rate" to refer to the fractional rate of change of the number of unfrozen drops throughout the paper.**

Reviewer: Line 106: It should be added "for immersion freezing" after "This is seldom observed", to make clear that this sentence does not refer to homogeneous ice nucleation, which is indisputable stochastic.

Response:   Agreed.   This is replaced as required (line 108).

Reviewer: Lines 259–260: SiO2, CaO, and Al2O3 are not minerals present in mineral dusts but the oxides that form after ignition of the samples, which is performed to determine the elemental composition. Mineral composition of dusts can e.g. be found in Murray et al. (2012), Kaufmann et al. (2016), and Boose et al. (2016).

Response:  Agreed and text is changed as required with inclusion of these new references (line 266).

Reviewer: Line 290 and Table 1: The PM10 and ACSM derived aerosol concentrations are not always consistent. E.g. PM10 of the combustion dominated sample is 20.85 ug/m3, but the sum of Org, NH4, Cl, NO3, and SO4 adds up to 24.5 ug/m3. The reason for such discrepancies should be discussed.

Response: We have added a little more technical information about the PM measurements regarding size range and inlet heating in subsection 2.2.2.1 (lines 229-231).

At the end of subsection 2.2.2.2 (lines 337-342) we have added the following paragraph:

"In general, there was a clear tendency of the optically measured $PM_1$ to be lower than what was obtained from the summation of individual chemical components detected in the $PM_1$ fraction. We mainly ascribe the offset between the different measurement approaches to (i) the size range up to 0.18 μm not being detected optically, and (ii) the heated inlet before the optical measurements, which may lead to evaporation of (semi-) volatile aerosol particle components. The latter effect is likely to be more pronounced when nitrate species and (semi-) volatile organic species contribute significantly to the $PM_1$ (e.g. Huffman et al., 2009)."

Reviewer: Line 326: "aerosol components" might be a more appropriate wording than "aerosol properties".

Response:  Done (line 333).

Reviewer: Lines 473 – 477: this discussion is quite confuse and should be formulated clearer.

Response:  We agree the paragraph was confusingly over-written.

It has now been re-written (lines 485-500).

Reviewer: Table 1: Bio Trak OPC showed elevated number concentration during the dust event from February 20 to 26. As mineral dusts can also be fluorescing, the Bio Trak signal should not be directly identified with PBAP in the presence of mineral dust as it is done in Table 1.

Response:  Agreed.  The heading in the table has been changed accordingly to "Fluorescing particles" instead of "PBAP".   The caption has been modified (lines 1591-1594).

Reviewer: Lines 492–494: what is meant by "relatively small"? Please specify. What is meant by "a likely candidate"? The marine biogenic components? If yes, it should be "candidates".

Response: We have included these changes (lines 518-519).

Reviewer: Line 563 and Figure 8: it might be more meaningful to indicate the min–max value range rather than one standard deviation.

Response:  We have updated Figure 8 to display the min-max value range for each drop instead of the standard deviation. We have also updated the text (lines 583-589).

Reviewer: Line 579: again, it might be meaningful to also state the min–max difference.

Response: We have added the requested information in more detail in Table 2.

Reviewer: Line 594–596: the definition of fice(t*) seems imprecise. More precisely, it should be "the fraction of droplets freezing starting from the isothermal phase (i.e. f_fice(t*=0) = 0).

Response:  There is a misunderstanding here.  f_ice is the total number of drops frozen since the start of cooling at 0 degC (when t* < 0).   f_ice is zero at 0 degC and is non-zero when the isothermal temperature is first reached (t* = 0).

We have modified the text to clarify (lines 620-622).

Reviewer: Line 605: The fractional rate of freezing of unfrozen drops is the freezing rate! Please change accordingly.

Response:  Text is added to clarify (line 632-636).  We now call it the 'real freezing rate'.

Reviewer: Line 606: should this be fice(t*)/dt* as on line 603?

Response:  The derivative is the same quantity, whether written as d fice(t*)/dt*  or as dfice/dt*.

**We now include the functionality '(t*)' in the written expression, to avoid confusion (line 634).**

Reviewer: Lines 606–608 and Figure 11b: the presentation of the data in terms of the freezing fraction per fraction of unfrozen drops makes more sense than the presentation per fraction of frozen droplets, as the former represents the freezing rate, which should remain constant for stochastic freezing. It should be discussed why this quantity increases again for times since the start of the isothermal phase larger than $10^3$ s for most samples. Also check for correctness as this behavior seems inconsistent with Fig. 9. Maybe discuss the uncertainty of the data points. Was this evaluation done for the averaged (yellow) data points shown in Fig. 9? If yes, you should consider smoothing them before analysis. As it seems, their scatter is increased compared with the individual datasets through averaging the datasets.

Response:   We agree.

We now see that in the previous version of the paper, the lack of decrease of the fractional freezing rates after half an hour was due to a "noise floor" at long times due to too few raw data-points determining the plotted yellow points in Fig. 9 (averages among experiments in a sample) and increased random variability.

We have followed the reviewer's suggestion and now have included extra smoothing to eliminate any decrease in frozen fraction with time, before computing the derivatives for Figure 11.

**The new Figure 11 is included, showing a steady decrease with time in the real freezing rate throughout the isothermal period, without such a noise-floor (lines 636-637).**

Reviewer: Lines 614–615: what chemical kinetics are meant here? This should be explained.

Response:  Done (lines 644-646).

Reviewer: Line 616: what is the "natural time scale of freezing"? This should also be explained.

Response:  Done (line 648).

 We now write that "*The reciprocal of the real freezing rate is the natural time scale of the freezing at any instant*".

Reviewer: Line 626–627: this sentence is confusing. Do you mean: "such less efficient drops" instead of "such less efficient INPs"?

Response: No, we meant "such INPs that are less efficient" at nucleating ice.

The sentence is now modified to clarify (line 657).

Reviewer: Lines 654–657: Making the fraction of droplets freezing per time interval a function of the unfrozen droplet fraction would indeed be a physically more meaningful formulation. If it were formulated like this, 1/tau(t*) could indeed be viewed as the probability of unfozen drops freezing during the isothermal phase. Consider to revise the manuscript in this respect.

Response:  1/tau always was that probability of unfrozen drops freezing during the isothermal phase. We show this now with an extra Equation (a new Eq (1)) which is the basis for the same isothermal formulation as before (now Eq (2), previously Eq (1)).

Text is added to explain the new Eq (1) (lines 667-675).

Reviewer: Line 709: what is meant by "somehow"? Please specify.

Response:  We agree, the text was unclear and we now include some explanation of what we mean (line 747).

This assumed correspondence is discussed later, in Section 5.  So we now include a cross-reference just there (line 747).

Reviewer: Line 710: the way the equation is formulated, the temperature shift in Eq. (4) applies to all INPs. Shouldn't it then be "applied to all INPs"? The sentence should be changed accordingly or it should be commented why only "most INPs".

Response:  Yes, the term "all INPs" now replaces "most INPs" as required, and we add an extra sentence to clarify (lines 747-750).

The fixed shift downward is the approximation for the model; the shift upwards is what is real and is for the freezing temperature, which follows a statistical distribution instead of being fixed.

Reviewer: Line 831–832: this sentence is not complete.

Response:  Now altered to clarify (line 873).

Reviewer: Line 964: Does the statement in the bracket refer to the treatment of temperature dependence suggested in this study? Please clarify.

Response: Yes, it is the modelled temperature shift for the INP scheme that is being referred to.  It is now clarified (line 1007).

Reviewer: Line 974–976: this last sentence should be formulated in view of the discussion above, because it sounds as if the temperature dependence could indeed account for the observations of Westbrook and Illingworth.

Response:   The sentence is now re-phrased to avoid implying that we expect a role of time-dependence (line 1019).

**However, in light of ongoing simulations of cloud cases we now begin to wonder if the Westbrook and Illingworth hypothesis about this role just might be partly correct. "The jury is still out" on this question, as is now discussed in the text (lines 978-985).**

**A cloud simulation is needed to resolve this matter.**

Reviewer: Table 2: it should be stated whether low or high numbers stand for uniqueness.

Response:  We presume the reviewer meant "Table A2".  The caption is now modified to clarify.

Reviewer:  Table B1: The parameter "Q" should be explained better in the list of symbols: "passive tracer of what?" What is meant by "freezing level"? Why does Q have units of kg[air]-1? On line 836, a value of Q is given without units.

Response:  The problem with units is now corrected by introducing a new variable for the value of Q prescribed as unity outside the cold-cloud region.   The text and equations are slightly changed accordingly.

Reviewer: Figure 9b: Axis labels in panels (b) need to be enlarged to the size shown in panels (a).

Response:  We have updated the labels as requested.

Reviewer: Table 3: chi for the mineral dust influenced sample should be 1.48 instead of 1.5.

Response:  No, the chi value is from the exact measured values, which are not displayed.  Only two decimal places are displayed in the various columns.

Technical comments:

Reviewer: Line 59: The meaning of PK97 should be given here, at the first mentioning of Pruppacher and Klett, and not on line 91.

Response:  Corrected.

Reviewer: Line 585: "time increases" instead of "times increase".

Response:  The sentence is now changed (line 610).

Reviewer: Table 4: the upper value of the continental pristine sample is not correctly displayed.

Response: Does the reviewer mean the extra decimal for the upper bound for the "Continental polluted" sample?  -> 446 ?  Corrected.

Reviewer: Table 5: "°C" should be removed from "-16.3°C". The line numbers appear within the table.

Response: Done.

Reviewer: References:

Boose, Y., Welti, A., Atkinson, J., Ramelli, F., Danielczok, A., Bingemer, H. G., Plötze, M., Sierau, B., Kanji, Z. A., and Lohmann, U.: Heterogeneous ice nucleation on dust particles sourced from 9 deserts worldwide – Part 1: Immersion freezing, Atmos. Chem. Phys., 16, 15075–15095, https://doi.org/10.5194/acp-16-15075-2016, 2016.

Kaufmann, L., Marcolli, C., Hofer, J., Pinti, V., Hoyle, C. R., and Peter, T.: Ice nucleation efficiency of natural dust samples in the immersion mode, Atmos. Chem. Phys., 16, 11177–11206, https://doi.org/10.5194/acp-16-11177-2016, 2016.

Murray, B. J., O'Sullivan, D., Atkinson, J. D., and Webb, M. E.: Ice nucleation by particles immersed in supercooled cloud droplets, Chem. Soc. Rev., 41, 6519–6554, doiI:10.1039/c2cs35200a, 2012.

**Replies to Reviewer 2**

**Author response**

We are grateful to the reviewer for their effort in scrutinizing the manuscript.

It is a fair criticism to make of the paper about the lack of certainty in attribution of INP types to some of the observed freezing behavior of samples. However, more error could be introduced into a model from treating all INP types with the same time-dependence treatment (same expression for the temperature shift) than for making use of the likely attribution noted below and in the paper.

**Point-by-point Responses**

**Reviewer: General Comments**

The authors have done a commendable job of addressing my points and those of other reviewers. I still very much appreciate this study overall, and wish to see it formally in the literature. I also appreciate the effort made to more fully characterize aerosol compositions involved in the classification of cases, to clean up the equations and make more precise explanation, and to add methodological details.

Response: We appreciate the encouraging and constructive style of the review.

Reviewer: While the aerosol chemistry does clearly indicate differences between the cases, and the statistical differences of some of the INP populations is supported, my opinion is that it is no clearer that such categorizations tag reliably specific INPs or makes them any more meaningful for categorization in models. This is acknowledged in the paper. Nevertheless, I think it is entirely possible that in follow-ups to this study, **the authors or other authors will use these categories or consider the polluted or high black carbon or high dust cases as representative of more specific INPs categories in numerical models. I think that would be in error, as more effort is needed** in future research to get to the point of a true "closure" study.

Response: We disagree that such a future initiative would constitute an "error".

We have provided much experimental data characterizing the various airmass types and their freezing behaviours in our experiments in the present paper.

It is curious that the review provides no evidence to falsify the likely attributions we have made about dominant INPs in the various samples (e.g. the mineral dust influenced sample we assumed to be probably reflecting the freezing by mineral dust INPs).

**We now have included extra text with more analysis of our own chemical characterization to indicate likely INP species dominating the freezing of the samples (lines 485-504).**

**Creative modellers do not have the luxury of waiting years for perfect lab data to become available, as noted already.**

Reviewer: This regardless of the apparently noble efforts modelers are making to deal with imperfect observational data, to paraphrase one of the more ludicrous responses I have seen (i.e., modelers not having the "luxury" to wait).

Response: **Our responses were fine. It is difficult to respond to such a tangential negative comment in a review when no coherent argument is provided.**

Regarding our philosophical comment about modellers not having the "luxury" of waiting years for perfect empirical data to become available, it was quite reasonable.

A key difference between the activities of state-of-the-art modeling and experimental observations is that any modeler must show that their model is realistic before using it for scientific questions. That involves comparing the model with observations of the real phenomenon being simulated, which consists of many processes and is to be represented somehow in all its complexity. The toughness of that challenge is why the modeler cannot wait for perfection of published experimental results when faced with any given incompletely characterized process during development of the model. The modeller must attempt some representation of it, since not to do so likely introduces more bias in the overall simulation of the wider phenomenon than use of albeit imperfect data.

This is a perpetual dilemma that modellers will always face, one could argue. Two of the co-authors are modellers, so we speak from experience. In fact, this dilemma is in a sense a reason for the current project happening in the first place, about how to treat time-dependence in a cloud model. When planning the project, we knew there would be such difficulties in the lab observations and that any data we would acquire would be incomplete.

It goes without saying that in the above comment, we are not referring to the simplest modeling that involves little validation or development of a model, or perhaps uses a model already created elsewhere.

**By contrast, observationalists can focus on characterizing a single process in any study, or can simply record their observations of a complex phenomenon. They face no such necessity to address many processes simultaneously.**

**Of course, we are not saying that generally creative modeling is more challenging than observations overall. The nature of the difficulties differ between both types of scientific activity.**

Reviewer: It is well known that biological/biogenic INPs play a significant role in the atmosphere at temperatures >-20C, and also that they are the most difficult category to quantify and pin down. Over a land location especially, these must be present at some level always, and they cannot simply be characterized by total biological particle concentrations from a given sensor. INPs are always more specific, even within some broad categories, and this is likely especially true for the most efficient ice nucleators that are active at modest to moderate supercooling.

Response: Agreed.

Even if one could measure the concentrations of individual types of bioaerosol (fungal, bacterial, pollen…) in a given ambient aerosol sample, as indeed we did a recent study (Patade et al. 2020), a grave

experimental obstacle is that in the freezing experiments it is uncertain which aerosol types from the immersed sample caused drops to freeze.  It is not an insuperable obstacle but would take much effort to overcome.

Reviewer:  In the end, I think that readers will focus most on the quantification and possible generalization of time dependence for ambient INPs (i.e., the focus of the title and what is in the abstract), and will focus less on the classifications of cases that represent some unknown mix of different INP types in all cases. I list a few specific points below for possible attention.

Response:  Perhaps.

Reviewer: Specific Comments

1) In the new discussion circa line 942, is it necessary only to point out the utility of findings only for INP measurements that have short residence times (i.e., real-time measurements like a CFDC)? One imagines that it means that slow cooling or isothermal measurements, while helpful for validating results such as presented in this paper, are overall not necessary even for classical immersion freezing measurements.

Response:  This comment seems rather subjective.   The paper has not proven that time-dependence is unimportant for all clouds.  Thin stratiform cloud without much SIP, as observed by Westbrook and Illingworth (2013), might be affected by time-dependence of ice nucleation somehow, because it lacks SIP.  We are trying to ascertain this by simulations now for another project.

Reviewer:  2) The authors note that they found weaker time dependence across cases than found in the literature, and they reference Herbert et al. (2014) as supporting that different INP types should show little difference. That is not a result highlighted in Herbert et al., as far as I read that paper, and it is also the case that it was heavily focused on inorganic materials, especially minerals.

Response:  This comment prompted us to re-read the Herbert et al. paper and we now agree there was a problem with the way we cited it.  Herbert *et al.* never wrote that the degree of time-dependence is independent of INP composition.

**The citation of Herbert *et al.* has now been corrected (lines 128-131).**

Reviewer:  Also, Wright et al. (2013; doi:10.1002/jgrd.50365), Fig. 3, shows that some types of INPs vary in time dependent character when isolated. But this all leaves me to **wonder, again, if there is any reason to think that the different cases are representative for future application to specific INP scenarios, versus a more generalized time dependence to use for any INP category** (at least to the extent that data at -15C characterizes things across temperature of relevance to mixed-phase clouds)?

Response:  Yes.

We agree that the degree of time-dependence varies with INP composition.

We see the various samples display differing degrees of time-dependence, which would reflect different INP types.  Also, the higher the initial frozen fraction, the lower the time-dependence and vice versa.

So we think that the various cases are likely to be representative of specific INP types, the identity of which is uncertain for our samples.  Our mineral dust influenced sample displays a stronger time-dependence than the other samples, consistent with Herbert et al. also observing a stronger time-dependence (montmorillinite).   There are reasons to suppose that the IN activity of the rural continental sample is likely influenced by PBAPs, since other components are low, and since the temperature gradient of atmospheric INPs is weak towards the warmest temperatures in Fig. 6, where PBAP-IN are uniquely active, (lines 500-504 and 859-863).

**However, we also argue that the differences in the degree of time-dependence observed among samples are rather limited.  So, lack of certainty in inferring the dominant INP type for each is not a prohibitive problem for modeling overall effects from time-dependence, as we now say in the text (lines 863-865).**

Reviewer:  I know that categorization to fit model categories was an imagined goal of this paper, but it remains the one that has the least clear support. I would even say that the careful effort put in to attempting to categorize aerosols and air mass characteristics as related to INPs stands as testament to how extremely difficult it is to characterize INP scenarios, since INPs are but a limited fraction of the entire aerosol population. The INP data are certainly representative for the region, but parsing them out to different sources is not possible from the data collected. The story on time dependence is the true feature result here.

Response:  Well, the mineral dust influenced sample originates from the Sahara and the degree of time-dependence differs from say the continental pristine sample.   Mineral dust is the most prolific INP type active at the temperatures we studied (near -15 degC) in the background troposphere.  So, it is likely that the freezing behavior for that sample reflects that of mineral dust IN.

**Similarly, there are reasons to suppose that the rural continental sample may have active INPs influenced by PBAPs, partly since it was taken in a warmer season and due to the measured composition of low amounts of other components (lines 500 and 863).  Also, its active atmospheric IN seems enhanced at the warmest subzero temperatures (see the gradient for the rural continental sample in Fig. 6).   PBAP-IN are often identified by their activity at uniquely warm temperatures.**

Reviewer:  3) Regarding new Fig. 5 and discussion around it (much appreciated), background is important whether in the temperature regime where it is "rare" or where the frozen fraction gets very high. The question I have, and which needs a clear statement, is if any corrections are actually applied, and if the rare occurrences at higher temperatures are simply ignored. I am not judging, just saying that

it needs to be said. Some would average all background cases and do corrections, but one at least needs to say what was decided.

Response:  On average, we observe a frozen fraction of 0.01 for a temperature of -20°C for the ultra-pure water experiments presented in Fig. 5. As can be observed from Fig. 7, the frozen fraction was typically between 0.8 and 1 at a temperature of -20°C for the ambient samples. Hence, if the substrate potentially could induce a very minor frozen fraction around that temperature – it is a lot more likely that INPs in the samples already induced freezing at a higher temperature in those few potential cases. Thus, we find no reason to correct for that background in the results presented in Fig. 6 – and any such minor correction would not influence the presented results.

We would not expect any low-quality slides to bias the presented results significantly since we observed very high reproducibility between different droplet populations for the same sample (Fig. 7).

The following statement has been included early in section 3.1.1: "The presented INP spectra have not been corrected for the background, as the background was negligible relative to INP concentrations in the ambient samples."  (lines 466).

Reviewer: 4) The data availability statement is not up to current standards and expectations, in my opinion

Response:  The statement has been improved.